# Gut microbial factors predict disease severity in a mouse model of multiple sclerosis

Alex Steimle[1,6], Mareike Neumann [1,2,6], Erica T. Grant [1,2], Stéphanie Willieme[1], Alessandro De Sciscio[1], Amy Parrish[1,2], Markus Ollert [1,3], Eiji Miyauchi[4], Tomoyoshi Soga [5], Shinji Fukuda [5], Hiroshi Ohno [4] & Mahesh S. Desai [1,3] ✉

Gut bacteria are linked to neurodegenerative diseases but the risk factors beyond microbiota composition are limited. Here we used a pre-clinical model of multiple sclerosis (MS), experimental autoimmune encephalomyelitis (EAE), to identify microbial risk factors. Mice with different genotypes and complex microbiotas or six combinations of a synthetic human microbiota were analysed, resulting in varying probabilities of severe neuroinflammation. However, the presence or relative abundances of suspected microbial risk factors failed to predict disease severity. *Akkermansia muciniphila*, often associated with MS, exhibited variable associations with EAE severity depending on the background microbiota. Significant inter-individual disease course variations were observed among mice harbouring the same microbiota. Evaluation of microbial functional characteristics and host immune responses demonstrated that the immunoglobulin A coating index of certain bacteria before disease onset is a robust individualized predictor of disease development. Our study highlights the need to consider microbial community networks and host-specific bidirectional interactions when aiming to predict severity of neuroinflammation.

Patients with autoimmune disease exhibit distinct gut microbiota compositions compared with healthy controls[1], including in multiple sclerosis (MS)[2]. Thus, to develop patient-targeted microbiota modulations, a necessary precondition is to examine whether susceptibility or progression of MS can be predicted using the microbiota composition (Fig. 1a). A common approach to elucidate MS-promoting microbial predictors entails comparison of bacterial relative abundances between patients and healthy controls[2–9]. Certain differentially abundant bacteria identified across different MS cohort studies tend to be concordant, for example, increased abundances of *Akkermansia*[2,5,7–9] or decreased abundances of *Prevotella*[3,6,8] in patients compared with controls. However, these case–control studies are not designed to

explain inter-individual differences in disease progression[10]. Therefore, it remains challenging to reliably link taxa abundances across individuals to microbiota characteristics that impact MS disease course.

Experimental autoimmune encephalomyelitis (EAE) is a routinely utilized pre-clinical mouse model to examine causality between the microbial risk factors and development of autoimmune neuroinflammation[11–14]. However, the translatability of such an approach—studying the causality of a singular species alone or within mice harbouring a relatively consistent, specific-pathogen-free (SPF) microbiota—to the plethora of distinct microbiota compositions found in MS patients remains unexplored[1]. Although certain EAE-promoting intermicrobial interactions were reported[11,15], the mutual impact between

[1]Department of Infection and Immunity, Luxembourg Institute of Health, Esch-sur-Alzette, Luxembourg. [2]Faculty of Science, Technology and Medicine, University of Luxembourg, Esch-sur-Alzette, Luxembourg. [3]Department of Dermatology and Allergy Center, Odense Research Center for Anaphylaxis, University of Southern Denmark, Odense, Denmark. [4]RIKEN Center for Integrative Medical Sciences, Yokohama City, Kanagawa, Japan. [5]Institute for Advanced Biosciences, Keio University, Yamagata, Japan. [6]These authors contributed equally: Alex Steimle, Mareike Neumann. ✉e-mail: mahesh.desai@lih.lu

the background microbiota and potential microbial risk factors on disease-promoting properties of the microbiota is poorly understood.

To evaluate common approaches to identify disease-associated commensals, we implemented a prospective cohort study design in mice of varying genetic backgrounds and diets, harbouring distinct complex microbial communities (Extended Data Fig. 1). We identified a potential disease-related bacterial feature and thoroughly evaluated in a gnotobiotic setting whether this feature is causally linked to EAE. We provide evidence that conclusions made from microbiome analyses that consider only relative abundances of certain bacterial features are unsuitable to reliably assess disease-mediating properties of the microbiota. Instead, we report on the IgA coating index of certain commensals that reflect individual EAE-promoting properties of reduced and complex microbial communities. These indices can be used to predict host-specific disease development.

## Results

### Mining neuroinflammation-associated gut bacteria in mice

Since the microbiota composition is known to impact EAE development, we first sought to shortlist potential EAE-associated bacterial taxa that might predict disease progression (Fig. 1a and Extended Data Fig. 1). To compare EAE progression in mice harbouring distinct complex microbiotas, we used SPF mice of three different origins and genotypes (Fig. 1b): (1) wildtype C57BL/6J from Charles River (CR), (2) mice deficient for the Muc2 protein ($Muc2^{-/-}$) and (3) homozygous controls ($Muc2^{+/+}$), which were littermate controls after breeding $Muc2^{+/-}$ mice. The $Muc2^{-/-}$ mice are expected to harbour a substantially different microbiota as a result of the impaired intestinal mucus layer[16,17], which would aid in critically assessing the validity of conclusions drawn from microbiota-related information only, without taking into account the host genetics. Mice of all backgrounds were fed a fibre-rich (FR) standard chow. Furthermore, we fed a subgroup of CR and $Muc2^{+/+}$ mice a fibre-free (FF) diet to further perturb the bacterial communities[18,19] (Fig. 1b). As expected, we detected distinct baseline microbiota compositions across all five background–diet combinations (Fig. 1c and Extended Data Fig. 2).

Next, to perform groupwise comparison of emerging disease phenotypes and to link these observed groupwise phenotypes to baseline differences in microbiota composition, we induced EAE in all mouse groups (Fig. 1d) and performed daily scoring (Extended Data Fig. 3). We observed a statistically significant difference in disease progression between the genotypes, with $Muc2^{-/-}$ mice being less susceptible to EAE induction compared with $Muc2^{+/+}$ and CR mice (Fig. 1e–g), regardless of diet (Fig. 1h). Intriguingly, the overall microbiota β-diversity was disconnected from the EAE disease course (Fig. 1i). As the five different genotype–diet combinations clearly split into two distinct EAE phenotype groups, we assessed potential EAE-relevant microbiota differences by comparing $Muc2^{-/-}$ mice (KO), providing a 'moderate' phenotype, with all Muc2-expressing mice combined (WT), irrespective of origin or diet, as they provided a 'severe' phenotype. We identified

11 differentially abundant genera that explained more than 70% of the variance detected in the Bray−Curtis distance matrix between WT and KO mice before induction of EAE (pre-EAE). Pre-EAE relative abundance of the genus *Akkermansia* alone: (1) explained 14.4% of said variance (Fig. 1j (right)); (2) correlated negatively with various EAE readouts upon induction of disease (Fig. 1j (left)); and (3) was significantly higher in $Muc2^{-/-}$ mice compared with WT counterparts (Fig. 1k). These results suggest possible disease-preventing properties of *Akkermansia*. However, these cross-sectional, groupwise microbiome analyses ignore any host-driven factors, an important limitation which we address in detail in the subsequent experiments.

### Causality of *Akkermansia muciniphila* in EAE severity

To verify a causal role of *Akkermansia* in EAE development and to evaluate its potential as a disease-risk predictor, we colonized germ-free (GF) C57BL/6N mice with a functionally characterized 14-member synthetic human microbiota (SM14), which includes *A. muciniphila*[18,20,21] (Fig. 2a and Supplementary Table 1), or a 13-member microbiota (SM13) lacking *A. muciniphila*[22–24]. Next, we induced EAE in these mouse groups (Fig. 2b). SM13-colonized mice exhibited a less severe EAE phenotype compared with SM14-colonized counterparts (Fig. 2c (left), d–f and Extended Data Fig. 4a), highlighting the general contribution of the microbiota to EAE development and the disease-driving role of *A. muciniphila* in the SM14 microbiota-based mouse model. As controls, we induced EAE in *A. muciniphila*-monoassociated (SM01) and GF mice (Extended Data Fig. 4b–d). SM01-colonized and GF mice provided a low-to-intermediate EAE disease phenotype (Extended Data Fig. 4b–d).

To evaluate whether changes in relative abundances of SM14-constituent strains might affect EAE disease course, we fed SM14- and SM13-colonized mice an FF diet, followed by EAE induction (Fig. 2b). GF mice were also fed an FF diet to exclude microbiota-independent but diet-mediated effects on EAE. In line with our previous studies[18,21–23,25], feeding SM14-colonized mice the FF diet resulted in significantly increased *A. muciniphila* relative abundances compared with equally colonized FR-fed mice (Fig. 2g). However, we did not detect any statistically significant differences in any EAE-associated readout between FR- and FF-fed mice harbouring the same microbiota (Fig. 2c (right), f). Removal of *A. muciniphila* from the SM14 community explained less than 28.5% of the variance for different EAE-associated readouts (Fig. 2e,f and Extended Data Fig. 4a (right)). Given that SM01-colonized mice only provided an intermediate disease phenotype (Extended Data Fig. 4b–d), our results suggest that: (1) the presence of *A. muciniphila* represents a potential microbial risk factor for severe EAE when combined with certain other strains from SM14 and (2) changes in its relative abundance within *A. muciniphila*-containing communities negligibly impact EAE disease course.

### *A. muciniphila*-related γ-amino butyric acid in EAE severity

To evaluate how *A. muciniphila* might alter microbiota function within SM14, we performed metabolomic and metatranscriptomic analyses

---

**Fig. 1 | Increased abundance of *Akkermansia* associated with lower neuroinflammation in mice with complex microbiota. a**, Summary of the central study objective. **b**, C57BL/6J mice from Charles River Laboratories (CR), $Muc2^{+/+}$ and $Muc2^{-/-}$ littermates housed under SPF conditions were fed a fibre-rich (FR, standard chow) or a fibre-free (FF) diet for 20 d. **c**, β-diversity analyses of faecal microbial communities after 20 d feeding on FR or FF diet. Left: non-metric multidimensional scaling (NMDS) plot based on a Bray−Curtis distance matrix. Right: principal coordinates analysis (PCoA) using a weighted UniFrac distance matrix. Ellipses show 95% confidence intervals. **d**, Mice depicted in **b** were subjected to EAE induction and observed for 30 d with daily scoring. **e**, EAE disease scores as a function of time. **f**, EAE-associated readouts analysed by one-way ANOVA followed by Tukey's post-hoc test (AUC and RelM) or Wilcoxon rank-sum tests (Max), with *P*-value adjustment using the Benjamini–Hochberg method. **g**, Sankey diagram of key event occurrence (in % of all mice within one

group) during EAE. **h**, Variance explained by diet and genotype (CR and $Muc2^{+/+}$ versus $Muc2^{-/-}$) comparing AUC among all five diet–genotype combinations ($n = 43$) as determined by eta-squared ($\eta^2$) calculation. **i**, Projections from **e** grouped by SusO and RelO, as defined in **g**. **j**, Spearman correlations between relative abundances of the indicated genera before EAE induction, with EAE-associated readouts (as defined in **f** and **g**) for each mouse across all five groups ($n = 28$). Statistically significant ($P < 0.05$) correlations by linear regression are indicated by asterisks (*). Horizontal bars (right) depict cumulative explained variance by genotype (CR and $Muc2^{+/+}$ versus $Muc2^{-/-}$) using the Bray−Curtis dissimilarity index for combinations of 11 genera ordered from highest single contribution (bottom) to lowest single contribution (top). **k**, *Akkermansia* relative abundance before EAE induction. One-way ANOVA followed by Tukey's post-hoc test. Mouse numbers are indicated on the respective panels and treated as biological replicates. Boxplots (**f,k**) show median, quartiles and 1.5 × IQR.

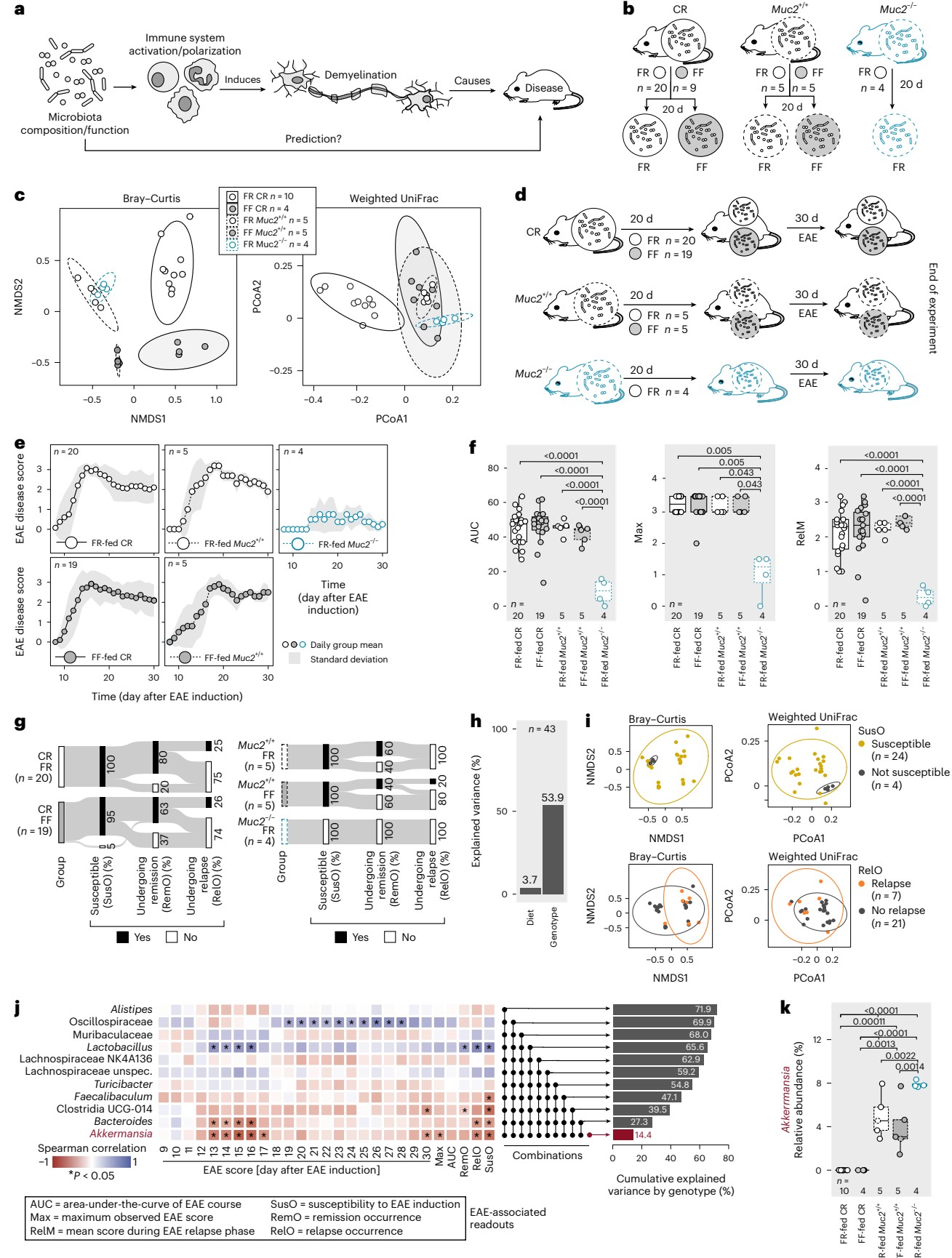

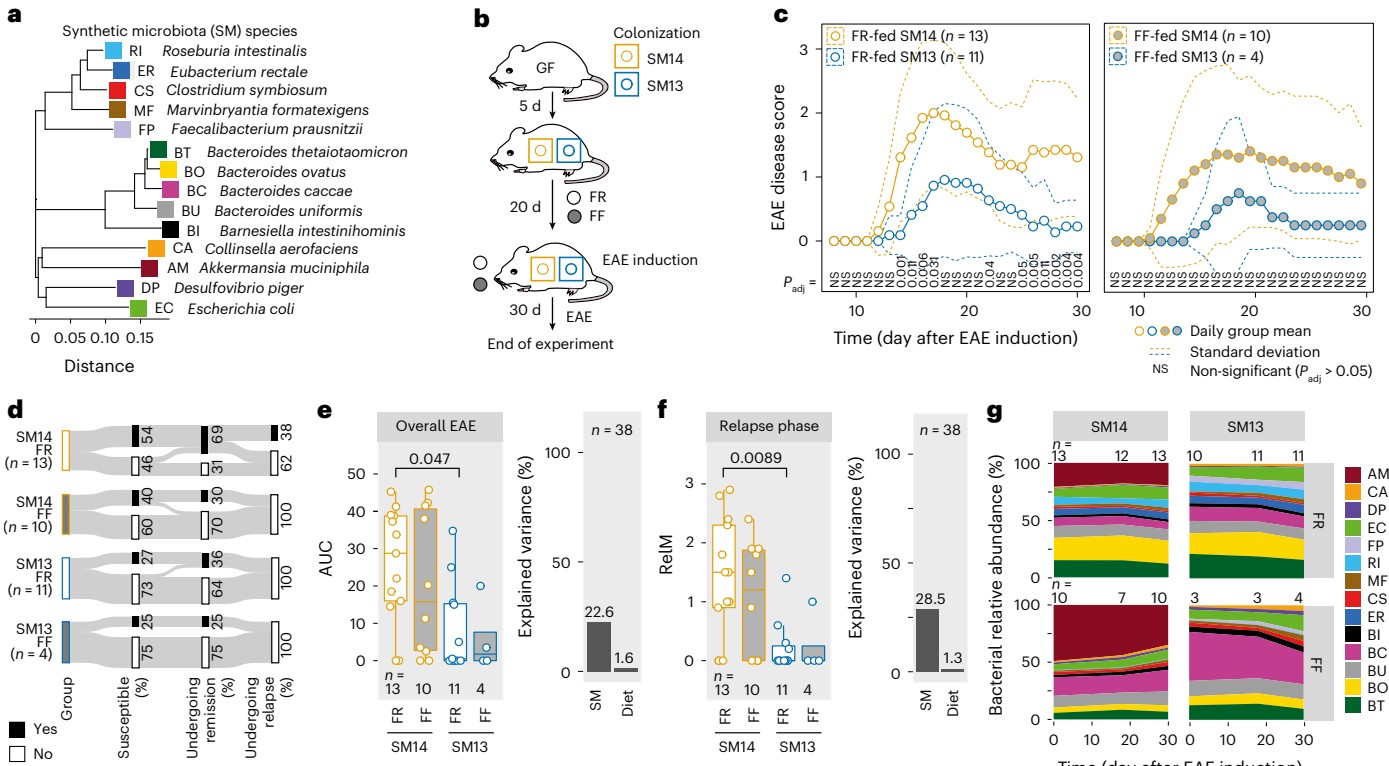

**Fig. 2 | A synthetic microbiota (SM) without *A. muciniphila* results in reduced neuroinflammation. a**, Neighbour-joining phylogenetic tree based on full-length 16S rRNA gene sequences of the SM14 strains (see Supplementary Table 1 for full strain designation and accession numbers for sequences to build tree). **b**, GF C57BL/6N mice were colonized with either SM14 or SM13 (SM14 without *A. muciniphila*) communities. After 5 d, both groups were switched to FR (standard chow) or FF diet. After 20 d feeding, EAE was induced in all mice and disease course was observed for 30 d. **c**, EAE disease scores as a function of time. Dashed lines represent s.d. Daily EAE scores were compared using a Wilcoxon rank-sum test with *P*-value adjustment using the Benjamini–Hochberg method. Left: FR-fed SM14- and SM13-colonized mice. Right: FF-fed mice harbouring the same SM combinations. Daily EAE scores for mice harbouring the same SM and fed different diets were statistically non-significant (*P* > 0.05) for all timepoints. **d**, Sankey diagram of key event occurrence (in % of all mice within one group)

during EAE. **e**, Left: AUC analysis of the disease course depicted in **c**. One-way ANOVA followed by Tukey's post-hoc test. Right: variance explained by diet or colonization (SM) when comparing AUC among all four groups, as determined using $\eta^2$ calculation. **f**, Left: mean EAE score during relapse phase (RelM, day 26 to day 30 after EAE induction). One-way ANOVA followed by Tukey's post-hoc test. Right: variance explained by diet or colonization (SM) when comparing RelM among all four groups, as determined using $\eta^2$ calculation. **g**, Mean relative abundances of SM strains over time (day after EAE induction), faceted by group. Species abbreviations are depicted in **a**. Missing values reflect times when faecal samples could not be collected. Mouse numbers are indicated on the respective panels and treated as biological replicates. FR-fed SM14 and SM13 data are from three independent experiments; FF-fed SM14 data are from two independent experiments; FF-fed SM13 data are from one experiment. Boxplots (**e**,**f**) show median, quartiles and 1.5 × IQR. All statistical tests were two-sided.

in caecal and serum samples from EAE-induced and non-EAE-induced GF, SM01-, SM13- and SM14-colonized mice (all only FR diet). The caecal metabolite profiles were similar between EAE-induced and non-EAE-induced groups harbouring the same microbiota, as well as between EAE-induced SM13-colonized and SM14-colonized mice (Fig. 3a,b and Supplementary Table 2). As broader metabolic profiles were disconnected from the EAE disease course (Fig. 3a), we reasoned that only a few caecal metabolites, if any, might causally influence the EAE disease course.

To identify such potential EAE-impacting metabolites, we developed a metabolite-of-interest screening pipeline evaluating four different criteria (Extended Data Fig. 5a–c; see 'Metabolite-of-interest screening pipeline' in Methods). We did not identify any metabolite of interest in serum samples (Supplementary Table 3), indicating that potential metabolite-driven impacts on EAE disease course occur locally in the intestine. Among the 14 metabolites (Fig. 3c) whose concentrations at the end of the disease course significantly correlated with AUC, only GABA (γ-amino butyric acid) emerged as a metabolite of interest in caecal samples, showing a positive association with EAE (Fig. 3c). Importantly, GABA concentration was significantly elevated in non-EAE-induced mice harbouring *A. muciniphila*-containing

synthetic microbiota (SM) combinations (Fig. 3d). These results suggest that increased caecal GABA levels, even before disease induction, were primarily a result of *A. muciniphila* presence and not a consequence of EAE induction. Thus, assessment of caecal GABA concentrations might allow for prediction of EAE severity. However, not all SM01-colonized mice developed severe disease upon EAE induction (Extended Data Fig. 4b–d), despite consistent GABA concentrations in non-EAE-induced (Fig. 3d (left)) and EAE-induced (Fig. 3d (right)) SM01-colonized mice, raising the need for individualized disease predictors.

To understand how the inclusion of *A. muciniphila* impacts the transcriptional activity of all members of the microbial community, we compared caecal microbial metatranscriptomic profiles of SM14 to SM13 groups (Fig. 3e) and identified 117 genes transcribed only in SM14-colonized mice (Fig. 3f). Although we expected that these transcripts would be mostly from *A. muciniphila*, in fact, most of these genes were exclusively transcribed by either *Roseburia intestinalis* or *Marvinbryantia formatexigens* (Fig. 3g). Of the 30 genes transcribed only in SM13-colonized mice, the majority were expressed by *Eubacterium rectale* (Fig. 3g). These findings highlight the crucial impact of the presence of a single commensal on the gene expression

pattern of other microbial community members, probably impacting their 'function' (Fig. 1a) within a given community. These indirect influences on community function might also contribute to microbiota-mediated effects on EAE development and thus impact disease-mediating properties of potential risk-predicting species.

**Microbial mucin degrading capacity unrelated to EAE severity**

We observed that four strains were significantly higher in abundance in SM13 mice compared with SM14 mice (Fig. 4a). To address the potential contribution of these four strains to EAE development, we colonized mice with three additional SM combinations (Fig. 4b and Extended Data Fig. 6a). *Faecalibacterium prausnitzii*—a species known for gut health-promoting properties[26] and reduced in MS patients[10]—was significantly increased in SM13 mice (Fig. 4a (far right)). Thus, to examine whether the reduced EAE severity in SM13 mice is due to the protective effect of *F. prausnitzii*, we colonized GF mice with an SM12 community (Fig. 4b) lacking *A. muciniphila* and *F. prausnitzii*. Intriguingly, SM12-colonized mice (Fig. 4c–e) provided a comparable disease course as SM13-colonized mice (Fig. 2), suggesting that *F. prausnitzii* expansion in SM13-colonized mice is not responsible for decreased EAE in SM13-colonized mice. At the same time, these data point out the *A. muciniphila*-mediated inhibitory effects on the expansion of an anti-inflammatory bacterium, *F. prausnitzii*.

Removal of *A. muciniphila* from the SM14 community further resulted in expansion of three mucin glycan-degrading[18] Bacteroidota species (Fig. 4a) in the SM13 community. Thus, we next investigated whether colonization with the three mucin glycan-degrading strains alone (SM03) resulted in decreased EAE compared with SM14-colonized mice and whether addition of *A. muciniphila* (SM04) might counteract a potential beneficial effect. While SM03- and SM04-colonized mice showed comparable EAE disease courses (Fig. 4c–e) to SM14-colonized mice (Fig. 2), they differed significantly from SM13-colonized mice. In addition, the three mucin glycan-degrading Bacteroidota species appeared to not provide disease-reducing properties but instead disease-promoting properties in the absence of the remaining 10 strains within the SM13 community. To evaluate whether dysregulated mucin turnover might contribute to the observed results in these mice, we assessed various indirect measures for intestinal barrier integrity. We did not detect any correlations between EAE outcome and glycan-degrading enzymatic activities (Extended Data Fig. 6b–d), serum concentrations of lipopolysaccharides (LPS), occludin or zonulin, as well as faecal concentrations of lipocalin (Extended Data Fig. 6e,f), or short-chain fatty acids (Extended Data Fig. 6g). Altogether, our results suggest that bacterially mediated mucus glycan degradation or barrier integrity impairment, in the context of the microbiota combinations used in this study, was not an individual predictor of EAE disease development.

**Microbiota composition estimates probability of severe EAE**

Thus far, groupwise comparisons of EAE-associated readouts and microbiota compositions failed to identify reliable predictors for disease development in EAE-induced mice. Therefore, we next aimed to elucidate common denominators on a group-based and individual level to help uncover more reliable potential predictors for microbiota-mediated impacts on disease course. First, we conducted group-based comparison of EAE outcomes between all 10 tested diet–colonization combinations ('groups') (Fig. 5a,b). Performing hierarchical clustering (Fig. 5c) based on group means of key EAE-associated readouts (Fig. 5b) revealed three distinct group phenotypes: 'moderate', 'intermediate' and 'severe'. Our flow cytometry analyses (Extended Data Fig. 7a,b) revealed that these group phenotypes were reflected by a characteristic T-cell polarization pattern in mesenteric lymph nodes (MLN) and the colonic lamina propria (CLP) even before EAE induction (Extended Data Fig. 7c). While diet explained a maximum of 7.3% of the variance observed for EAE-associated readouts, microbiota composition (SM) explained between 11.2% and 27.2% (Fig. 5d). Given these low values, which are rooted in considerable intragroup variances (Fig. 5b), we performed individual EAE phenotype clustering, treating all mice across all groups individually (Fig. 5e). T-distributed stochastic neighbour embedding (t-SNE) analysis of all EAE-induced individuals resulted in two disease clusters: 'Cluster 1', comprising mice showing strong EAE symptoms, and 'Cluster 2', comprising mice showing mild EAE symptoms (Fig. 5e). While proportions of most analysed T-cell subsets in MLN and CLP of EAE-induced mice were similar in mice of both clusters (Extended Data Fig. 7d), we found significant infiltration of IL-17- and IFNγ-expressing Th cells in the spinal cords (SC) in Cluster 1 mice (Extended Data Fig. 7e). Except for SM03 and SM04 mice, all other groups contained mice of both phenotypes with varying proportions (Fig. 5f). These proportions broadly, but not completely, corresponded to the group-based phenotype classification (Fig. 5c). In summary, these results (Fig. 5a–f) indicate that knowing the microbiota composition, in combination with information on dietary conditions, enables estimation of the probability for either moderate or severe disease, but is unsuitable to predict individual EAE outcomes.

**B. ovatus IgA coating predicts EAE severity in SM setting**

We next aimed to identify the microbiota-associated factors suitable for predicting individual outcomes of EAE. For each given strain (Fig. 5g), we first assessed whether its relative abundance before EAE induction (Extended Data Fig. 8a) allowed for prediction of individual disease course after EAE induction (Fig. 5h,i). To do so, we only analysed mice harbouring at least 12 different strains (SM12, SM13, SM14). Correlations for each strain were only assessed for mice that were gavaged with the respective strain and calculations were performed by either including

---

**Fig. 3 | *A. muciniphila*-mediated neuroinflammation is associated with increased caecal concentrations of γ-amino butyric acid. a–d**, GF C57BL/6N mice were colonized with *A. muciniphila* only (SM01), SM13, SM14 or remained GF. Caecal contents were collected 25 d after colonization (−EAE) or 30 d after EAE induction (+EAE) and subjected to CE–TOF/MS-based metabolomics analysis. **a**, Principal component analysis (PCA) of log$_2$-normalized metabolite concentrations, faceted by colonization. **b**, Ward hierarchical clustering based on scaled group means of log$_2$-normalized metabolite concentrations. **c**, Statistically significant positive (PCor) or negative (NCor) Spearman correlations by linear regression across all samples (both −EAE and +EAE mice) and groupwise comparison criteria. Correlations referring to EAE-associated readouts (abbreviations as in Fig. 1) were calculated from +EAE mice only. Groupwise comparisons include significantly different metabolites based on unpaired *t*-tests of log$_2$-normalized concentrations with *P*-value adjustment using the Benjamini–Hochberg method. Barplots indicate the total number of metabolites that fulfil each criterion. Of 175 measured metabolites, only the 14 metabolites that demonstrate a significant correlation with AUC are displayed. Grey squares indicate that a given metabolite fulfilled a specific criterion,

while white squares indicate failure to fulfil a given criterion. **d**, Boxplots showing median, quartiles and 1.5 × IQR of log$_2$-concentrations of GABA −EAE or +EAE conditions. One-way ANOVA followed by Tukey's post-hoc test. **e**, Multidimensional reduction of caecal metatranscriptome profiles of −EAE SM14- and SM13-colonized mice. In SM14-colonized mice, transcripts attributed to *A. muciniphila* were removed and counts were renormalized to allow for fair comparison with metatranscriptome profiles of SM13-colonized mice. Two dots from SM13 −EAE overlap in the plot. **f**, Volcano plot showing log$_2$(fold change, FC) of gene product-annotated transcript abundances in SM14- vs SM13-colonized mice (*x* axis) and −log$_{10}$(*P* value) (*y* axis), calculated using an exact test in edgeR. Dashed line represents significance threshold. Yellow and blue dots represent transcripts found only in SM14- or SM13-colonized mice, respectively, while grey dots represent transcripts found in both groups. **g**, Two left columns: transcripts found only in SM14-colonized or SM13-colonized mice. Two right columns: transcripts up- or downregulated in SM14- vs SM13-colonized mice but present in both groups. Mouse numbers are indicated on the respective panels and treated as biological replicates. All statistical tests were two-sided.

mice harbouring SM12, SM13 and SM14 communities, or combinations thereof, into the analysis. We found statistically significant correlations between pre-EAE bacterial relative abundances with EAE-associated readouts for some strains (Fig. 5h). However, the few statistically significant correlations that we determined were generally weak ($R < 0.4$) and Pearson correlation values for a given strain were highly dependent on the background microbiota (Fig. 5h). Likewise, the relative abundances of strains only explained very low proportions of the variances across all groups for all assessed EAE-associated readouts (Fig. 5i).

Next, to determine whether the presence or absence of a given strain might be a better predictor of individual EAE development, we performed a linear mixed model regression for three EAE-associated readouts, with presence of the strain as an independent variable and colonization as a random intercept effect (Fig. 5j and Extended Data Fig. 8b). Given the setup of our tested SM combinations, we could only assess *A. muciniphila* and *F. prausnitzii* separately and had to analyse the remaining 12 strains in groups of two combinations. Presence or absence of a specific strain or strain combination was insufficient to predict the individual outcome of any of the tested EAE-associated readouts (Fig. 5j and Extended Data Fig. 8b), indicating that the potential disease-driving or -preventing properties of a given strain or strain combination is determined by the background microbiota.

Coating of intestinal commensals by host plasma cell-derived IgA represents a crucial host response for maintaining immune

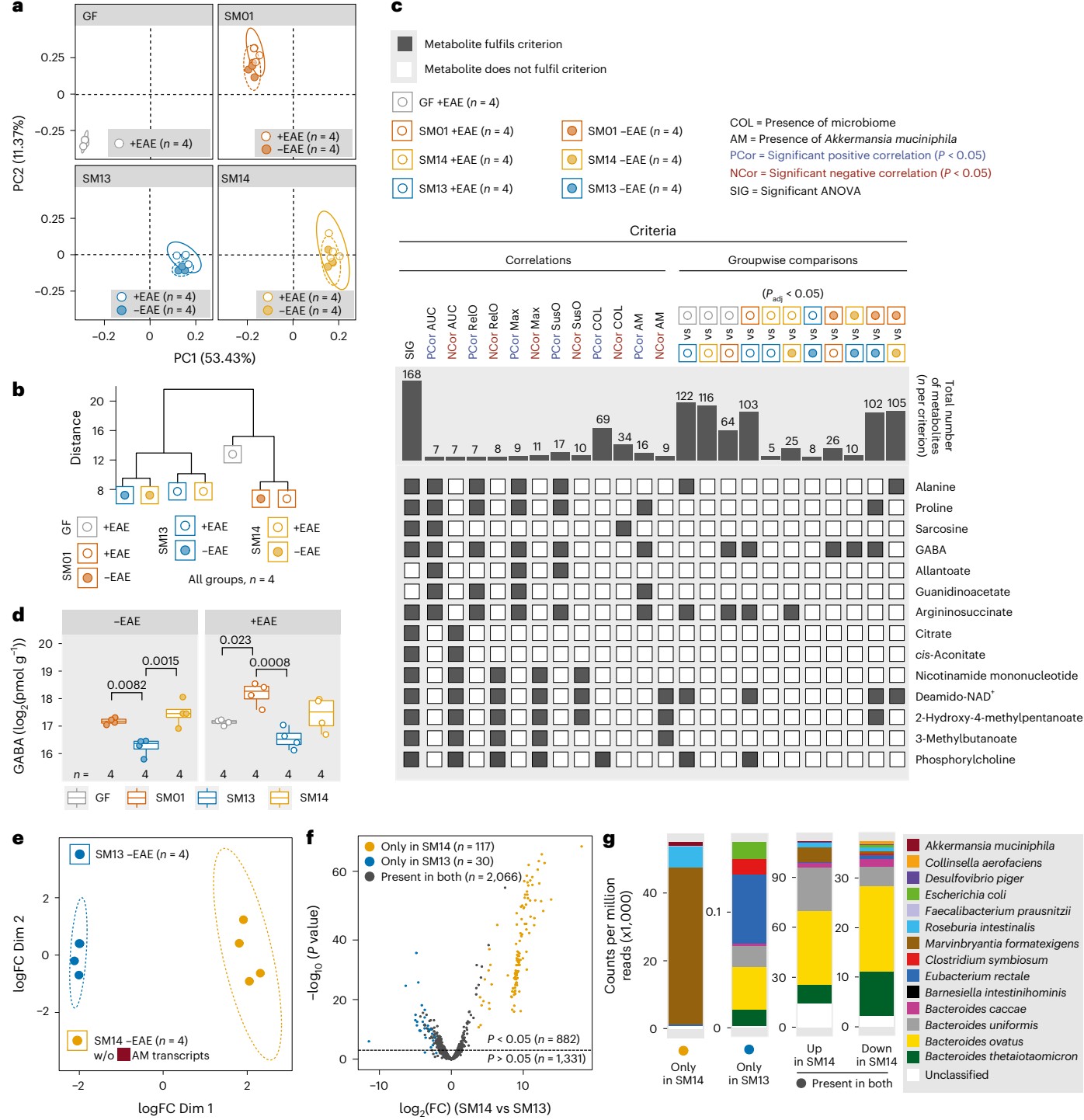

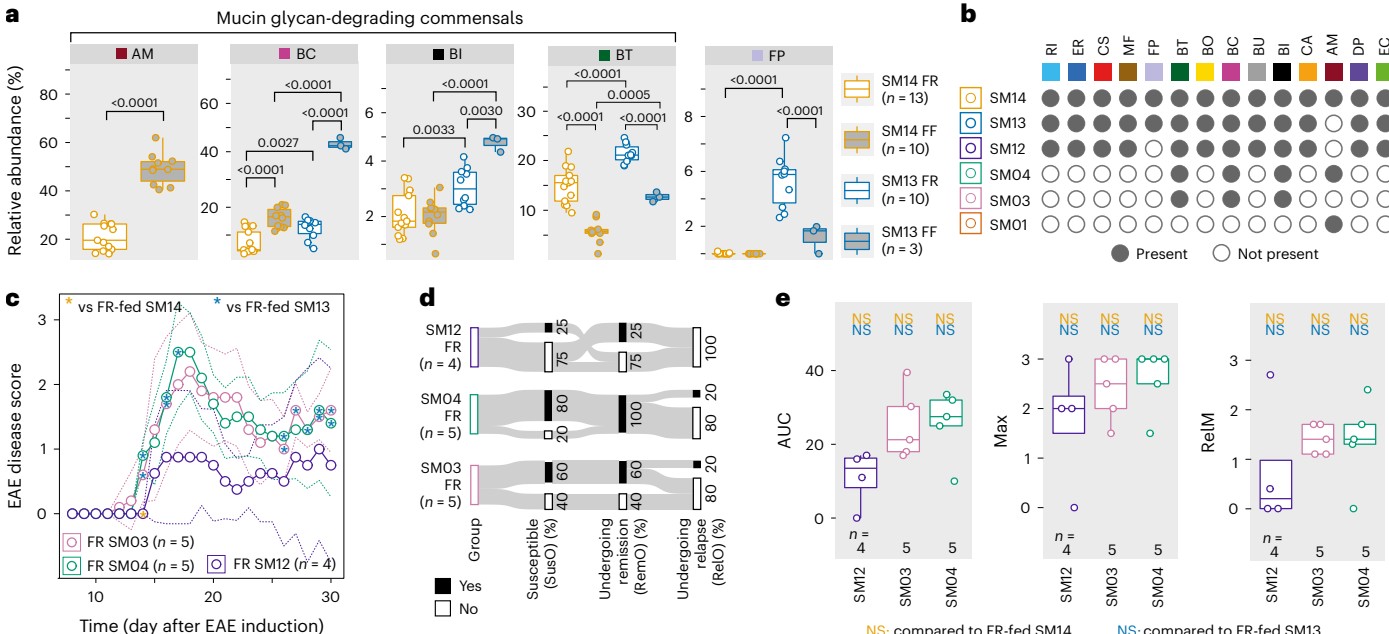

**Fig. 4 | Microbial mucin degradation is disconnected from EAE disease course. a**, Relative abundances of strains that provided statistically significant differences between FR-fed SM14-colonized mice and FR-fed SM13-colonized mice on the day of EAE induction, as determined using one-way ANOVA followed by multiple comparisons with $P$-value adjustment using the Benjamini–Hochberg method. Only biologically meaningful multiple comparisons made (same SM, different diet; different SM, same diet). **b**, Constituent strains of the SM communities. Strain abbreviations as in Fig. 2a. **c**, EAE disease scores as a function of time. Daily EAE scores were compared using Wilcoxon rank-sum tests with $P$-value adjustment using the Benjamini–Hochberg method. Blue asterisks represent comparison with FR-fed SM13-colonized mice, while yellow asterisks represent comparison with FR-fed SM14-colonized mice. *$P$ < 0.05. **d**, Sankey diagram of key event occurrence (in % of all mice within one group)

during EAE. **e**, Left: AUC analysis of the disease course depicted in **c**. Each mouse is depicted by a separate dot. Middle: maximum EAE score per mouse (Max). Right: mean EAE score during relapse phase (RelM, day 26 to day 30 after EAE induction). Analysed using one-way ANOVA followed by Tukey's post-hoc test for groupwise comparisons (AUC and RelM) or Wilcoxon rank-sum tests with $P$-value adjustment using the Benjamini–Hochberg method (Max). Blue text indicates comparison with FR-fed SM13-colonized mice, while yellow text indicates comparison with FR-fed SM14-colonized mice. NS, not significant. Mouse numbers are indicated on the respective panels and treated as biological replicates. FR-fed SM14 and SM13 data are from three independent experiments; FF-fed SM14 data are from two independent experiments; all other groups are from one experiment. Boxplots (**a**,**e**) show median, quartiles and 1.5× IQR. All statistical tests were two-sided.

homoeostasis[27,28], also in the context of autoimmune neuroinflammation[29,30]. Secretory IgA (sIgA) levels in the mouse faeces were disconnected from individual EAE outcomes (Extended Data Fig. 8c) but were reflective of the microbiota composition (Extended Data Fig. 8d). Interestingly, we found a significant correlation between group means of sIgA concentrations and corresponding EAE susceptibility prevalence (Extended Data Fig. 8e). Owing to these observations and given that the changes in 'IgA coating index' (ICI) of a given commensal species could be linked to inflammation[31], we determined ICIs for each strain within each higher-diversity SM combination (SM12, SM13 and SM14) in every individual (Fig. 5k–m and Extended Data Fig. 8f–h). Interestingly, there was a tendency for ICIs to differ between distinct SM combinations (Extended Data Fig. 8f), and microbiota composition explained 40.5% and 55.0% of the variance between ICIs of *B. ovatus* and *B. uniformis*, respectively (Fig. 5k and Extended Data Fig. 8g), suggesting a crucial role of the background microbiota on strain-specific IgA coating (Extended Data Fig. 8h). In addition, ICIs of these strains varied not only between distinct groups but also between individuals within groups. Thus, we reasoned that the individual ICI of these strains might reflect individual EAE-promoting properties of the microbiota in a certain host. Correlation analysis of strain-specific ICI, as determined from faecal samples obtained before EAE induction, with EAE outcome in the same individual, revealed significant correlations with some EAE-associated readouts for two strains (Fig. 5l). However, the only strain whose individual ICI provided significant correlations with the two most important EAE-associated readouts (AUC and maximum achieved EAE score) was *B. ovatus* (Fig. 5l,m), thus

allowing for individual prediction of EAE disease course across all *B. ovatus*-encompassing SM combinations.

## IgA coating predicts EAE development in complex communities

Next, we checked whether the ability to predict individual EAE disease course from the ICI of a particular species is also possible in mice harbouring a complex microbiota. As we started our quest to find a reliable microbiota-associated EAE predictor in SPF-housed mice of different genotypes (Fig. 1), we decided to evaluate disease prediction quality of species-specific ICIs in a similar setting. Importantly, we included *Muc2*[−/−] mice in this evaluation to address host-related genetic factors, which were neglected in the gnotobiotic experiments. To increase the number of distinct complex microbiota compositions, as defined by species presence rather than species relative abundance, we decided to perform a cross-genotype faecal microbiota transfer given the impact of the host on the microbiota engraftment[32,33]. To do so, we housed GF *Muc2*[−/−] and GF C57BL/6 mice as microbiota recipients in the spent litter of either SPF-housed *Muc2*[−/−] or SPF-housed *Muc2*[+/+] mice as microbiota donors (Fig. 6a). As our goal was to identify species-specific ICIs, we performed full-length 16S ribosomal RNA (rRNA) gene amplicon analysis after a 21-day colonization period to obtain reliable taxonomic information on a species level[34]. As expected, our microbiota transfer approach resulted in distinct microbiota compositions across all microbiota recipients (Fig. 6b,c and Extended Data Fig. 9a,b) with considerable inter-individual variations for most donor–recipient combinations (Fig. 6c). The genotype of the microbiota donor

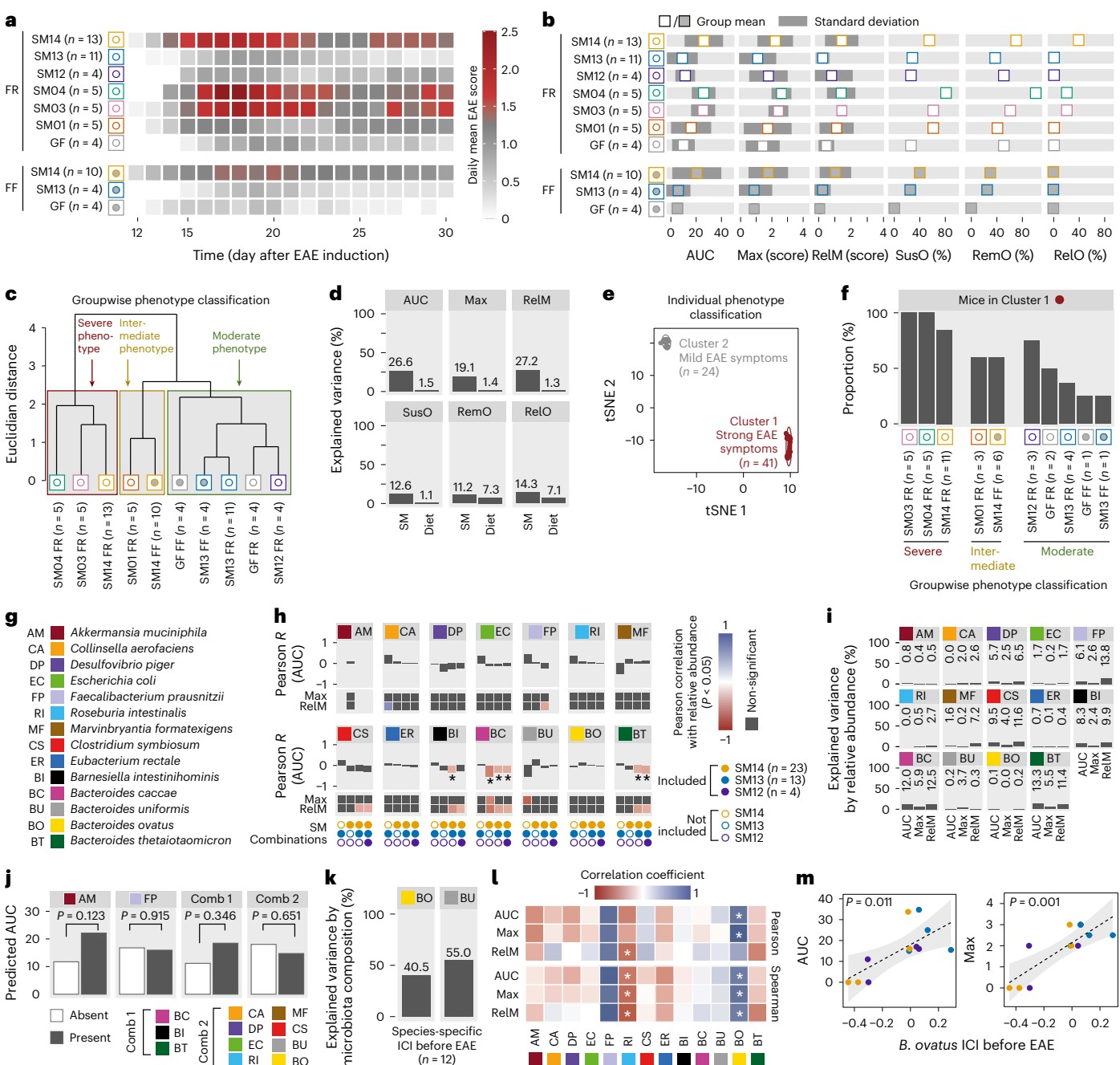

**Fig. 5 | Groupwise and individual prediction of EAE based on microbiota characteristics. a**, Heat map of EAE disease course of all tested colonization–diet combinations. **b**, Horizontal barplots summarizing EAE-associated readouts of all tested colonization–diet combinations. **c**, Cluster dendrogram of scaled group means of EAE-associated readouts based on a Euclidian distance matrix. Group phenotypes (moderate, intermediate and severe) classified according to the three clusters. **d**, Variance explained by diet or SM when comparing EAE-associated readouts among all colonization–diet combinations ($n = 65$) using $\eta^2$ calculation. **e**, Individual-level EAE phenotype classification by t-SNE analysis (perplexity of 20 with 6 initial dimensions) of EAE-associated readouts across all tested colonization–diet combinations. **f**, Proportion of mice in Cluster 1 (strong EAE symptoms) per SM–diet combination, with groupwise EAE phenotype classification indicated at the bottom. **g**, Colour codes and abbreviations of SM14 constituents. **h**, Pearson correlation between strain relative abundance before EAE induction and EAE-associated readouts for mice fed both diets. AUC–strain correlations as barplots; Max–strain and RelM–strain correlations as colour-coded squares. Significant correlations ($P < 0.05$) by linear regression

in colour; non-significant correlations in grey. Correlations calculated for four different SM combinations: SM13 only; SM14 only; SM13 and SM14; and SM12, SM13 and SM14. **i**, Variance of EAE-associated readouts explained by strain relative abundance before EAE induction, performed by combining SM12-, SM13- and SM14-colonized mice irrespective of diet ($n = 40$). **j**, Linear mixed model regression for predicted AUC, with strain presence as an independent variable and SM as a random intercept effect ($n = 40$). **k**, Variance in ICI explained by high- (SM12, SM13, SM14) versus low- (SM03, SM04) diversity background microbiota compositions in two strains providing the highest explained variance using $\eta^2$ calculation. **l**, Individual-based Pearson (top) and Spearman (bottom) correlations of key EAE-associated readouts with ICIs of SM14-constituent strains in SM12-, SM13- and SM14-colonized mice ($n = 12$) before EAE induction. *$P < 0.05$ by linear regression. MF strain absent due to lack of data. **m**, Correlation of *B. ovatus* ICI with AUC (left) and maximum EAE score (Max, right) by linear regression in all mice harbouring *B. ovatus*, irrespective of the background microbiota. Dashed line represents linear regression, with the confidence interval shaded in grey.

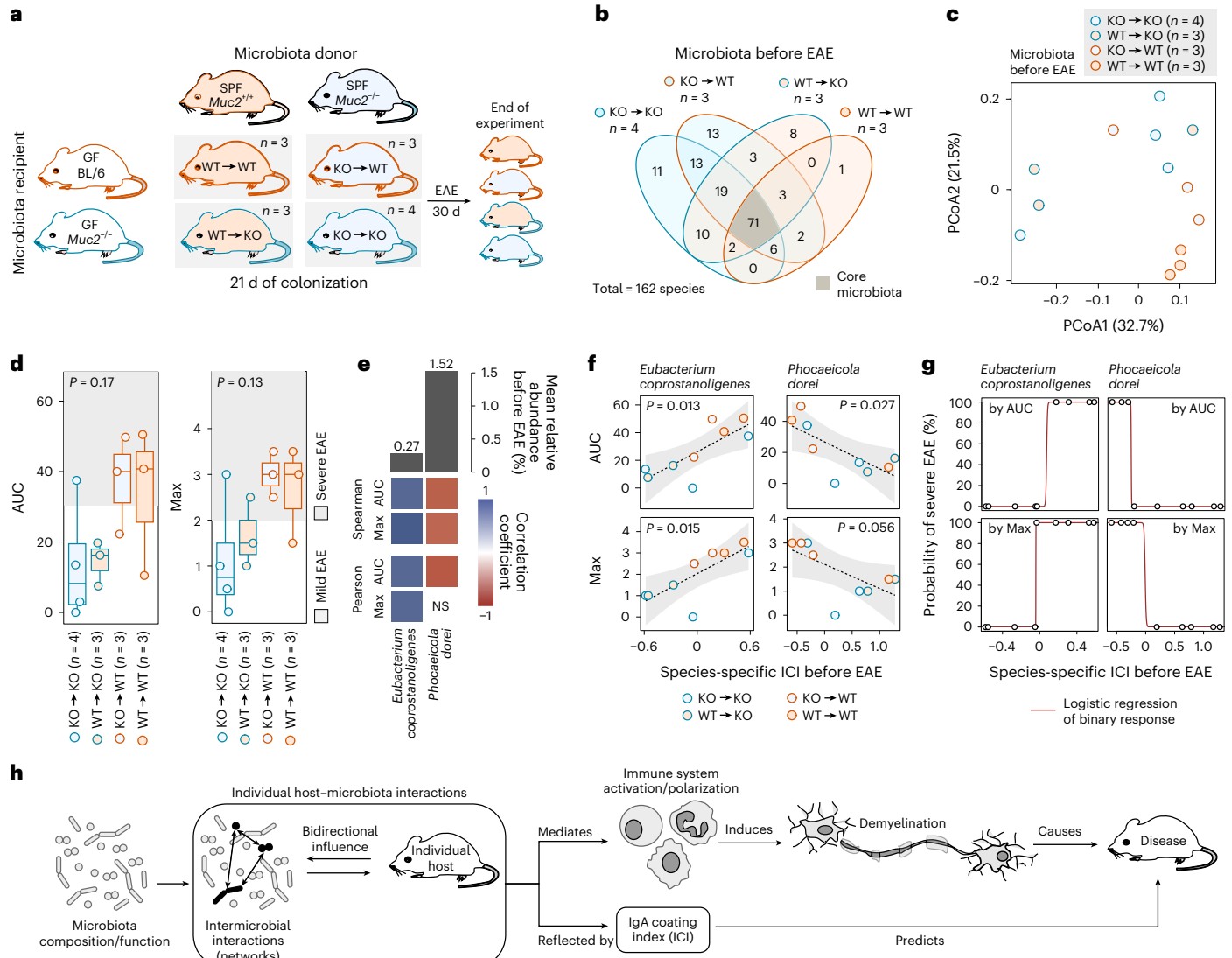

**Fig. 6 | IgA coating index of reporter species to predict neuroinflammation-promoting properties of complex microbiotas. a**, Microbiota from SPF-housed *Muc2⁻/⁻* (KO) and *Muc2⁺/⁺* (WT) mice was transferred to GF C57BL/6J (BL/6) and *Muc2⁻/⁻* mice. After 21 d of colonization with the donor microbiota, EAE was induced and disease course monitored for 30 d. **b**, Venn diagram of microbiota compositions by full-length 16S rRNA gene sequencing across the four donor–recipient combinations. Numbers reflect shared species for the specified donor–recipient combination, with core microbiota, shared by all, highlighted in grey. **c**, β-diversity of pre-EAE microbiota compositions as determined from a Bray–Curtis distance matrix of arcsine square root-transformed relative abundance data, ordinated using principal coordinate analysis. **d**, AUC (left) and maximum EAE disease score (Max, right) of individual disease courses depicted in Extended Data Fig. 9d. AUC, one-way ANOVA with donor–recipient combinations as tested variable (AUC) or Kruskall–Wallis test with donor–recipient combinations as tested variable (Max). Individual EAE phenotype classification ('mild' vs 'severe') based on Extended Data Fig. 9f. Boxplots show median, quartiles and 1.5× IQR.

**e**, Individual-based Pearson and Spearman correlations of EAE-associated readouts (AUC, Max) with ICIs of *Eubacterium coprostanoligenes* and *Phocaeicola dorei* before EAE induction. Mice of all donor–recipient combinations were included in this analysis. Significant correlations are depicted in shades of blue or red. Grey bars represent mean relative abundance across all mice before EAE induction. **f**, Correlation of individual ICIs for *E. coprostanoligenes* and *P. dorei* with individual values for AUC (top) and Max (bottom) by linear regression. Only mice where *E. coprostanoligenes* or *P. dorei* were detectable in both fractions after sorting are shown. Dashed line represents linear regression, with the confidence interval shaded in grey. **g**, Binomial regression model to predict probability of severe disease based on classification by either AUC (top panels) or Max (bottom panels) using ICI of *E. coprostanoligenes* or *P. dorei*. For definition of disease severity, see Extended Data Fig. 9f. **h**, Graphical summary of results. Microbiota composition alone fails to predict individual EAE susceptibility or development. However, individual host–microbiota interactions, which are reflected by the ICI of certain reporter species, were suitable for predicting individual EAE disease courses.

non-significantly impacted final microbiota composition (Extended Data Fig. 9c (top)). However, the final microbiota composition was significantly impacted by the genotype of the microbiota recipient (Extended Data Fig. 9c (bottom)).

Following microbiome characterization before EAE, we induced EAE in all mice and monitored disease progression for 30 d (Fig. 6a). The emerging individual EAE disease course (Extended Data Fig. 9d) was independent of the genotype of microbiota donor (Extended Data Fig. 9e) and the donor–recipient combination (Fig. 6d), but dependent on the genotype of the recipient mouse (Extended Data Fig. 9e). Categorizing each mouse into a binary EAE phenotype (Extended Data Fig. 9f) revealed that the EAE phenotype was disconnected from the taxonomic β-diversity of the microbiota before EAE induction (Extended Data Fig. 9g), supporting our observation that assessing the overall microbiota composition alone is an inadequate measure for predicting EAE development.

Next, we separated IgA-coated from non-IgA-coated bacteria (Extended Data Fig. 10a) in faecal samples collected on the day of EAE induction and performed full-length 16S rRNA gene sequencing on each fraction (Extended Data Fig. 10b,c), followed by determining species-specific ICIs for each mouse. Correlation analyses of species-specific ICIs before EAE induction with individual EAE-associated readouts revealed the ICIs of *Eubacterium coprostanoligenes*, *Phocaeicola dorei* (Fig. 6e–g) and *Enterocloster bolteae* (Extended Data Fig. 10d,e) to be reliable predictors of individual EAE development, irrespective of background microbiota composition and recipient genotype. We could not assess the prediction quality of *B. ovatus* ICI in these mice, as *B. ovatus* prevalence was too low (Extended Data Fig. 10f). The only SM14 constituent that was sufficiently prevalent for correlation calculations was *A. muciniphila* and, in line with findings using mice harbouring a reduced microbiota (Fig. 5l), *A. muciniphila* ICI was disconnected from subsequent EAE development (Extended Data Fig. 10g). Nevertheless, our data in complex communities corroborate that ICI of some microbial species could be reliable predictors of disease and such predictor species might differ in mice and humans. On the basis of our results in both reduced and complex communities, we therefore conclude that assessing microbiome characteristics before disease induction can indeed predict EAE disease development. However, microbial networks within a given microbiota and the bidirectional impact of host–microbiota interactions must be taken into account (Fig. 6h).

## Discussion

In this study, we evaluated whether commonly employed cross-sectional analysis approaches are appropriate to uncover potential disease-related microbial risk factors. As human cohorts naturally involve volunteers with already established disease, it is impossible to decipher whether changed relative abundances of risk factor candidates precede disease pathogenesis or whether they are a consequence of the disease. However, using an established pre-clinical mouse model for MS, EAE, we were able to shortlist microbial risk factors on the basis of pre-disease microbiota compositions and link them to disease outcome. Using mice of different genetic backgrounds and distinct complex microbiota compositions, we found the genus *Akkermansia* to be the most negatively associated with EAE disease development.

Next, we examined whether this finding could be reproduced in gnotobiotic, genetically homogenous mice harbouring different combinations of a reduced reference microbiota, with or without *A. muciniphila*. Intriguingly, we found *A. muciniphila* to be positively associated with EAE severity in certain mice harbouring specific reduced communities. These contradictory results on *A. muciniphila* are corroborated by findings from other groups. *A. muciniphila* was reportedly increased in MS patients across various human cohorts[2,5,7,10,35]. Other studies, however, report on positive effects of *A. muciniphila* on maintaining general gut homoeostasis[36,37] or on progression of autoimmune neuroinflammation in mice[38,39] or humans[40]. Given the genotypic and phenotypic diversity of *A. muciniphila*[41], these reported discrepancies might be rooted in strain-specific effects on the host. However, our study using the same *A. muciniphila* strain in all gnotobiotic experiments implies that mutual influences between a suspected risk species and the microbial environment crucially shape the overall microbiota's disease-impacting potential. This potential was unrelated to the presence or absence of a particular taxon, suggesting that focusing on certain combinations of taxa, rather than single taxa alone, is essential to derive more reliable conclusions from microbiota profiles.

Our metatranscriptome analyses suggest that even minor changes in microbiota composition, that is, by removing *A. muciniphila* from a reduced community, resulted in profound changes in gene expression patterns of some, but not all, intestinal microbes. Therefore, disentangling specific intermicrobial networks might help to predict disease-mediating properties of a given overall microbiota.

Although not yet a technically and analytically refined approach[42], metagenome-based network analyses are currently being explored as an analysis tool[43]. A key study evaluating the effects of multiple defined microbiota compositions on fitness of *Drosophila melanogaster* already pointed out that microbial network interactions are more important than relative abundances of a given species alone[44]. Our study documents similar innovative findings based on comprehensive datasets in a controlled vertebrate gnotobiotic disease model.

We next sought to investigate whether concentrations of certain bacterial metabolites might predict disease development, as these metabolites could be a result of intermicrobial networks. Importantly, our analysis approach demonstrated that focusing only on correlations between metabolite concentrations and disease phenotypes resulted in shortlisting false-positive potential predictors. Applying a more stringent, context-focused analysis pipeline revealed that only elevated caecal concentrations of GABA were associated with increased EAE in two *A. muciniphila*-encompassing microbiota compositions. However, this was tested only in four reduced microbiota compositions, and pre-EAE GABA concentrations only allowed assessment of risk for severe EAE but did not predict individual disease development. Contrary to our findings, faecal GABA concentrations in MS patients were decreased compared with healthy controls[45]. Although the ability of *A. muciniphila*[46] and other gut commensals[47,48] to produce GABA has previously been reported, the effect on other microbes and the resulting neuroinflammation-promoting or -preventing properties remain elusive. Thus, the potential predictive qualities of GABA or other metabolites must be seen in the context of a given microbiota, which reduces their practicability as universal or even individual disease predictors.

In addition to these microbiota-specific effects, host-specific effects appeared to be a decisive factor for individual EAE development in our experiments. Effects of host genetics on microbiota composition are well described[32,33]; however, even in genetically homogeneous mice of the same sex and age, harbouring the exact same set of commensal bacteria and living under the same standardized conditions, we found considerable individual differences in EAE disease courses (Fig. 5b). This suggests that disease progression derives from a bidirectional interaction between a given host and its individual microbial networks[49,50] (Extended Data Fig. 1 and Fig. 6h). After extensive evaluation of multiple microbiota-associated readouts, we found that the ICI of certain commensals reflects these disease-driving host–microbiota interactions. Thus, the ICIs of these species before disease onset allowed us to correctly predict the individual EAE progression across all microbiota–host combinations. However, due to minimal core microbiota overlap between reduced and complex communities, we could not single out the ICI of one particular species as a universal predictor: while we identified *B. ovatus* ICI in reduced communities, we determined ICIs of *E. coprostanoligenes* and *P. dorei* to be reliable predictors in complex communities in a mouse model. We propose that these species act as 'reporter species', reflecting the individual bidirectional host–microbiota influences on EAE progression. A limitation of our study is that we have not mechanistically verified the associations of these reporter species or metabolites such as GABA with disease severity.

In summary, we demonstrate that making disease-course predictions based on microbiota characteristics is generally possible but is not as straightforward as surveying community member presence or abundance. While we were not able to evaluate our model in early stages of disease, such a validation would boost the clinical relevance of using ICI to identify reporter species. Nonetheless, it is possible that the signal reflecting these bidirectional host–microbiota influences is masked by broader immune changes when symptoms manifest. Thus, the identification of such microbiota-associated predictors would require long-term longitudinal sampling of yet-undiagnosed individuals who eventually develop disease. Although such a design

is challenging, the currently ongoing Finnish Health and Early Life Microbiota (HELMi) longitudinal birth cohort[51] represents one such study implementing this approach. Ultimately, the ability to predict disease course in a given individual represents a valuable factor to make better-informed clinical decisions regarding that patient's treatment plan. Therefore, in line with the findings of our study, we recommend a reconsideration of microbiota-related data analysis approaches and the design of prospective cohort studies aimed at identifying microbiota-based predictors of disease course.

## Methods

### Ethics statement for mouse experiments

All mouse experiments followed a two-step animal protocol approval procedure. Protocols were first evaluated and pre-approved by either the Animal Experimentation Ethical Committee of the University of Luxembourg (AEEC) for experiments with germ-free and gnotobiotic animals, or the Animal Welfare System (AWS) of the Luxembourg Institute of Health for experiments with SPF mice, followed by final approval by the Luxembourgish Ministry of Agriculture, Viticulture and Rural Development (Protocol numbers: LUPA2020/02, LUPA2020/27, LUPA2020/32, LUPA2019/43, LUPA2020/2 and LUPA2019/51). All experiments were performed according to the Federation of European Laboratory Animal Science Association (FELASA). The study was conducted according to the 'Règlement grand-ducal du 11 Janvier 2013 relatif à la protection des Animaux utilisés à des fins Scientifiques' based on the 'Directive 2010/63/EU' of the European Parliament and the European Council from September 2010 on the protection of animals used for scientific purposes. Based on EAE disease course (AUC) of SPF-housed wildtype and *Muc2*$^{-/-}$ mice (Fig. 1f (left)) and considering an α error of 0.0033 (after Bonferroni correction for up to 15 groupwise comparisons) and a power of 80%, a minimum of four mice per group was determined necessary to ensure that our study was sufficiently statistically powered.

### Origin of mice and housing conditions

All animals were exposed to 12 h of light daily, with water and diets provided ad libitum. For gnotobiotic experiments, female GF C57BL/6N mice purchased from Taconic Biosciences were bred and housed in the GF facility of the University of Luxembourg. Mice were randomly allocated to different experimental groups and were maintained in ISO cages with a maximum of five animals per cage. Before the start of experiments, GF status of all mice was confirmed by ensuring that no bacterial growth was observed following the anaerobic and aerobic incubation of faecal samples in 5 ml culture tubes containing two different non-selective media: brain heart infusion broth and nutrient broth.

For experiments performed under SPF conditions, as shown in Fig. 1, we used female C57BL/6J wildtype mice purchased from Charles River Laboratories (France) at the age of 5–8 weeks and housed in the SPF facility of the Luxembourg Institute of Health. Furthermore, we used mice lacking the *Muc2* gene (strain designation: 129P2/OlaHsd×C57BL/6-Muc2<tm1Avel>), which were originally obtained from the University of Bern, Switzerland under GF conditions. GF 129P2/OlaHsd×C57BL/6-Muc2<tm1Avel> mice were mated with SPF-housed C57BL/6J mice, resulting in offspring heterozygous for the presence of the *Muc2* gene (*Muc2*$^{+/-}$). *Muc2*$^{+/-}$ mice were kept under the same SPF conditions as the SPF-housed parental C57BL/6J mice. Next, male and female *Muc2*$^{+/-}$ mice were mated and offspring were genotyped for the absence or presence of the *Muc2* gene. Homozygous *Muc2*$^{-/-}$ and *Muc2*$^{+/+}$ mice obtained from this breeding were then used for experiments.

### Genotyping for the presence or absence of the *Muc2* gene

Genotyping for the presence or absence of the *Muc2* gene from mouse ear tissue was performed using the SampleIN Direct PCR kit (HighQu, DPS0105) according to manufacturer instructions. Three different primers were used at a final concentration of 0.4 µM in the PCR reaction (Primer 1: 5′-TCCACATTATCACCTTGAC-3′; Primer 2: 5′-GGATTGGGAAGACAATAG-3′; Primer 3: 5′-AGGGAATCGGTAGACATC-3′). The PCR was conducted with an annealing temperature of 56 °C and 40 cycles. Presence of the *Muc2* gene resulted in an amplicon of 280 bp, while its absence resulted in an amplicon of 320 bp. Amplicons were visualized on a 1.5% agarose gel using gel electrophoresis.

### Colonization of germ-free mice with a synthetic microbiota

All 14 bacterial strains of the human synthetic microbiota were cultured and processed under anaerobic conditions using a Type B vinyl anaerobic chamber from Coy Laboratories, as published in detail previously[20]. A total of six different SM combinations were used to colonize GF mice in a randomized manner by cage. Non-colonized GF mice were used as a control group. Intragastric gavage and verification of proper colonization of administered strains were performed as described in detail elsewhere[20]. Details on the 14 different strains used are summarized in Supplementary Table 1.

### Mouse diets

Mice were either maintained on standard mouse chow (fibre-rich) or switched to a fibre-free diet in a randomized manner by cage. We used two different FR diets with ~15% dietary fibre: SAFE A04 chow (SAFE Diets, U8233G10R) for gnotobiotic mice, sterilized by 25 kGy gamma irradiation, and SDS Standard CRM (P) Rat and Mouse Breeder and Grower diet (Special Diets Service, 801722), sterilized by 9 kGy gamma irradiation, for SPF-housed mice. The FF diet was used for both gnotobiotic and SPF-housed mice and was custom manufactured by SAFE Diets on the basis of a modified version of the Harlan TD.08810 diet, as described previously[18].

### Experimental timeline of mouse experiments

**Experiments performed under gnotobiotic conditions.** At the age of 5–8 weeks, mice were colonized with various SM combinations (see above) while fed an FR diet. At 5 d after initial colonization, mice were either maintained on an FR diet or switched to an FF diet until the end of experiment. Mice were then either induced with EAE (labelled as '+EAE' in the manuscript) 20 d after the initial gavage, or euthanized for organ collection 25 d after the initial gavage without induction of EAE (labelled as '−EAE' in the manuscript).

**Experiments performed under SPF conditions.** Mice of all genotypes were raised and maintained on an FR diet. At the age of 6 weeks, mice were either switched to an FF diet or kept on the FR diet. After 20 d, mice fed with either diet were subjected to induction of EAE. The course of EAE under both gnotobiotic and SPF conditions was observed for 30 d.

### Co-housing experiments

For microbiota transfer from SPF mice into GF mice, we used 3-day-old litter from SPF-housed *Muc2*$^{-/-}$ or *Muc2*$^{+/+}$ mice ('donor mice') and mixed it 1:1 (v/v) with fresh litter. This mix was equally distributed across multiple cages. GF *Muc2*$^{-/-}$ or C57BL/6J mice ('recipient mice') were then transported from a GF facility to an SPF facility. Recipient mice were housed in these litter mix-containing cages for 21 d under SPF conditions. To avoid cage-related microbiota transfer effects, cage rotation was performed as follows: all mice receiving the microbiota from mice of a given genotype were rotated between these cages so that each recipient mouse spent at least 5 d with each other recipient mouse, irrespective of the genotype of the recipient mouse.

### Experimental autoimmune encephalomyelitis

Mice were immunized using the Hooke kit MOG$_{35-55}$/CFA Emulsion PTX (Hooke Laboratories, EK-2110) according to manufacturer instructions. In brief, mice were immunized with a subcutaneous injection of a myelin oligodendrocyte glycoprotein-derived peptide (MOG$_{35-55}$) and

complete Freund's adjuvant (CFA) delivered in pre-filled syringes. Subcutaneous injection of two times 100 μl (200 μl in total) of MOG/CFA (1 mg ml$^{-1}$ MOG$_{35-55}$ and 2–5 mg ml$^{-1}$ killed *Mycobacterium tuberculosis* H37Ra per ml emulsion) mixture was performed on two sides bilateral in each of the mouse's flank. In addition, Pertussis toxin (PTX) solution was injected on the day of MOG peptide immunization and 48 h after the first injection. Glycerol-buffer stabilized PTX was diluted in sterile PBS for application of 400 ng PTX (gnotobiotic experiments) or 150 ng (experiments performed under SPF conditions) by intraperitoneal injection of 100 μl PTX solution. The EAE clinical symptom scores were assessed daily according to the scheme depicted in Extended Data Fig. 3. Due to the nature of the experiment (for example, visibly different diets, handling of mice from low to high SM to prevent contamination), complete blinding was impossible; however, the EAE scoring was performed by two independent researchers (A.S. and M.N.) to prevent bias. EAE-associated readouts were defined as follows: AUC, area-under-the-curve of individual EAE disease scores as a function of time; Max, maximum EAE score; RelM, mean EAE score during relapse phase (days 27–30); SusO, susceptibility occurrence (score of 2.5 for at least 1 d); RemO, remission occurrence (decrease of EAE score by 1.5 points compared with Max); RelO, relapse occurrence (increase by 1.0 point compared with remission score).

### Euthanasia and sample collection

Mice of all groups ('−EAE' and '+EAE') were subjected to terminal anaesthesia through intraperitoneal application of a combination of midazolam (5 mg kg$^{-1}$), ketamine (100 mg kg$^{-1}$) and xylazine (10 mg kg$^{-1}$), followed by cardiac perfusion with ice-cold PBS. Colonic content, caecal content, blood and organs were collected for downstream analysis. To isolate serum, whole blood was incubated for 30 min at 37 °C, followed by centrifugation for 30 min at 845 × *g* and r.t. Serum supernatant was then stored at −80 °C until use. MLNs were removed and homogenized by mechanical passage through a 70 μm cell strainer. MLN cells were washed once in ice-cold PBS for 10 min at 800 × *g*, resuspended in ice-cold PBS and stored on ice until further use. Colons and ilea were removed and temporarily stored in Hank's balanced salt solution without Ca$^{2+}$ and Mg$^{2+}$ (HBSS (w/o)) buffered with 10 mM HEPES on ice, while removed spinal cords were temporarily stored in Dulbecco's phosphate-buffered saline with calcium (D-PBS). All three organs were then subjected to lymphocyte extraction as described below.

### Lymphocyte extraction

After organ removal, lymphocytes from the CLP, small intestine lamina propria (SILP) and SC were extracted. CLP and SILP lymphocytes were extracted using the lamina propria dissociation kit (Miltenyi Biotec, 130-097-410), while SC lymphocytes were extracted using a brain dissociation kit (Miltenyi Biotec, 130-107-677), according to manufacturer instructions. In brief, colon and ileum were dissected and stored in HBSS (w/o). Faeces and fat tissue were removed, organs were opened longitudinally, washed in HBSS (w/o) and cut laterally into 0.5-cm-long pieces. Tissue pieces were transferred into 20 ml of a predigestion solution (HBSS (w/o), 5 mM EDTA, 5% fetal bovine serum (FBS), 1 mM dithiothreitol) and kept for 20 min at 37 °C under continuous rotation. Samples were then vortexed for 10 s and applied on a 100 μm cell strainer. The last two steps were repeated once. Tissue pieces were then transferred into HBSS (w/o) and kept for 20 min at 37 °C under continuous rotation. After vortexing for 10 s, tissue pieces were applied on a 100 μm cell strainer. Tissue pieces were then transferred to a GentleMACS C Tube (Miltenyi Biotec, 130-093-237) containing 2.35 ml of a digestion solution and homogenized on a GentleMACS Octo Dissociator (Miltenyi Biotec, 130-096-427, programme 37C_m_LPDK_1). Homogenates were resuspended in 5 ml PB buffer (phosphate-buffered saline (PBS), pH 7.2, with 0.5% bovine serum albumin), passed through a 70 μm cell strainer and centrifuged at 300 × *g* for 10 min at 4 °C. Cell pellets were resuspended in ice-cold PB buffer and stored on ice until

further use. Spinal cords were stored in ice-cold D-PBS until they were transferred to a GentleMACS C Tube containing a digestion solution. Samples were processed on a GentleMACS Octo Dissociator (programme 37C_ABDK_01) and rinsed through a 70 μm cell strainer. The cell suspension flow through was then centrifuged at 300 × *g* for 10 min at 4 °C. Debris removal was performed by resuspending the cell pellet in 1,550 μl D-PBS, adding 450 μl of debris removal solution and overlaying with 2 ml of D-PBS. Samples were centrifuged at 4 °C at 3,000 × *g* for 10 min. The two top phases were aspirated and the cell suspension was diluted with cold D-PBS. Samples were then inverted three times and centrifuged at 1,000 × *g* at 4 °C for 10 min, and the cells were resuspended and stored in ice-cold D-PBS until further use.

### Cell stimulation and flow cytometry

Cells (10$^6$) (MLN cell suspensions as well as lymphocyte extracts from CLP, SILP and SC) were resuspended in 1 ml complete cell culture medium (RPMI containing 10% FBS, 2 mM glutamine, 50 U ml$^{-1}$ penicillin, 50 μg ml$^{-1}$ streptomycin and 0.1% mercaptoethanol) supplemented with 2 μl Cell Activation Cocktail with Brefeldin A (Biolegend, 423304) and incubated for 4 h at 37 °C. Cells were centrifuged at 500 × *g* for 5 min, resuspended in 100 μl Zombie NIR (1:1,000 in PBS, Zombie NIR Fixable Viability kit, Biolegend, 423106), transferred into a 96-well plate and incubated for 20 min at 4 °C in the dark. Cells were washed two times with 150 μl PBS (centrifuged for 5 min at 400 × *g* at 4 °C) and resuspended in 50 μl Fc-block (1:50, purified rat anti-mouse CD16/CD32, BD, 553142) diluted in FACS buffer (1x PBS/2% FBS/2 mM, EDTA pH 8.0). Cells were incubated for 20 min at 4 °C in the dark and washed two times with 150 μl PBS with centrifugation for 5 min at 400 × *g* at 4 °C. All cells were fixed for 30 min with BD Cytofix/Cytoperm solution (BD, 554722) and stored in PBS overnight. For extracellular and intracellular fluorescent cell staining, cells were permeabilized with BD Perm/Wash buffer (BD, 554723) for 15 min. T lymphocytes were evaluated using the following antibodies: rat anti-mouse IL-17A (TC11-18H10.1, 1:50, Biolegend, 506922), rat anti-mouse RORγt (AFKJS-9, 1:44, eBiosciences, 17-6988-82), rat anti-mouse CD3 (17A2, 1:88, Biolegend, 100241), rat anti-mouse CD45 (30-F11, 1:88, BD, 564225), rat anti-mouse CD4 (RM4-5, 1:700, Biolegend, 100548), rat anti-mouse IFN-γ (XMG1.2, 1:175, eBiosciences, 61-7311-82), rat anti-mouse FoxP3 (FJK-16s, 1:200, ThermoFisher, 48-5773-82), rat anti-mouse CD8 (53-6.7, 1:700, Biolegend, 100710). Optimal staining concentrations of all antibodies were evaluated before staining. Cells were incubated with FACS buffer diluted antibodies for 30 min at 4 °C in the dark. Samples were washed twice with 150 μl of BD Perm/Wash buffer, resuspended in 200 μl PBS and acquired using NovoCyte Quanteon flow cytometer (NovoCyte Quanteon 4025, Agilent). All acquired data were analysed using FlowJo software (v.10.7.2, BD, 2019). Fluorescence minus one controls (FMOs) were used for each antibody–fluorophore combination to properly evaluate signal-positive and -negative cells. Single antibody-stained UltraComp eBeads Compensation Beads (Fisher Scientific, 01-2222-42) were used to create the compensation matrix in FlowJo. Single antibody-stained compensation beads were created for each run separately, using the exact same antibody lot that was used for the samples. Due to insufficient binding of the BV786-coupled rat anti-mouse CD45 antibody (30-F11, 1:88, BD, 564225) to the compensation beads, we used the BV786-coupled anti-mouse Siglec-F antibody (E50-2440, BD, 740956) to calculate the compensation matrix. Compensation samples were gated on the population of compensation beads within the FSC-H and SSC-H channels, and the positive and negative population for the corresponding antibody were identified in a two-dimensional depiction of channels with strong fluorescence spillover. Compensation matrices were calculated for each run separately and applied to the samples of this particular run. After applying the compensation matrix, samples underwent the gating strategy, which is depicted in Extended Data Fig. 7a,b. FACS analysis of isolated lymphocytes was performed blinded and the gating strategy was verified by at least two persons.

Sample quality was evaluated by assessing the event distribution in the 'SSC-H vs FSC-H' and the 'Live/Dead vs SSC-H' depiction, and samples of insufficient quality and event counts were removed from the analysis. To highlight the differential infiltration of CD45+ lymphocytes into spinal cords of EAE-induced mice, proportions of EAE-associated lymphocyte populations were calculated as a percentage of cells in the 'Viable' gate. To draw samples of comparable %V between the Mild/Severe EAE clusters and therefore reduce downstream analysis bias, we only included samples that provided %V within a range of mean ± s.d. of 'all' analysed samples (Extended Data Fig. 7e (top left panel)). After sample drawing, we reassigned samples to their respective clusters and their %V revealed no significant difference, thus eliminating remaining gating bias and providing a homogeneous parental population to address differences in downstream gates. Downstream analysis of relative proportions of target cell populations was performed using RStudio (v.4.2.1). Individual samples were grouped by target cell population, organ and EAE group phenotype (for non-EAE-induced mice) or by target cell population, organ and individual EAE severity cluster (for EAE-induced mice). Outliers were determined by group and values not within a range of mean ± 2σ were removed from the datasets.

## V4 16S rRNA gene sequencing and analysis

Bacterial DNA extraction from colonic and ileal content was performed as described previously[18]. A Qubit dsDNA HS assay kit (Life Technologies, Q32854) was used to quantify dsDNA concentrations. The V4 region of the 16S rRNA gene was amplified using dual-index primers (forward: 5′-GTGCCAGCMGCCGCGGTAA-3′; reverse: 5′-TAATCTWTGGGVHCATCAGG-3′) described in ref. 52. Library preparation was performed according to manufacturer protocol using the Quick-16S NGS Library Prep kit (Zymo Research, D6400). The pooled libraries were sequenced on an Illumina MiSeq system using MiSeq Reagent kit v.2 (500-cycle) (Illumina, MS_102_2003) at the Integrated BioBank of Luxembourg (IBBL, Dudelange, Luxembourg). The programme mothur (v.1.44.3)[53] was used to process the reads according to the MiSeq protocol. For gnotobiotic samples, taxonomy was assigned using a *k*-nearest neighbour consensus approach against a custom reference database corresponding to the SM14 taxa and potential contaminants (*Citrobacter rodentium*, *Lactococcus lactis* subsp. *cremoris*, *Staphylococcus aureus* and *S. epidermidis*). For SPF samples, taxonomy was assigned using the Wang approach against the SILVA v.132 database. Groupwise analysis of annotated reads was performed using RStudio (v.4.2.1) with an initial seed set at 8,765. Operational taxonomic units (OTUs) not constituting >0.01% of reads within at least one group (group means) were removed from the analysis. Diversity indices were determined using the 'diversity()' function of the 'vegan' package (v.2.6.2). Non-metric multidimensional scaling for Bray–Curtis distance matrices was calculated using the 'metaMDS()' function of the vegan package, and principal coordinate decomposition of weighted UniFrac distance matrices was calculated using the 'pcoa()' function of the 'ape' package (v.5.6.2). All analyses were performed at OTU, genus and family levels. OTUs and genera contributing most to community differences between selected groups were extracted using the 'simper()' function of the vegan package. Unless specified otherwise, all reported strain compositions for SM mice are based on V4 16S rRNA gene sequencing.

## Phylogenetic tree of SM14 constituent strains

The phylogenetic tree was constructed on the basis of full-length 16S rRNA gene sequences and analysed with Geneious Prime v.2021.2.2. European Nucleotide Archive (ENA) accession numbers can be found in Supplementary Table 1. A neighbour-joining tree build model was created using global alignment with free end and gaps, 65% similarity index cost matrix and a Tamura–Nei genetic distance model.

## Caecal and serum metabolomics processing and analysis

Caecal metabolites were extracted from ~10 mg of a freeze-dried sample by vigorous shaking with 500 μl of 100% methanol supplemented with 20 μM methionine sulfone as well as 20 μM D-camphor-10-sulfonic acid, as internal standards. Four 3 mm zirconia beads (BioSpec) and ~100 mg of 0.1 mm zirconia/silica beads (BioSpec) were added to this mix. Afterwards, samples were shaken vigorously for 5 min using a Shake Master NEO (BioSpec). Next, 500 μl of chloroform and 200 μl of Milli-Q water were added, and samples were again shaken vigorously for 5 min, followed by centrifugation at 4,600 × *g* for 30 min at 4 °C. The resulting caecal content supernatants as well as serum samples were transferred to a 5 kDa cut-off filter column (Ultrafree MC-PLHCC 250/pk) for metabolome analysis (Human Metabolome Technologies). The flow through was dried under vacuum. Residue was dissolved in 50 μl of Milli-Q water containing reference compounds (200 μM 3-aminopyrrolidine and 200 μM trimesic acid). The levels of extracted metabolites were measured by capillary electrophoresis–time of flight mass spectrometry (CE–TOF/MS) in both positive and negative ion modes, using an Agilent 7100 capillary electrophoresis system (Agilent) equipped with an Agilent 6230 TOF LC/MS system (Agilent). Metabolome spectra and concentration calculations (Supplementary Tables 2 and 3) were carried out using the software 'MasterHands' v.2.19.0.2 (Keio University, Japan), as previously described[54], in a blinded manner.

Caecal metabolite concentrations were converted from nmol g$^{-1}$ caecal content to pmol g$^{-1}$. Only 175 metabolites were included in downstream analyses, these metabolites being detectable in at least 50% of the samples in one of the seven tested groups. Undetectable concentrations of these 175 metabolites in certain samples were replaced by a value of 1/3 of the lowest detectable value of this particular metabolite across all groups. Concentrations were then normalized using a log$_2$ transformation for downstream analyses. Short-chain fatty acid quantification was performed as described previously[19].

## Metabolite-of-interest screening pipeline

Given that EAE phenotypes were disconnected from the overall metabolome pattern, we looked for single metabolites that might explain observed differences in EAE disease course. In particular, we were interested in determining which metabolites might be associated with the presence of *A. muciniphila*. Thus, we implemented a screening pipeline comprising 18 independent analyses to identify potential metabolites of interest that might explain *A. muciniphila*-associated differences in EAE outcomes. These analyses included correlation analyses of metabolite concentrations with EAE-associated readouts (Extended Data Fig. 5a), groupwise comparisons of metabolite concentrations (Extended Data Fig. 5b) and correlations with the presence of *A. muciniphila* or the presence of any microbiota. By combining information obtained from these analyses, our goal was to shortlist microbiota-induced caecal metabolites that enable prediction of EAE development in EAE-induced mice as well as to evaluate whether these are associated with *A. muciniphila*. We concluded that a potential metabolite of interest should fulfil four criteria: (1) an overall significantly different concentration among all 7 groups (Fig. 3c, 'SIG' and Extended Data Fig. 5c, 'SIG'), as determined by one-way analysis of variance (ANOVA). As we have observed differences in EAE outcome on a group-based level, this should also be reflected in different concentrations of a metabolite of interest among the groups; (2) a significant correlation with all tested EAE-associated readouts (Fig. 3c, 'AUC', 'RelM', 'SusO', 'Max' and Extended Data Fig. 5a,c) in EAE-induced mice; (3) a significant correlation with the presence of *A. muciniphila* (Fig. 3c, 'AM' and Extended Data Fig. 5c), since we observed different EAE phenotypes based on the presence or absence of *A. muciniphila*; and (4) a significantly different concentration when comparing non-EAE-induced SM13-colonized mice with non-EAE-induced SM14-colonized mice, as harbouring these microbiota compositions led to different EAE phenotypes upon EAE induction (Fig. 2c (left)). This criterion would allow for assessing the prediction aspect of caecal metabolite concentrations.

## Metatranscriptome analyses

Flash-frozen caecal contents were stored at −80 °C until further processing. RNAProtect Bacterial Reagent (1 ml, Qiagen, 76506) was added to each sample and thawed on wet ice for 10 min. Samples were centrifuged at 10,000 × $g$ at 4 °C for 10 min and RNAProtect Bacterial Reagent was removed by pipetting. Next, 250 µl acid-washed glass beads (212–300 µm), 500 µl of buffer A (0.2 M NaCl, 0.2 M Trizma base, 20 mM EDTA pH 8), 210 µl of 20% SDS and 500 µl of phenol–chloroform–isoamyl alcohol (125:24:1, pH 4.3) were added to the pellet. Bead beating on the highest frequency (30 Hz) was performed for 5 min using a mixer mill and the aqueous phase was recovered after centrifugation for 3 min at 18,000 × $g$ at 4 °C. Phenol–chloroform–isoamyl alcohol (500 µl, 125:24:1, pH 4.3) was added to each sample and centrifuged as previously described. Again, the aqueous phase was recovered and 1/10 volume of 3 M sodium acetate (pH ~5.5) and 1 volume of ice-cold 100% ethanol were added and gently mixed by inversion. Samples were incubated for 20 min on wet ice, washed twice with 500 µl of ice-cold 70% ethanol and centrifuged for 5 min at 18,000 × $g$ at 4 °C. Pellets were air dried for 10 min and resuspended in 50 µl nuclease-free water. DNase treatment was performed by adding 10 µl 10X buffer, 40 µl nuclease-free water (to reach 100 µl final volume) and 2 µl DNase I (Thermo Scientific, DNase I, RNase-free kit, EN0521), followed by 30 min incubation at 37 °C. 0.5 M EDTA (1 µl per sample) was added and heat inactivated for 10 min at 65 °C. RNA purification was performed with the RNeasy Mini kit (QIAGEN, 74106) according to manufacturer instructions. RNA quantity and quality were assessed using the RNA 6000 Nano kit on an Agilent 2100 Bioanalyzer. Library preparation was performed using a Stranded Total RNA Prep, Ligation with Ribo-Zero Plus kit (Illumina, 20040529). Pooled libraries were then sequenced in a 2 × 75 bp configuration on an Illumina NextSeq 550 platform using a High Output flow cell, followed by Medium Output flow cell at the LuxGen Platform. RNA sequencing files were pre-processed using kneaddata (https://github.com/biobakery/kneaddata). Adapters were removed using Trimmomatic[55] and fragments below 50% of the total expected read length (75 bp) were filtered out. BowTie2 (ref. [56]) was used to map and remove contaminant reads corresponding to either rRNA databases or the *Mus musculus* genome. Clean fastq files were concatenated before passing to HUMAnN3 (ref. [57]). A custom taxonomy table based on pooled 16S rRNA sequencing abundances was provided to MetaPhlAn for metagenome mapping. Unaligned reads were translated for protein identification using the UniRef90 database provided within HUMAnN3. Data for all samples were joined into a single table and normalized to counts per million reads (CPM). Results were grouped by annotated gene product per individual. In case no annotation from UniRef90 transcript IDs was possible, distinct IDs were treated as separate gene products. Only gene products that provided >50 CPM in at least two of the eight investigated samples were included in downstream analyses. This resulted in 2,213 transcripts being included in downstream analyses, representing 80%–85% of the total CPM, with no significant differences between the analysed groups. CPMs were recalculated to account for removed transcripts, followed by further analysis using the 'edgeR' package (v.3.38.4) in R Studio (v.4.2.1). Multidimensional reduction of the transcriptome profiles was calculated using the log(fold change (FC)) method within the 'plotMDS.DGEList' function. Groupwise comparison of gene expression was calculated using the 'exactTest()' function.

## Bacterial IgA coating index

To determine bacterial IgA coating indices, we followed a previously published approach[31], with adaptations as described. Faecal samples stored at −20 °C were resuspended in 500 µl ice-cold sterile PBS per faecal pellet and mechanically homogenized using a plastic inoculation loop. Pellets were then thoroughly shaken on a thermomixer (Eppendorf) for 20 min at 1,100 r.p.m. and 4 °C. After adding 2× volume of ice-cold PBS, samples were centrifuged for 3 min at 100 × $g$ at 4 °C to

sediment undissolved debris. Clear supernatant was recovered and passed through a 70 µm sieve (PluriSelect, 43-10070-40), followed by centrifugation for 5 min at 10,000 × $g$ at 4 °C to sediment bacteria. Supernatant was removed and the pellet resuspended in 1 ml ice-cold PBS. Next, optical density of this suspension at 600 nm ($OD_{600}$) was detected and the approximate concentration of bacteria was estimated on the basis of the assumption that $OD_{600} = 1$ equals $5 × 10^8$ bacteria per ml. The respective volume corresponding to $10^9$ bacteria was centrifuged for 5 min at 10,000 × $g$ at 4 °C. The pellet was then resuspended in 400 µl of 5% goat serum (Gibco, 11540526) in PBS and incubated for 20 min on ice. After incubation, the pellet was washed once in ice-cold PBS and centrifuged for 5 min at 10,000 × $g$ at 4 °C. The pellet was then resuspended in 100 µl ice-cold PBS containing 4 µg of fluorescein isothiocyanate (FITC)-coupled goat anti-mouse IgA antibody (SouthernBiotech, Imtec Diagnostic, 1040-02). The ratio of 4 µg of the respective antibody to stain $10^9$ bacteria was previously evaluated to be the maximum amount of antibody that can be used without resulting in unspecific staining of non-IgA-coated bacteria, using faecal samples from *Rag1*$^{-/-}$ mice as non-IgA-coated negative controls. Samples were then incubated for 30 min at 4 °C on a thermomixer (Eppendorf) at 800 r.p.m. After incubation, 1 ml ice-cold PBS was added, followed by centrifugation for 5 min at 10,000 × $g$ at 4 °C. Samples were then washed once in ice-cold PBS and subjected to either flow-cytometry detection or separation of IgA$^+$ and IgA$^-$ bacteria. For immediate flow cytometric detection, pellets were resuspended in 200 µl DNA staining solution (0.9% NaCl in 0.1 M HEPES, pH 7.2, 1.25 µM Invitrogen SYTO 60 Red Fluorescent Nucleic Acid Stain, Fisher Scientific, 10194852), followed by incubation for 20 min on ice. After washing once with PBS, pellets were resuspended in 100 µl PBS and run immediately on a NovoCyte Quanteon (NovoCyte Quanteon 4025, Agilent). For separation of IgA$^+$ from IgA$^-$ bacteria, we used the MACS cell separation system from Miltenyi. Pellets were resuspended in 100 µl staining buffer (5% goat serum in PBS) containing 10 µl anti-FITC microbeads (Miltenyi, 130-048-701) per $10^9$ bacteria, mixed well and incubated for 15 min at 4 °C. After the end of the incubation time, 1 ml of staining buffer was added, followed by centrifugation for 5 min at 5,000 × $g$. Pellets were then resuspended in 5 ml staining buffer per $10^9$ bacteria, loaded onto MACS LD separation columns (Miltenyi, 130-042-901), and flow through containing the IgA$^-$ fraction was collected. After removing columns from the magnet, the IgA$^+$ fraction was flushed out and collected. The IgA$^+$ fraction was then loaded on a MACS LS separation column (Miltenyi, 130-042-401). Flow through was collected and combined with the previous IgA$^-$ fraction. After removing columns from the magnet, the IgA$^+$ bacteria fraction was flushed out, collected and combined with the previous IgA$^+$ fraction. Both combined fractions, IgA$^+$ and IgA$^-$, were then centrifuged for 10 min at 10,000 × $g$ at 4 °C. Pellets were then resuspended in 1 ml PBS and subjected to two different downstream analyses: (1) To test the purity of both fractions for each sample by flow cytometry, 10% of the suspension volume was used for bacterial DNA staining using SYTO 60 Red Fluorescent Nucleic Acid Stain as described above; (2) To purify bacterial DNA for subsequent V4 (SM mice) or full-length (SPF mice) 16S rRNA gene sequencing of bacteria within the different fractions, 90% of the suspension volume was centrifuged for 10 min at 10,000 × $g$ at 4 °C, supernatant was discarded and the dry pellet was stored at −20 °C. DNA isolation and 16S rRNA gene sequencing was then performed as described below. The ICI for a given species $x$ ($ICI_x$) was calculated using the following equation, with $A_x^+$ representing the strain-specific relative abundance in the IgA$^+$ fraction and $A_x^-$ representing the strain-specific relative abundance in the IgA$^-$ fraction: $ICI_x = \log_{10}\left(\frac{A_x^+}{A_x^-}\right)$.

## Full-length 16S rRNA gene amplicon sequencing and analysis

DNA was isolated from IgA$^+$, IgA$^-$ and total (unsorted) fractions of mouse faecal samples as described previously[18]. After isolation, we performed PCR-assisted full-length 16S rRNA gene amplification using 16S

rRNA gene-specific forward (5′-AGAGTTTGATCCTGGCTCAG-3′) and reverse (5′-ACGGCTACCTTGTTACGACTT-3′) primers with an annealing temperature of 51 °C. All 16S rRNA gene-amplified samples were purified using the NucleoSpin Gel and PCR Clean-up kit (Macherey Nagel) according to manufacturer instructions and following recommended adaptations for reads >1 kbp. Double-stranded DNA concentrations after cleanup were determined using the Qubit dsDNA HS assay kit (Life Technologies, Q32854) on an Invitrogen Qubit 4.0 fluorometer (Life Technologies, Q33226). DNA repair and end preparation, native barcode ligation, adapter ligation and cleanup were performed as described in the Native Barcoding Kit 96 V14 (SQK-NBD114.96) protocol from 15 September 2022, which was provided by the manufacturer in the Nanopore Community (Oxford Nanopore). The barcoded sample library was loaded dropwise onto a MinION flow cell (R10.4.1) and the sequencing run was initiated in 400 bps output mode in MinKNOW (v.23.07.15) for 65 h, generating ~6.8 million reads (N50 1.5 kb). Raw fast5 data files were converted to pod5 format using POD5 Tools (v.0.2.0), then basecalled in super-accuracy mode and demultiplexed according to the barcodes of the SQK-NBD114-96 kit on a gpu partition using Dorado (v.0.4.3). The demultiplexed bam files were converted to fastq format using samtools (v.1.16.1) bam2fastq, then filtered using NanoFilt (v.2.8.0) such that only Phred quality scores above 10 and read lengths between 1,300 and 1,700 bp were retained. Taxonomic classification was carried out using emu (v.3.4.5) with the −keep-counts flag and the default Emu 3.0+ database, which combines rrnDB v.5.6 and NCBI 16S RefSeq from 17 September 2020, and taxonomy from NCBI on the same date. Completely unassigned reads were removed from downstream analyses. The 'rarecurve()' function of the vegan package (v.2.6-4) was used to identify undersampled samples, which were also removed from downstream analyses. Count tables of the remaining samples were used to determine alpha-diversity using the phyloseq package (v.1.40.4). Bacterial features not constituting at least 0.01% of reads in at least two mice within each group of microbiota–donor and microbiota–recipient combination were removed from the analysis. Read counts were normalized by calculating relative abundances, followed by arcsine square root transformation. Beta-diversity measures were determined using the phyloseq package (v.1.40.0). Relative abundances per taxonomic unit and sample were visualized using the fantaxtic package (v.0.2.0).

### Secretory IgA measurements in faeces

For overnight coating of high-binding 384-well plates (Greiner, 781061), we used 10 ng per well of rabbit anti-mouse IgA (Novus, NB7506) in 20 µl per well of carbonate-bicarbonate buffer (Sigma, C3041). Plates were then washed four times in wash buffer (10 mM Trizma Base, 154 mM NaCl, 1% (v/v) Tween-10). Next, 75 µl of a blocking buffer (15 mM Trizma acetate, 136 mM NaCl, 2 mM KCl, 1% (w/v) bovine serum albumin) was added to each well and incubated for 2 h at r.t. Following another washing step with wash buffer, samples and standards were diluted in a dilution buffer (blocking buffer + 0.1% (w/v) Tween-20). As standards, we used a mouse IgA isotype control (Southern Biotech, 0106-01). A volume of 20 µl of the dilutions was added to each well and incubated for 90 min at r.t. Following another washing step, a secondary goat anti-mouse IgA antibody conjugated with alkaline phosphatase (Southern Biotech, 1040-04) and diluted 1:1,000 in dilution buffer was added. Secondary antibody was incubated at r.t. for 90 min and plates were washed four times. As a substrate, 1 phosphate tablet (Sigma, S0642-200 TAB) was solubilized in 10 ml of substrate buffer (1 mM 2-amino-2-methyl-1-propanol and 0.1 mM MgCl$_2$ × 6H$_2$O). Of this substrate solution, 40 µl was added to each well, followed by incubation at 37 °C for 60 min. Final absorbance at 405 nm was detected using a SpectraMax Plus 384 microplate reader.

### Quantification of lipopolysaccharides in serum

Quantification of LPS levels in serum was performed using the Pierce Chromogenic Endotoxin Quantification kit (ThermoScientific, A39552)

according to manufacturer instructions. Thawed serum samples were heat shocked for 15 min at 70 °C and diluted 1:50 before performing the assay. After blank reduction, final endotoxin levels were calculated on the basis of detected OD of supplied standards and using RStudio (v.4.2.1), applying a 4-parameter nonlinear regression of standard ODs with the help of the 'drc' package (v.3.0.1) and using the 'drm(OD-concentration, fct = LL.4())' function. Sample concentrations were then extracted using the 'ED(type = 'absolute')' function of the same package.

### Quantification of zonulin and occludin in serum

To measure concentrations of zonulin (ZO-1) and occludin (OCLN) in serum, we used the Mouse Tight Junction Protein ZO-1 ELISA kit (MyBioSource, MBS2603798) and the Mouse Occludin (OCLN) ELISA kit (Reddot Biotech, RD-OCLN-Mu), respectively, according to manufacturer instructions. After blank reduction, final ZO-1 and OCLN concentrations were calculated on the basis of detected ODs of supplied standards and using R Studio (v.4.2.1), applying a 4-parameter nonlinear regression of standard ODs with the help of the drc package (v.3.0.1) and using the drm(OD-concentration, fct = LL.4()) function. Sample concentrations were then extracted using the ED(type = 'absolute') function of the same package.

### Quantification of lipocalin 2 in faeces

To measure faecal lipocalin 2 (LCN2) levels, faecal pellets were homogenized in 500 µl ice-cold PBS with 1% Tween-20. Samples were then subjected to agitation for 20 min at 4 °C at 2,000 r.p.m. on a thermomixer (Eppendorf), followed by centrifugation for 10 min at 21,000 × $g$ at 4 °C. Pellets were discarded, and supernatants were collected and stored at −20 °C until further use. Final LCN2 detection was conducted using the Mouse Lipocalin 2/NGAL DuoSet Elisa (R&D Systems, DY1857) according to manufacturer instructions.

### Bacterial relative abundances by quantitative real-time PCR

To detect relative abundances of commensal bacteria from faecal samples obtained from mice harbouring reduced communities (SM01 to SM14) in a gnotobiotic setting, we followed a qPCR protocol published elsewhere[20] without modifications. Primer sequences for strain-specific quantification are listed in Supplementary Table 1.

### Glycan-degrading enzyme activities in faeces

To detect activities of α-fucosidase, α-galactosidase, β-glucosidase, β-$N$-acetyl-glucosaminidase and sulfatase from faecal samples stored at −20 °C, we followed a previously published protocol[58] without modifications.

### Statistical analyses

All reported values derived per mouse represent biological replicates. Datasets that were expected to have normalized distribution of residuals were analysed using one-way ANOVA with Tukey's post-hoc test. Data distribution was assumed to be normal, but this was not formally tested. Non-normal or ordinal data were analysed using Kruskall–Wallis test followed by multiple Wilcoxon rank-sum tests, with $P$-value adjustment using the Benjamini–Hochberg method. Correlation analyses report unadjusted $P$ values. Directionality was never assumed, therefore all tests performed were two-sided. All boxplots display individual points overlain on the median as a measure of central tendency, which is boxed by the interquartile range (IQR, first or 25th percentile to third or 75th percentile), with whiskers at the 'minimum' (first quartile minus 1.5 × IQR) and the 'maximum' (third quartile plus 1.5 × IQR). Statistical significance was defined as an adjusted $P < 0.05$, and non-significant comparisons are either unlabelled or labelled with 'NS'.

### Reporting summary

Further information on research design is available in the Nature Portfolio Reporting Summary linked to this article.

## Data availability

The raw files for this study from 16S rRNA gene sequencing (V4 and full-length amplicons) and RNA sequencing have been deposited in the European Nucleotide Archive (ENA) at EMBL-EBI under accession number PRJEB60278 (https://www.ebi.ac.uk/ena/browser/view/PRJEB60278). Raw spectral data from CE–TOF/MS metabolomic analyses are available on the MetaboBank integrated metabolome data repository under Project ID MTBKS223 (https://mb2.ddbj.nig.ac.jp/study/MTBKS223.html). Flow cytometry data are available on the Zenodo data repository (https://doi.org/10.5281/zenodo.12528901)[59]. Source data are provided with this paper.

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

## Acknowledgements

This work was supported by the following grants in the laboratory of M.S.D.: Luxembourg National Research Fund (FNR) CORE grants (C15/BM/10318186 and C18/BM/12585940), BRIDGES grant (22/17426243) and FNR INTER Mobility grant (16/11455695) to M.S.D.; FNR AFR bilateral grant (15/11228353) to M.N.; FNR AFR individual PhD fellowship (11602973) to A.P.; E.T.G. was supported by the FNR PRIDE grant (PRIDE17/11823097) and the Fondation du Pélican de Mie et Pierre Hippert-Faber, under the aegis of the Fondation de Luxembourg. We thank MEDICE Arzneimittel Pütter GmbH and Co. KG, Germany, and Theralution GmbH, Germany, for funding support through the public–private partnership FNR BRIDGES grant (22/17426243). H.O. was supported by Japan Agency for Medical Research and Development (AMED), Moonshot Research and Development Program (JP22zf0127007) and the Japan Society for the Promotion of Science KAKENHI (22H00452). E.M. was supported by the JSPS KAKENHI (19K05907), Lotte Foundation, the Waksman Foundation of Japan, the Iijima Memorial Foundation for the Promotion of Food Science and Technology, and Astellas Foundation for Research on Metabolic Disorders. S.F. (including CE–TOF/MS metabolomic analyses) was supported by JSPS KAKENHI (22H03541), JST ERATO (JPMJER1902), AMED-CREST (JP22gm1010009) and the Food Science Institute Foundation. We gratefully acknowledge N. Nicot, B. De Witt, L. Castillo and W. Ammerlaan at the LuxGen Platform and the Integrated BioBank of Luxembourg (IBBL) for sequencing assistance; and M. Kostadinou and F. Hedin from the National Cytometry Platform (NCP) for assistance with flow cytometry. The NCP is supported by Luxembourg's Ministry of Higher Education and Research (MESR) funding. We also thank C. Jäger, X. Dong and F. Gavotto from the Luxembourg Centre for Systems Biomedicine (LCSB) Metabolomics Platform for support in GC–MS analyses, and K. McCoy for providing us with *Muc2*$^{-/-}$ mice from the University of Bern. Finally, we thank the members of the Nutrition, Microbiome and Immunity team at the LIH for providing constructive feedback during the revision of the manuscript: Y. Baumann, H. De Franco, M. Joja and A. Rolof. For the purpose of open access, and in fulfilment of the obligations arising from the grant agreement, the author has applied a Creative Commons Attribution 4.0 International (CC BY 4.0) license to any Author Accepted Manuscript version arising from this submission.

## Author contributions

M.S.D. supervised the research, obtained research funding and coordinated the study. A.S., M.N., M.O., E.M., H.O. and M.S.D. conceived the research. A.S., M.N., E.M., H.O. and M.S.D. designed the experiments. A.S., M.N., S.W. and A.D.S. performed the mouse experiments and carried out the downstream experiments. E.T.G. and A.P. assisted with mouse experiments and downstream analyses. A.S. and M.N. performed the EAE scoring. T.S. and S.F. carried out CE–TOF/MS metabolomics analyses. A.S., M.N. and E.T.G. analysed the data. A.S. prepared the R scripts and the graphs. A.S., M.N., E.T.G. and M.S.D. primarily wrote and edited the manuscript. All authors reviewed the manuscript.

## Competing interests

M.S.D. works as a consultant and an advisory board member at Theralution GmbH, Germany. S.F. is a founder and CEO of Metagen, Inc., Japan, focused on the design and control of the gut environment for human health. The other authors declare no competing interests.

## Additional information

**Extended data** is available for this paper at https://doi.org/10.1038/s41564-024-01761-3.

**Correspondence and requests for materials** should be addressed to Mahesh S. Desai.

**COMPLEX MICROBIOTA**

**STEP 1** EAE-associated bacterial genera in complex communities (**Fig. 1**)

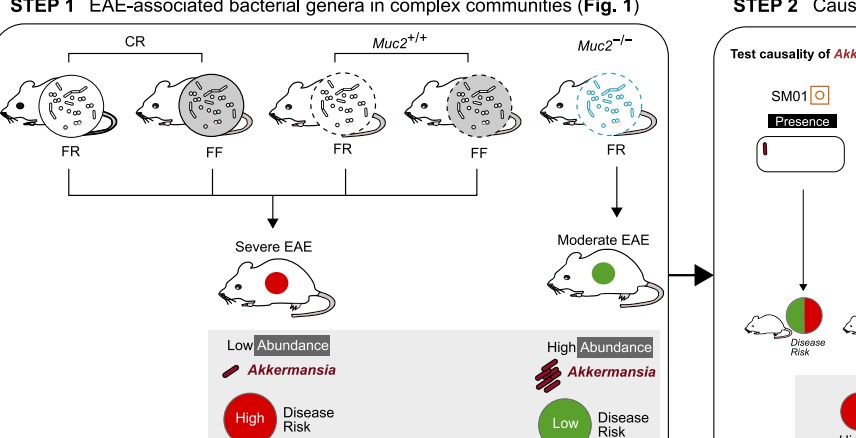

**REDUCED MICROBIOTA**

**STEP 2** Causality test (**Fig. 2** to **Fig. 4**)

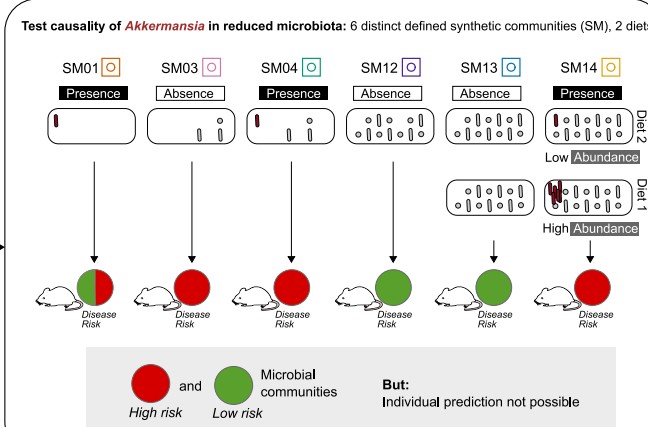

**STEP 4** Verification of predictor in complex microbial communities (**Fig. 6**)

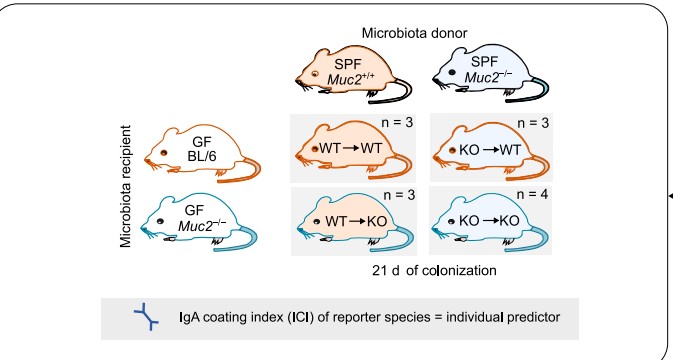

**STEP 3** Identification of individual predictors (**Fig. 5**)

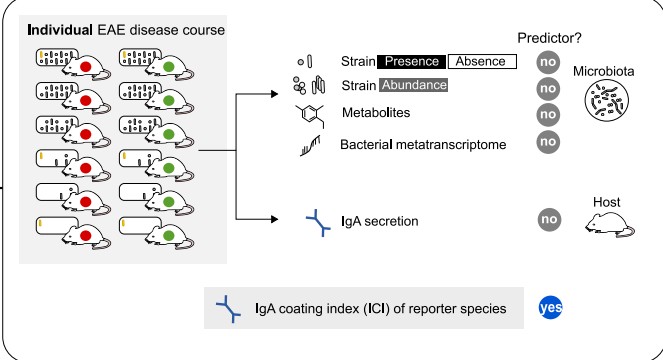

Conclusion

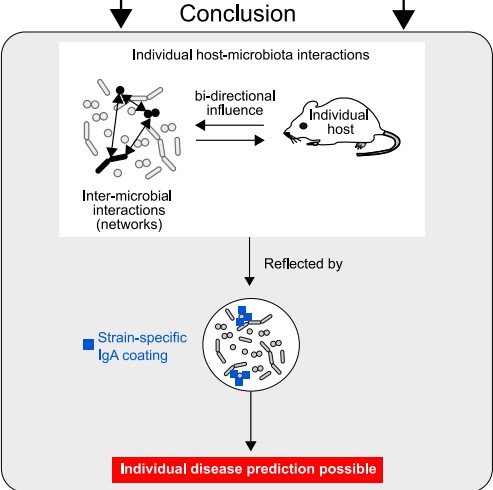

**Extended Data Fig. 1 | See next page for caption.**

**Extended Data Fig. 1 | Graphical summary of experiments, analyses and conclusion.** Step 1 (see also Fig. 1): First, we evaluated which bacterial genera were most associated with EAE disease development across genetically heterogenous mice harbouring 5 distinct complex microbiota compositions. We found that low relative abundance of *Akkermansia* (dark red bacteria icon) was associated with higher risk of severe EAE development (red circle), and higher abundances of this genus was associated with moderate disease and low disease risk (green circle). Step 2 (see also Figs. 2–4): Thus, we wondered whether presence or relative abundance of the *Akkermansia* type species *Akkermansia muciniphila* before induction of EAE could predict subsequent disease development. To test this, we induced EAE in mice harbouring 6 different combinations of a well-characterised 14-strain consortium under gnotobiotic conditions. In mice harbouring these microbiotas, *A. muciniphila* was either present or absent, and if present, *A. muciniphila* provided either high or low relative abundances. We found that distinct microbiota compositions resulted in 'high risk' and 'low risk' microbiota compositions, as determined by the proportion of mice developing severe disease. However, disease susceptibility was not uniform across mice harbouring the same microbiota. Step 3 (see also Fig. 5): We further determined that neither relative abundance, nor the presence or absence of *A. muciniphila*, or any other strain of the tested consortium, could reliably predict EAE development across distinct communities in individual mice. However, we found that the pre-EAE IgA coating index (ICI) of a consortium member strain, *Bacteroides ovatus*, significantly correlated with individual EAE outcome after disease induction, irrespective of the microbiota composition. Step 4 (see also Fig. 6): We then successfully verified the potential for species-specific ICIs, as determined before disease induction, to predict individual disease severity in genetically distinct mice containing various complex microbiotas. Conclusion: Making predictions on EAE development based on microbiota characteristics (that is assessing the individual disease risk) is possible, however, it must take into account inter-microbial interactions ('networks') within a given, individual community and host-specific responses to a certain microbiota composition.

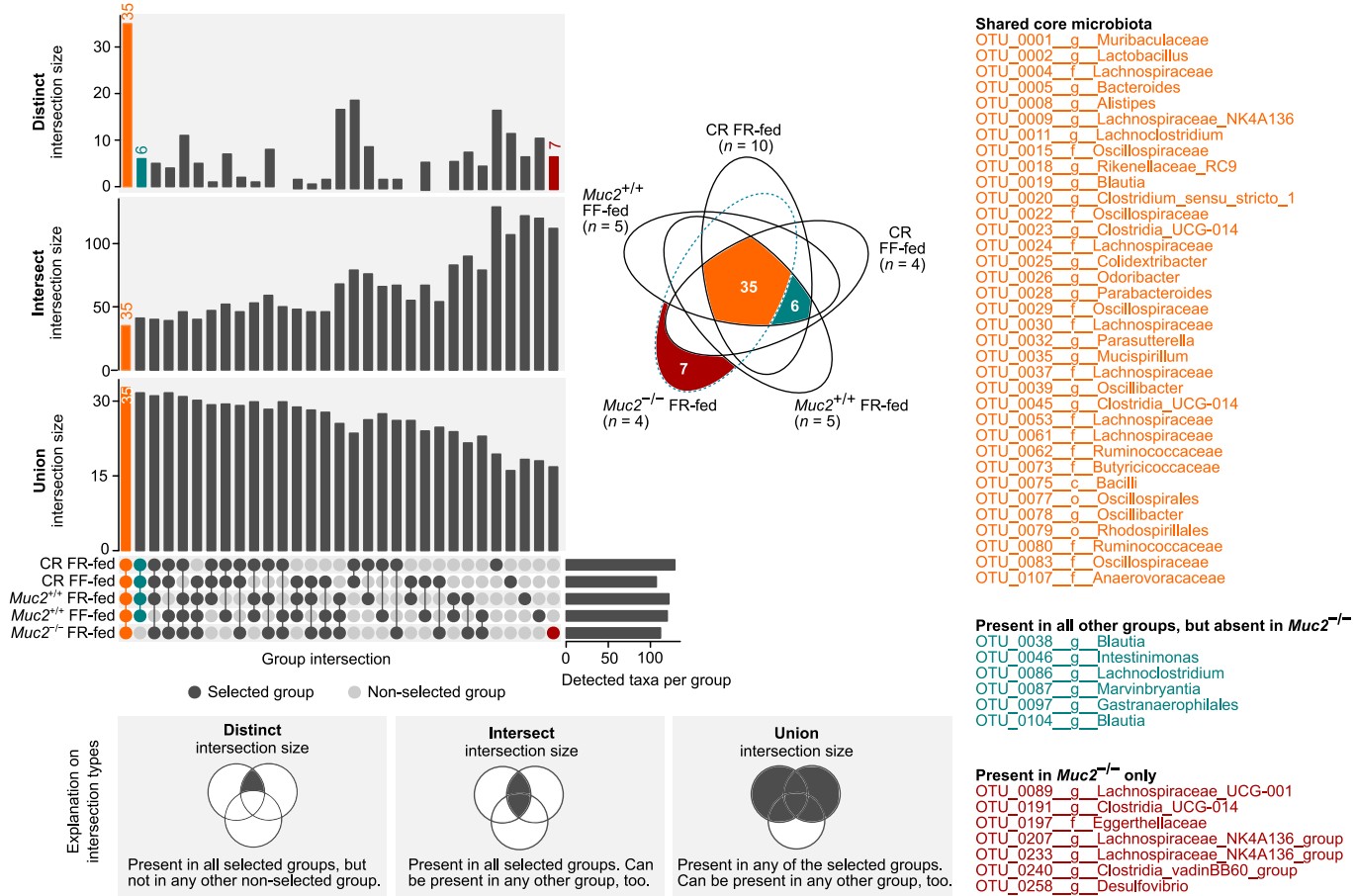

**Extended Data Fig. 2 | Operational taxonomic unit level-based microbiota analysis of SPF-housed mice.** All samples were analysed together using the same analysis pipeline. Only operational taxonomic units (OTUs) that provided a mean relative abundance of > 0.01% in at least one group were included in the analyses. Intersection analysis to identify taxa, which are either present or absent only in *Muc2⁻/⁻* mice. Taxa were considered 'present' within a certain group, when group mean relative abundance of a given taxon was > 0.01%. Otherwise, taxa were considered 'absent' in the respective group. Left panels, intersection sizes of OTUs between the five groups. Three different intersection sizes are shown ('Distinct', 'Intersect', and 'Union'), which are explained in the lower panels. Middle panel, Venn diagram highlighting the number of shared and distinct OTUs between the groups. Dark red, taxa only present in *Muc2⁻/⁻* mice; teal, taxa not present in *Muc2⁻/⁻* mice, but in all other mice; orange, core microbiome shared by mice of all five groups. List of OTUs belonging to these three groups provided on the right side.

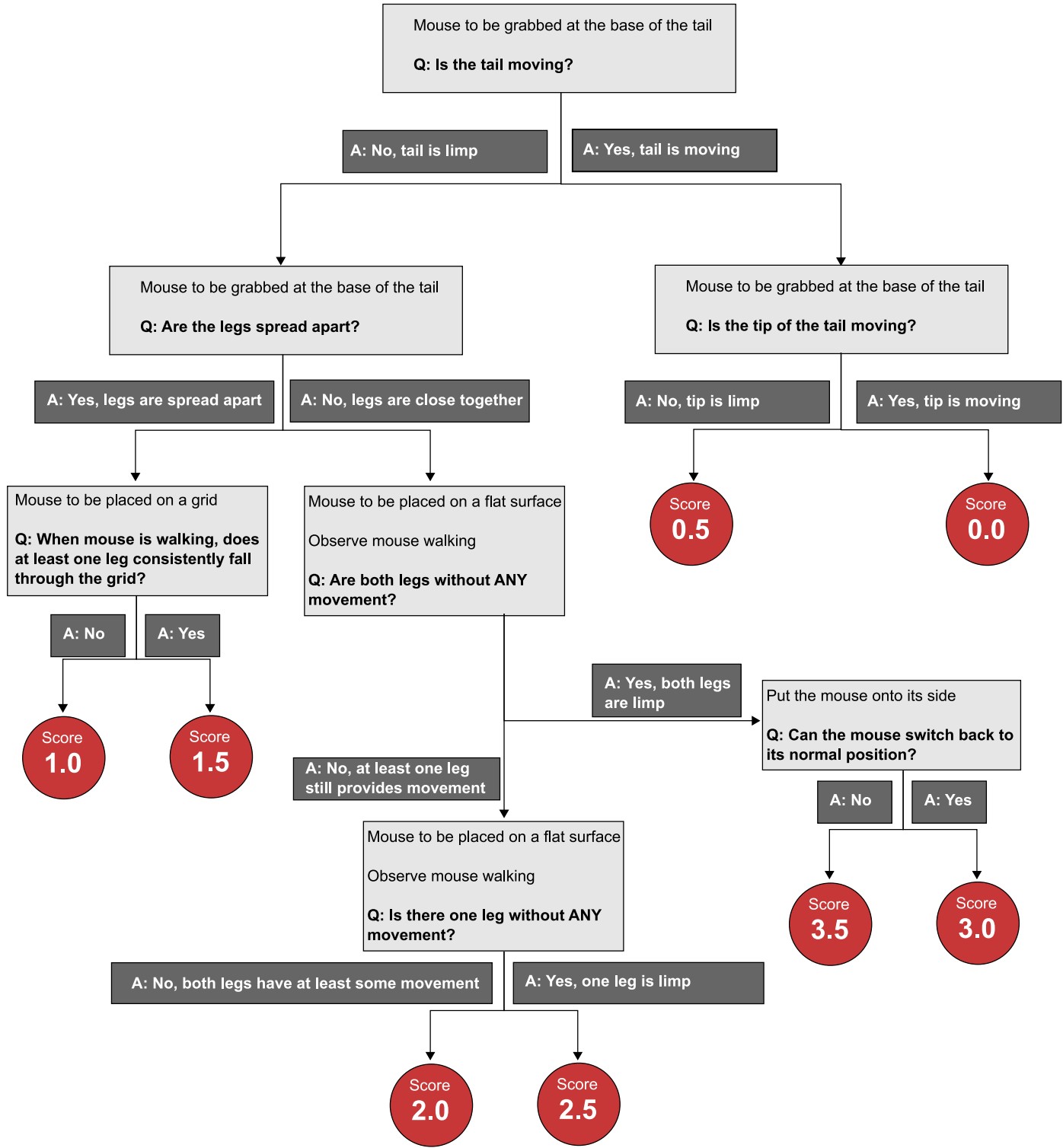

**Extended Data Fig. 3 | Experimental autoimmune encephalomyelitis scoring scheme.** This decision tree depicts how daily assessment of experimental autoimmune encephalomyelitis (EAE) scores was performed. Light grey boxes indicate instructions on mouse handling and EAE phenotype-associated questions (Q) which are highlighted in bold font. Dark grey boxes indicate possible answers (A) to the questions (Q). Red circles indicate the resulting EAE score. All arrows (answer options) are mutually exclusive.

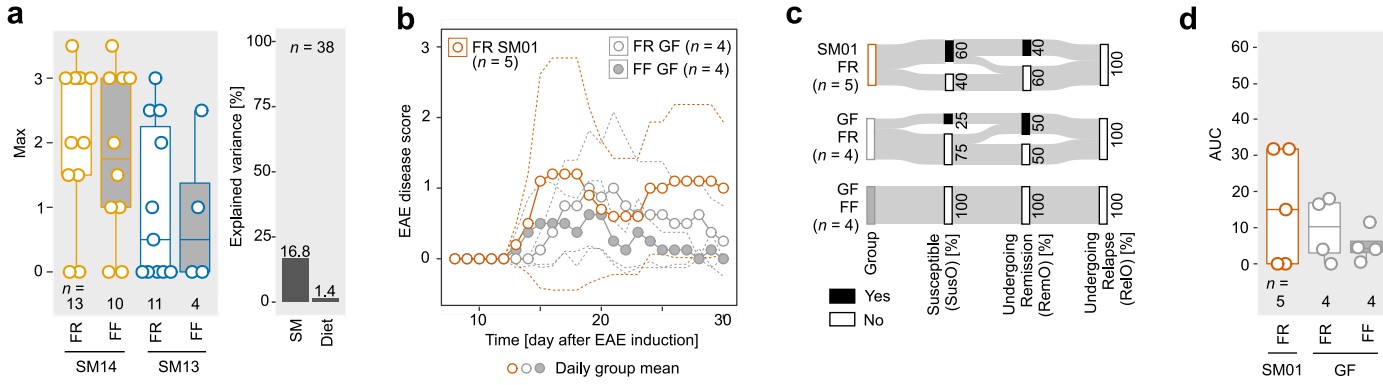

**Extended Data Fig. 4 | Experimental autoimmune encephalomyelitis disease course in germ-free, SM01-, SM13- and SM14-colonized mice. a**, Left, maximum achieved EAE score per individual (Max) within FR-fed SM13-colonized mice, FF-fed SM13-colonized mice, FR-fed SM14-colonized mice and FF-fed SM14-colonized mice. Wilcoxon rank-sum test with p-value adjustment using the Benjamini-Hochberg method. No statistically significant group differences were observed. Right, percentage of variance explained by diet (FR vs. FF) and SM combination (SM13 vs. SM14) as determined by eta-squared calculation when comparing the maximum achieved EAE score during the 30 d disease observation period between FR- and FF-fed SM14- and SM13-colonized mice. **b**–**d**, Germ-free (GF) C57BL/6N mice were either monocolonized with *A. muciniphila* (SM01) or remained GF and were fed either FR or FF diet. SM01 mice were only fed the

FR diet. EAE was induced in GF mice 20 d after diet switch. EAE was induced in SM01-colonized mice 25 d after initial colonization. Disease course in all groups was observed for 30 days after EAE induction. **b**, EAE disease scores as a function of time (days after EAE induction) for FR-fed GF mice, FF-fed GF mice and FR-fed SM01-colonized mice. Dots represent daily group means. Dashed lines represent SD. **c**, Sankey diagram of key event occurrence (in % of all mice within one group) during EAE. **d**, AUC analysis of the disease course depicted in panel **b**. Each mouse is depicted by a separate dot. One-way ANOVA followed by Tukey's post-hoc test. No statistically significant group differences were observed. Mouse numbers are indicated on the respective panels and treated as biological replicates. Boxplots (**a**, **d**) show median, quartiles, and 1.5 × interquartile range.

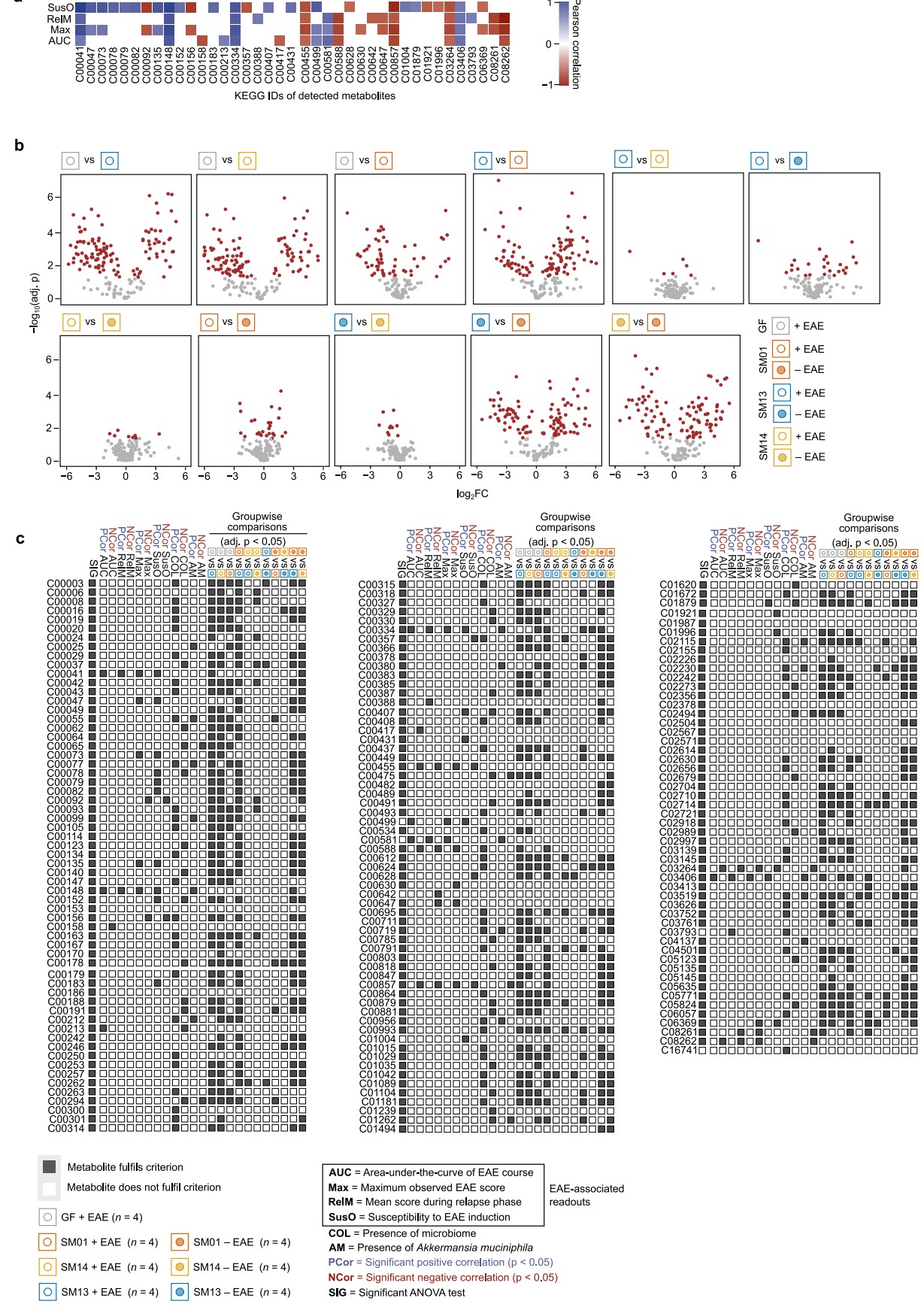

**Extended Data Fig. 5 | See next page for caption.**

**Extended Data Fig. 5 | Metabolite-of-interest analysis pipeline. a**, Pearson's correlation of metabolite concentrations with key EAE-associated readouts (AUC, Max, RelM, and SusO). Only samples from EAE-induced mice were used for analysis. Significant correlations are shown in either blue (positive) or red (negative). Non-significant correlations are not shown. Metabolites are listed under their KEGG ID (for GABA, C00334). A list of metabolite names with corresponding KEGG IDs is provided in Supplementary Table 2. **b**, Volcano plots of groupwise comparisons as indicated by the colour-coded legend for $\log_2$-normalized metabolite concentrations based on unpaired t-tests with p-value adjustment using the Benjamini-Hochberg method. Each dot represents one metabolite. Metabolites with significantly different (adjusted $p < 0.05$)

concentrations are highlighted in red. **c**, Criteria intersection analysis of the 175 detected metabolites that were identified in at least 50% of the samples of at least one group. Criteria were categorized into correlation criteria, summarizing the results of either statistically significant positive (PCor) or statistically significant negative (NCor) Spearman correlations across all samples of all groups (both EAE-induced and non-induced mice) and groupwise comparison criteria. Correlations referring to EAE-associated readouts were calculated using samples from EAE-induced mice only. Groupwise comparisons include metabolites found to be significantly different (adjusted $p < 0.05$) based on unpaired t-tests of $\log_2$-normalized concentrations with p-value adjustment using the Benjamini-Hochberg method.

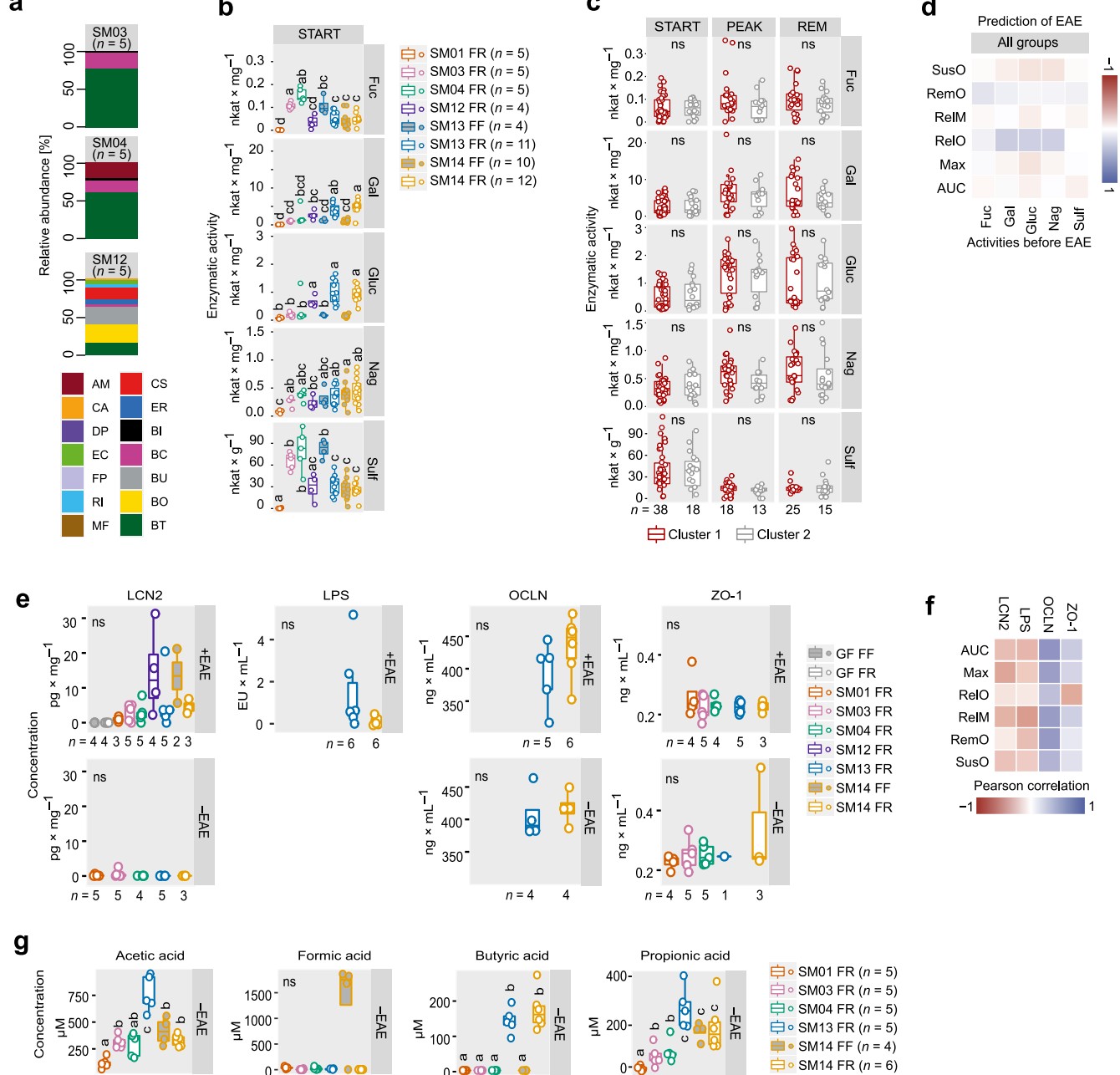

**Extended Data Fig. 6 | Analysis of barrier integrity and mucin-associated glycan degrading capacities of reduced microbiota compositions.**
**a**, Strain relative abundances in SM03-, SM04- and SM12-colonized mice after initial colonization as detected by qPCR using strain-specific primers (Supplementary Table 1). **b–e**, Activities of α-fucosidase (Fuc), α-galactosidase (Gal), β-glucosidase (Gluc), β-N-acetyl-glucosaminidase (Nag), and sulfatase (Sulf) in faeces collected before EAE induction (START), during the phase with the maximum EAE score (PEAK, day 14–20, depending on the individual), and remission (REM, day 21–25 after EAE induction). **b**, Boxplot of faecal enzymatic activities before EAE induction (START). One-way ANOVA followed by Tukey's post-hoc test. Statistical differences indicated by compact letter display: two groups sharing an assigned letter, non-significant; two groups not sharing an assigned letter, p < 0.05. **c**, Boxplot of faecal enzyme activities of EAE-induced mice, based on cluster affiliation (Fig. 5e) and phase (START, PEAK, REM). Unpaired t-test; ns, non-significant. **d**, Pearson correlations of faecal enzymatic activities of EAE-induced mice before EAE induction with key EAE-associated readouts after EAE-induction in the same individuals. All correlations

non-significant. **e**, Faecal concentrations of lipocalin 2 (LCN2; 1 outlier removed from SM01 FR) and serum concentrations of lipopolysaccharide (LPS), occludin (OCLN), and zonulin (ZO-1; 1 outlier removed from SM04 FR and SM14 FR) in non-EAE-induced mice (−EAE) or EAE-induced mice at 30 d after EAE-induction (+EAE). One-way ANOVA. ns, non-significant. **f**, Pearson correlations between faecal LCN2 concentrations, serum OCLN concentrations, serum LPS concentrations and serum ZO-1 concentrations before induction of EAE with key EAE-associated readouts in the same individual across all tested microbiota–diet combinations. All correlations non-significant. **g**, Faecal concentrations of short-chain fatty acids (SCFA) in non-EAE-induced mice of certain microbiota–diet combinations. One-way ANOVA followed by a Tukey's post-hoc test. Statistical differences indicated by compact letter display: two groups sharing an assigned letter, p > 0.05; two groups not sharing an assigned letter, p < 0.05. Mouse numbers are indicated on the respective panels and treated as biological replicates. Boxplots (**b**, **c**, **e**, **g**) show median, quartiles, and 1.5 × interquartile range.

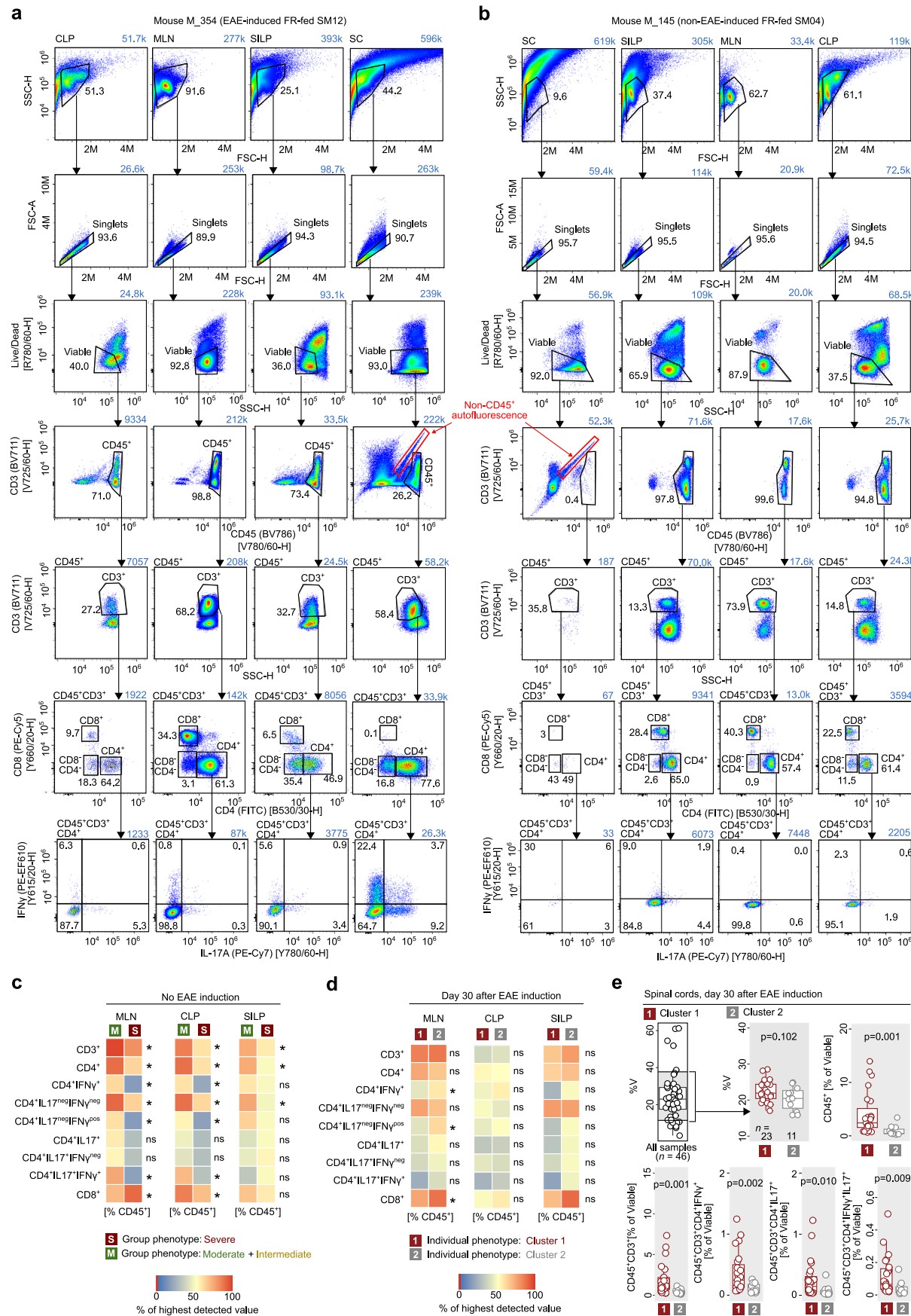

**Extended Data Fig. 7 | See next page for caption.**

**Extended Data Fig. 7 | Analysis of T-cell subsets in EAE-induced and non-EAE-induced mice harbouring reduced communities. a,b**, Gating strategy to determine IL-17a and IFNγ expression in T-helper cells of EAE-induced and non-EAE-induced mice. Axes labelled with forward or side light scatter (FSC, SSC), 'Live/Dead', or the detected antigen. The fluorochrome coupled to the antigen-specific antibody is indicated in round brackets, with the detection channel in square brackets. Number of detected events is in blue above each plot. Numbers inside the plots refer to the percentage of gated events compared to the parental population of the previous gating step. From the fourth gating step onwards, parental populations are indicated above the plots. Gates in the 'Live/Dead vs. SSC-H' depiction for mesenteric lymph nodes (MLN), colonic lamina propria (CLP) and small intestinal lamina propria (SILP), include a pre-gating step for CD45+ cells, which was verified through backgating. The representative EAE-induced (panel **a**) and non-EAE-induced mouse (panel **b**) highlight differences in CD45+ cell infiltration in the spinal cords (SC) during EAE. Due to autofluorescence of non-CD45+ cells in SC (highlighted in red), gates

set according to the CD45+ and CD3+ populations in other organs of the same individual. **c**, Proportions (% of CD45+ cells) of lymphocyte subsets in MLN, CLP, or SILP of non-EAE-induced mice. SC excluded due to insufficent CD45+ cell counts in non-EAE-induced mice. Mice grouped according to the group disease phenotype upon EAE induction (Fig. 5c). Means scaled by calculating the percentage of the highest observed value within the respective population–organ combination. ns, non-significant; *, p < 0.05 Unpaired t-test. **d**, Proportions (% of CD45+ cells) of lymphocyte subsets in MLN, CLP, or SILP of EAE-induced mice 30 d after EAE induction. EAE-induced mice assigned to Cluster 1 (strong EAE symptoms) or Cluster 2 (mild EAE symptoms; Fig. 5e). Scaling approach identical to panel **c**.: ns, non-significant; *, p < 0.05 Unpaired t-test. **e**, After filtering to address parental population bias (see Methods), proportions of EAE-associated lymphocyte populations as percentages of 'Viable' cells were compared for Cluster 1 and Cluster 2 mice. Unpaired t-test. Boxplots show median, quartiles, and 1.5 × interquartile range. All statistical tests were two-sided.

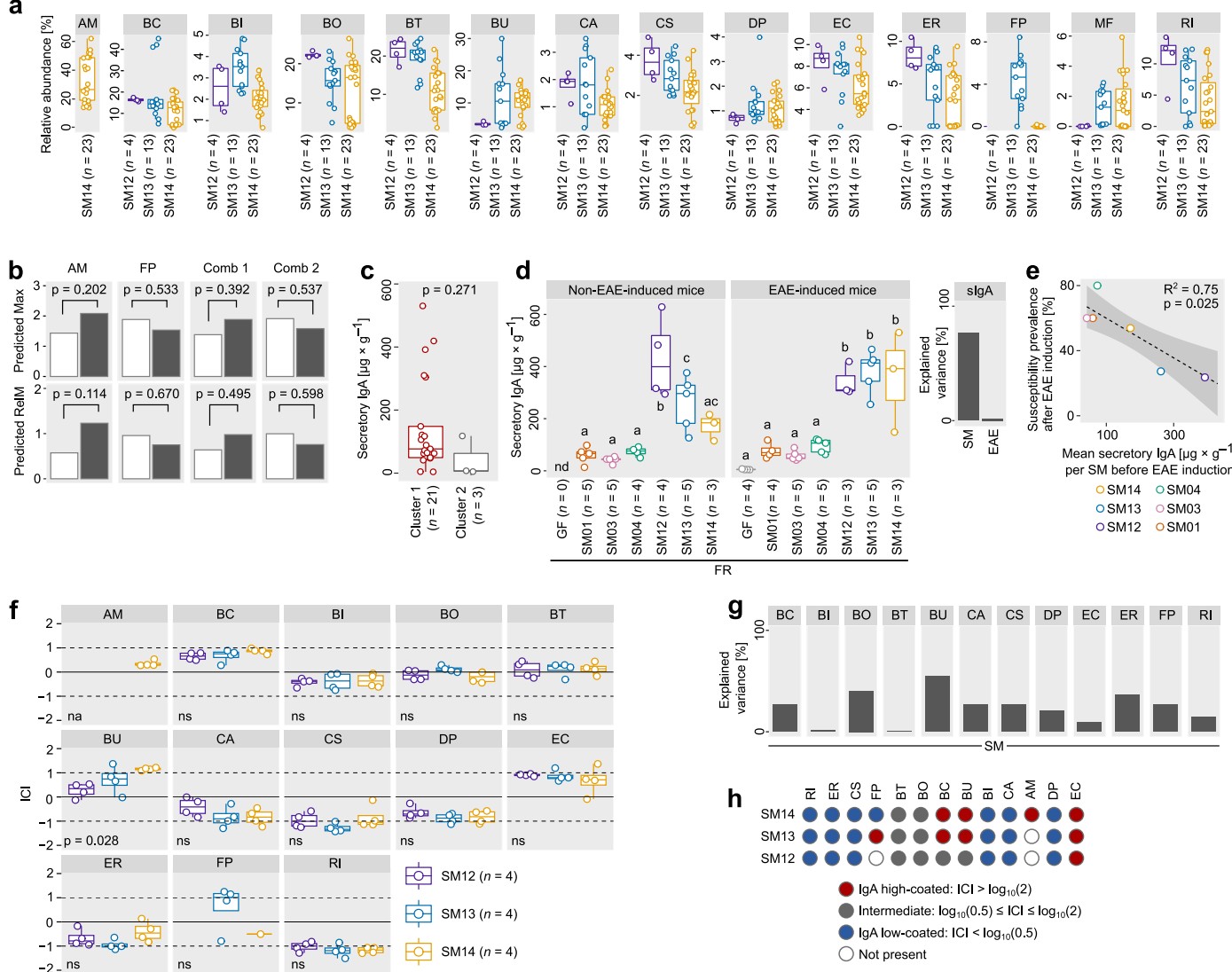

**Extended Data Fig. 8 | Microbiota-associated predictors for experimental autoimmune encephalomyelitis development. a**, Strain relative abundances of SM12-, SM13- and SM14-colonized mice, irrespective of diet, before induction of EAE. **b**, Linear mixed model regression for predicted maximum score during EAE (Max) and mean score during relapse phase (RelM) with presence of the strain as an independent variable and colonization as a random intercept effect. **c**, Concentrations of secretory IgA (sIgA) in faecal samples samples collected 30 d after EAE induction, normalized to faecal weight. Mice grouped by individual EAE phenotype (Fig. 5e). Due to lack of available material, sIgA could not be measured for some mice. Cluster 1, strong EAE symptoms; Cluster 2, mild EAE symptoms. Wilcoxon rank-sum test. **d**, Concentrations of faecal sIgA, normalized to faecal weight. Mice grouped by colonization–diet combination and EAE induction status (taken after 30 d of EAE for EAE-induced mice). Kruskal-Wallis test followed by Wilcoxon rank-sum test. Statistical differences are indicated by compact letter display: two groups sharing an assigned letter, non-significant; two groups not sharing an assigned letter, p < 0.05. Right panel, variance of sIgA concentrations

explained by either SM or diet. nd, not done. **e**, Correlation between disease susceptibility among EAE-induced mice harbouring a certain SM with mean faecal sIgA levels in non-EAE-induced mice harbouring the same SM by linear regression. Each dot represents one SM combination. **f**, IgA coating index (ICI) of each strain dependent on SM. One-way ANOVA. See Methods for details on ICI calculation. ICIs could not be calculated for *M. formatexigens* as the relative abundance was below the limit of detection for at least one of the sorted fractions across all samples (also applies for panels **g** and **h**). na, not applicable; ns, non-significant. **g**, Variance in IgA coating index (ICI) explained by background microbiota (SM12, SM13 and SM14; FR only) by eta-squared ($\eta^2$) calculation. Variances could not be calculated for *A. muciniphila*, as it was only present in one microbiota combination. **h**, Classification of SM14-constituent strains as IgA high-coated, intermediate, or low-coated, depending on microbiota composition. Boxplots (**a**, **c**, **d**, **f**) show median, quartiles, and 1.5 × interquartile range.

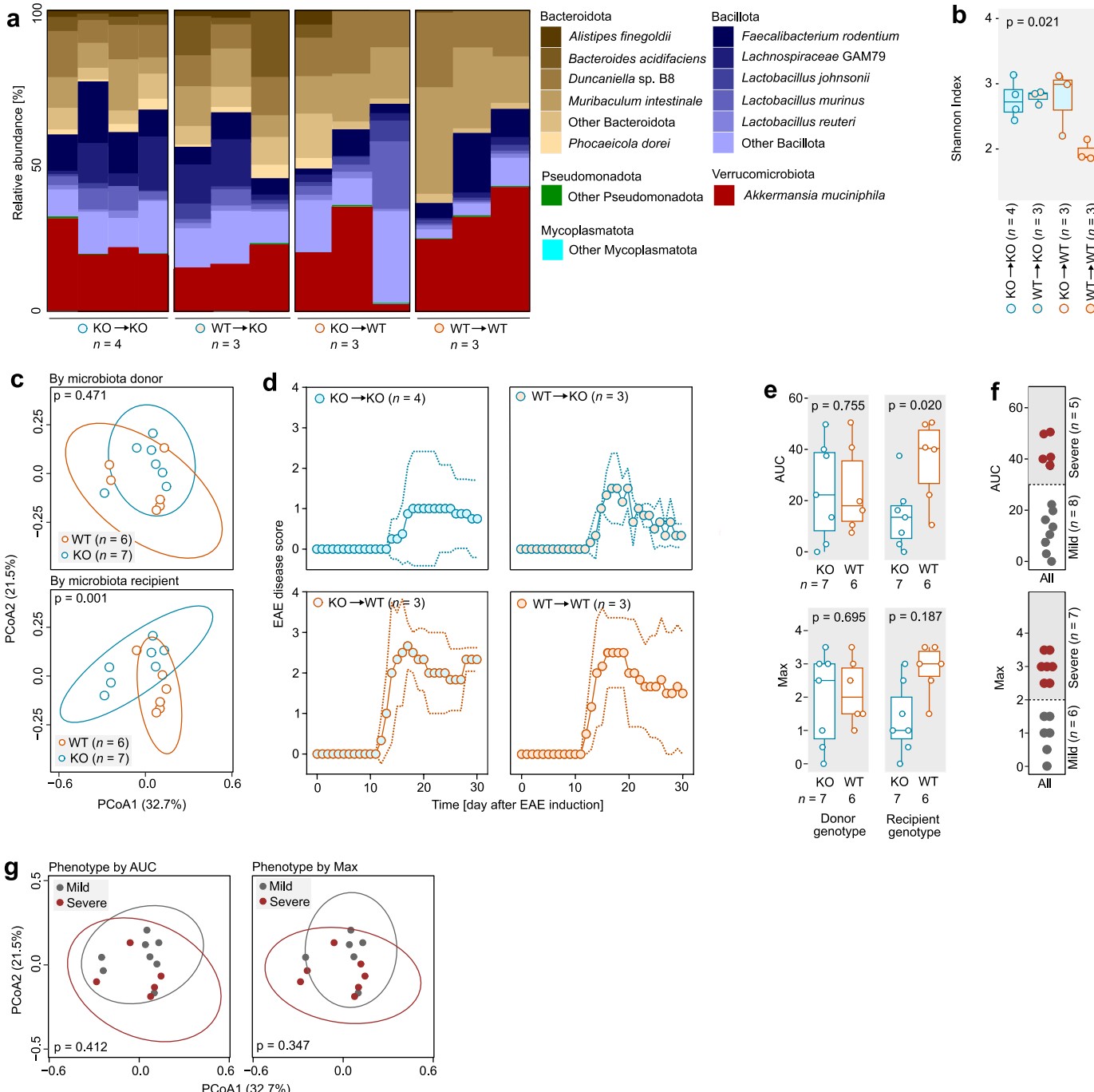

**Extended Data Fig. 9 | Experimental autoimmune encephalomyelitis in mice harbouring cross-genotype transferred complex microbiotas.** Microbiota from SPF-housed *Muc2*[−/−] (KO) and *Muc2*[+/+] (WT) mice was transferred to germ-free (GF) C57BL/6J and *Muc2*[−/−] mice (Fig. 6a). EAE was induced after 21 d exposure to the donor microbiota. **a**, Faecal microbiota composition in EAE-induced mice on the day of EAE induction (d 0; 'pre-EAE'), ordered by donor–recipient combination. Microbiota composition was determined on a species level by full-length 16S rRNA gene sequencing. The top 5 most abundant species per phylum are shown. In the case of Verrucomicrobiota, *Akkermansia muciniphila* was the only detected feature. **b**, Shannon index as a measure of α-diversity of the pre-EAE microbiota compositions. One-way ANOVA with donor–recipient combinations as tested variable. **c**, β-Diversity of pre-EAE microbiota compositions from a Bray–Curtis distance matrix of arcsine square root-transformed relative abundance data, ordinated using a principal coordinate analysis. Upper panel, mice coloured by genotype of the microbiota donor. Lower panel, mice coloured by genotype of the microbiota recipient. PERMANOVA

using the genotype of either donor (upper panel) or recipient (lower panel) as tested variable. **d**, EAE disease course of mice (same mice as in panel **a**), separated by donor–recipient combinations, visualized as EAE disease score as a function of time (day after EAE induction). Dots represents daily mean EAE score; dashed lines represent SD. **e**, Evaluation of EAE disease course using AUC of the individual disease course (panel **d**) or the maximum achieved EAE disease score per mouse (Max). Left, mice coloured by donor genotype. Right, mice coloured by recipient genotype. Wilcoxon rank-sum test with donor (left) or recipient (right) genotype as tested variable. Boxplots show median, quartiles, and 1.5 × interquartile range. **f**, Distribution of individual AUC and Max values to determine 'Severe' and 'Mild' EAE; accordingly, AUC = 30 and Max = 2 were selected as thresholds to establish a binary categorization of individual EAE phenotypes. **g**, Individual pre-EAE microbiota composition. Same underlying Bray–Curtis distance matrix and PCoA-ordination as panel **c**, coloured by binary phenotype classification (panel **f**) based on individual AUC (left) or Max (right). PERMANOVA with the indicated phenotype classification as tested variable.

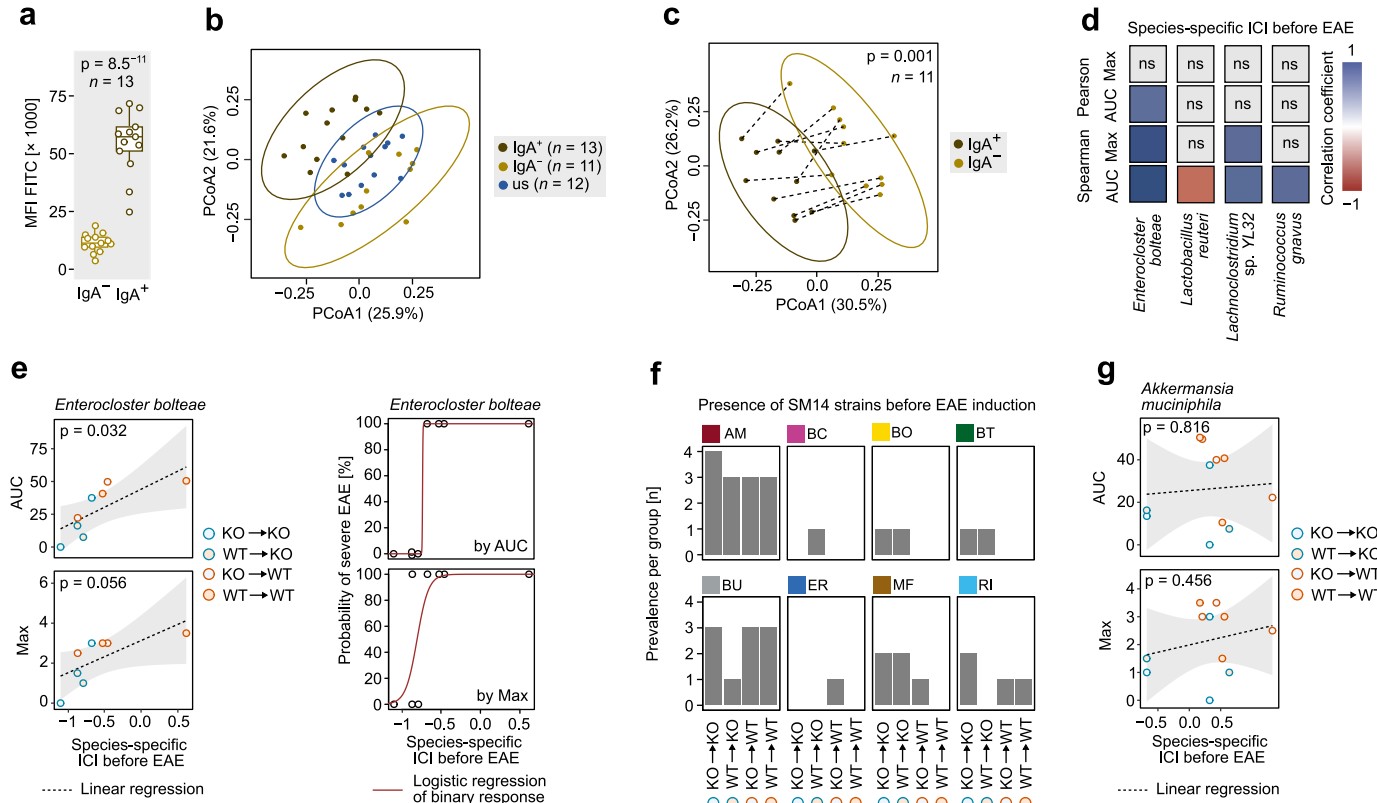

**Extended Data Fig. 10 | Evaluation of species-specific IgA coating indices as individual disease course predictors in mice with complex microbiotas. a–c**, Separation efficiency for IgA-coated (IgA⁺) and non-IgA-coated (IgA⁻) bacteria in faeces collected on the day of EAE induction ('pre-EAE'). **a**, IgA⁺ and IgA⁻ fractions obtained from the same faecal sample were stained with a FITC-coupled anti-mouse IgA antibody and mean fluorescence intensity (MFI) was determined by flow cytometry. Unpaired t-test. Boxplots show median, quartiles, and 1.5 × interquartile range. **b**, β-Diversity of the microbiota composition of three distinct samples (IgA⁺, IgA⁻, and unsorted) from a Bray–Curtis distance matrix obtained from arcsine square root transformed relative abundance data on a species level, determined by full-length 16S rRNA gene sequencing. Compositions of unsorted fractions are shown in Extended Data Fig. 9a. Samples of insufficient depth (*n* = 3) were excluded. **c**, β-Diversity of IgA⁺ and IgA⁻ fractions from a re-calculated Bray–Curtis distance matrix, after removal of unsorted samples from the analysis. Dashed lines connect IgA⁺ and IgA⁻ fractions from the same sample. PERMANOVA

using the post-separation fraction as tested variable based on the re-calculated Bray–Curtis distance matrix. **d**, Individual Pearson and Spearman correlation of key EAE associated readouts (AUC, Max) with pre-EAE ICIs of species which provided at least one significant correlation. Significant correlations are shown in shades of blue or red. ns, non-significant correlation. See Methods for details on ICI calculation. **e**, Left, Pearson correlation of *Enterocloster bolteae* ICI with AUC (upper panel) and Max (lower panel) by linear regression. Right, binomial regression model to predict probability of severe disease, as defined in panel **f**, based on AUC (top) or Max (bottom) using *E. bolteae* ICI. Only those mice are shown where *E. bolteae* was detectable in both fractions after sorting (n = 8). Dots were jittered in case of overlap (upper left panel). **f**, Prevalence of SM14 species in unsorted samples at the day of EAE induction. Species were considered present when relative abundance was > 0.01%. For species abbreviations and colour coding, see Fig. 5g. **g**, Pearson correlation of *Akkermansia muciniphila* ICI with AUC (top) and Max (bottom) by linear regression.

# Reporting Summary

## Statistics

For all statistical analyses, confirm that the following items are present in the figure legend, table legend, main text, or Methods section.

| n/a | Confirmed | |
|---|---|---|
| ☐ | ☒ | The exact sample size (*n*) for each experimental group/condition, given as a discrete number and unit of measurement |
| ☐ | ☒ | A statement on whether measurements were taken from distinct samples or whether the same sample was measured repeatedly |
| ☐ | ☒ | The statistical test(s) used AND whether they are one- or two-sided<br>*Only common tests should be described solely by name; describe more complex techniques in the Methods section.* |
| ☐ | ☒ | A description of all covariates tested |
| ☐ | ☒ | A description of any assumptions or corrections, such as tests of normality and adjustment for multiple comparisons |
| ☐ | ☒ | A full description of the statistical parameters including central tendency (e.g. means) or other basic estimates (e.g. regression coefficient) AND variation (e.g. standard deviation) or associated estimates of uncertainty (e.g. confidence intervals) |
| ☐ | ☒ | For null hypothesis testing, the test statistic (e.g. *F*, *t*, *r*) with confidence intervals, effect sizes, degrees of freedom and *P* value noted<br>*Give P values as exact values whenever suitable.* |
| ☒ | ☐ | For Bayesian analysis, information on the choice of priors and Markov chain Monte Carlo settings |
| ☒ | ☐ | For hierarchical and complex designs, identification of the appropriate level for tests and full reporting of outcomes |
| ☐ | ☒ | Estimates of effect sizes (e.g. Cohen's *d*, Pearson's *r*), indicating how they were calculated |

*Our web collection on statistics for biologists contains articles on many of the points above.*

## Software and code

Policy information about availability of computer code

| Data collection | No custom software was used for data collection. |
|---|---|
| Data analysis | No custom algorithms or software were used for data analysis. The analysis pipelines in this study employ tools that have been previously described in published literature and are described below.<br><br>All figures and graphs were created using R Studio (Version 4.2.1) and the package ggplot2 (Version 3.3.6). Subsequent figure modifications were performed using Inkscape.<br><br>All data analyses and statistical computations were performed using R Studio (Version 4.2.1) using existing packages and functions. Correlations were calculated using the rcorr package (Version 0.4.4). Linear and nonlinear regressions were calculated using the drc package (Version 3.0.1). One-way ANOVA, Kruskal-Wallis test, t-tests and Wilcoxon rank-sum-tests were calculated using rstatix package (Version 0.7.0). Normality distribution of data sets before statistical computation was determined by applying a Shapiro-Wilk test. Outliers were determined using the outlier package (Version 0.15) and, importantly, confirmed using a Dixon test.<br><br>Construction of the phylogenetic tree of SM14 constituent strains<br>The phylogenetic tree was constructed based on full-length 16S rRNA gene sequences and analysed with Geneious Prime version 2021.2.2. A neighbour-joining tree build model was created using global alignment with free end and gaps, 65% similarity index cost matrix, and a Tamura-Nei genetic distance model.<br><br>V4 16S rRNA gene sequencing analysis:<br>The program mothur (v1.44.3) (DOI: https://doi.org/10.1128/AEM.01541-09) was used to process the reads according to the MiSeq SOP. For |

gnotobiotic samples, taxonomy was assigned using a k-nearest neighbor consensus approach against a custom reference database corresponding to the SM14 taxa and potential contaminants (Citrobacter rodentium, Lactococcus lactis subsp. cremoris, Staphylococcus aureus, and Staphylococcus epidermidis). For SPF samples, taxonomy was assigned using the Wang approach against the SILVA v132 database. Count data was normalized by computing relative abundance. Diversity indices were determined using the diversity() function of the vegan package. Nonmetric multidimensional scaling for Bray-Curtis distance matrices were calculated using the metaMDS() function of the vegan package and principal coordinate decomposition of Weighted UniFrac distance matrices were calculated using the pcoa() function of the ape package (version 5.6.2). All analyses were performed on OTU level, genus level and family level. OTUs and genera contributing most to community differences between selected groups were extracted using the simper() function of the vegan package. Groupwise analysis of annotated reads was performed using RStudio (version 4.2.1) with an initial seed set at 8765. All operational taxonomic units (OTUs) not constituting at least 0.1% of reads within at least one group (group means) were removed from the analysis. Diversity indices were determined using the diversity() function of the vegan package (version 2.6.2). Nonmetric multidimensional scaling for Bray-Curtis distance matrices were calculated using the metaMDS() function of the vegan package and principal coordinate decomposition of weighted UniFrac distance matrices were calculated the pcoa() function of the ape package (version 5.6.2). All analyses were performed on OTU level, genus level and family level. OTUs and genera contributing most to community differences between selected groups were extracted using the simper() function of the vegan package.

Full-length 16S rRNA gene sequencing analysis:
Raw fast5 data files were converted to pod5 format using POD5 Tools (v0.2.0), then basecalled in super-accuracy mode and demultiplexed according to the barcodes of the SQK-NBD114-96 kit on a gpu partition using Dorado (v0.4.3). The basecalled and demultiplexed bam files for each sample have been uploaded to the European Nucleotide Archive (ENA) at EMBL-EBI under the study accession number PRJEB60278. Bam files were converted to fastq format using samtools (v1.16.1) bam2fastq, then filtered using NanoFilt (v2.8.0) such that only Phred quality scores above 10 and read lengths between 1,300 to 1,700 bp were retained. The taxonomic classification was carried out emu using emu (v3.4.5) with the --keep-counts flag and the default Emu 3.0+ database. Completely unassigned reads were removed from downstream analyses. The rarecurve() function from the vegan package (v2.6-4) was used to identify undersampled samples, which were also removed from downstream analyses. Count tables of the remaining samples were used to determine alpha-diversity using the phyloseq package (v1.40.4). To calculate beta-diversity and to perform further downstream analyses, only those detected bacterial features were included which provided more than 0.01% relative abundance in at least two mice within each group of microbiota-donor and microbiota-recipient combination. Beta-diversity measures were determined using the phyloseq package (v1.40.0) after read counts were normalized by calculating relative abundances followed by arcsine square root transformation. Relative abundances per taxonomic unit and sample were visualized using the fantaxtic package (v0.2.0).

ELISA analyses:
Concentrations were calculated based on detected optical densities (OD) of supplied standards and using R Studio (version 4.2.1) applying a 4-parameter nonlinear regression of standard ODs with the help of the drc package (version 3.0.1) using the function drm(OD~concentration, fct=LL.4()). Sample concentrations were then extracted using the ED(type="absolute") function of the same package.

Metatranscriptomics analyses:
RNA sequencing files were pre-processed using kneaddata (https://github.com/biobakery/kneaddata). Adapters were removed using Trimmomatic and fragments below 50% of the total expected read length (75 bp) were filtered out. BowTie2 was used to map and remove contaminant reads corresponding to either rRNA databases or the Mus musculus genome. Clean fastq files were concatenated before passing to HUMAnN3. A custom taxonomy table based on pooled 16S rRNA sequencing abundances was provided to MetaPhlAn (v30_CHOCOPhlAn_201901) for metagenome mapping. Reads were aligned within HUMAnN3 using bowtie2 (version 2.3). Unaligned reads were translated within HUMAnN3 using diamond (version 0.9.36) for protein identification using the UniRef90 database provided within HUMAnN3. Data for all samples were joined into a single table and normalized using count per million (CPM) method. Results were grouped by annotated protein product per individual. In case no annotation from UniRef90 transcript IDs was possible, distinct IDs were treated as separate gene products. Only gene products that provided >50 CPM in at least two of the eight investigated samples were included into downstream analyses. This resulted in 2213 transcripts being included into downstream analyses, representing 80% to 85% of the total CPM with no significant differences between the analyzed groups. CPM were recalculated to account for removed transcripts, followed by further analysis using the edgeR package (version 3.38.4) in R Studio (version 4.2.1). Multidimensional reduction of the transcriptome profiles was calculated using the logFC method within the plotMDS.DGEList function. Groupwise comparison of gene expression was calculated using the exactTest() function.

Metabolomics:
Metabolome spectra and concentration calculations were carried out using the software "MasterHands" version 2.19.0.2 (Keio University) as previously described (DOI: 10.1007/s11306-009-0178-y). Obtained cecal metabolite concentrations were converted from nmol × g−1 cecal content to pmol × g−1. Only the 175 cecal metabolites that were detectable in at least 50% of the samples within one of the seven tested groups were included in downstream analyses. Undetectable concentrations of these 175 metabolites in certain samples were replaced by a value of 1/3 of the lowest detectable value of this particular metabolite across all groups. Concentrations were then normalized using a log2-transformation for further downstream analysis.

## Data

Policy information about availability of data

All manuscripts must include a data availability statement. This statement should provide the following information, where applicable:

- Accession codes, unique identifiers, or web links for publicly available datasets
- A description of any restrictions on data availability
- For clinical datasets or third party data, please ensure that the statement adheres to our policy

The raw files for this study from 16S rRNA gene sequencing (V4 and full-length amplicons) and RNA sequencing have been deposited in the European Nucleotide Archive (ENA) at EMBL-EBI under accession number PRJEB60278 (https://www.ebi.ac.uk/ena/browser/view/PRJEB60278). Raw spectral data from CE-TOF/MS metabolomic analyses are available on the MetaboBank integrated metabolome data repository under Project ID MTBKS223 (https://mb2.ddbj.nig.ac.jp/study/MTBKS223.html). For the construction of the phylogenetic tree of SM14 constituent strains, 16S rRNA gene sequences were downloaded from the ENA (https://www.ebi.ac.uk/ena/browser/home) using the following accession numbers: AY271254.1 (A. muciniphila), AB510697.1 (B. caccae), EU136682.1 (B. ovatus), HQ012026.1 (B. thetaiotaomicron), AB050110.1 (B. uniformis), AB370251.1 (B. intestinihominis), AB626630.1 (E. rectale), HM245954.1 (C. symbiosum), AJ505973.1 (M. formatexigens), AF192152.1 (D. piger), AB011816.1 (C. aerofaciens), AJ413954.1 (F. prausnitzii), AJ312385.1 (R. intestinalis), and CP000802.1:4065789..4067330:rRNA (E. coli). The Emu database used for full-length 16S rRNA gene mapping combines rrnDB v5.6 and NCBI 16S RefSeq from 17 September, 2020, and taxonomy from NCBI on the same date. Flow cytometry data is available on Zenodo (https://zenodo.org/records/12528901), under doi: 10.5281/zenodo.12528901.

## Research involving human participants, their data, or biological material

Policy information about studies with human participants or human data. See also policy information about sex, gender (identity/presentation), and sexual orientation and race, ethnicity and racism.

| | |
|---|---|
| Reporting on sex and gender | N/A |
| Reporting on race, ethnicity, or other socially relevant groupings | N/A |
| Population characteristics | N/A |
| Recruitment | N/A |
| Ethics oversight | N/A |

Note that full information on the approval of the study protocol must also be provided in the manuscript.

# Field-specific reporting

Please select the one below that is the best fit for your research. If you are not sure, read the appropriate sections before making your selection.

☒ Life sciences      ☐ Behavioural & social sciences      ☐ Ecological, evolutionary & environmental sciences

For a reference copy of the document with all sections, see nature.com/documents/nr-reporting-summary-flat.pdf

# Life sciences study design

All studies must disclose on these points even when the disclosure is negative.

| | |
|---|---|
| Sample size | We determined the sample sizes presented in the manuscript to be statistically sufficient to support the conclusions. See details on statistical calculation in the manuscript an in this report. |
| Data exclusions | For FACS analysis, low quality samples were excluded from analysis. "Low quality" was defined as "less than 1000 events after gating on CD45+ cells".  Additionally, outliers were determined by group and values not within a range of mean ± 2σ were removed from the datasets.<br>For EAE-associated readouts, no data were excluded.<br>For groupwise comparisons of ELISA readouts, apparent outliers were identified using the outlier() function in R and confirmed by a Dixon-Test before being removed from the analysis.<br>In 16S rRNA gene sequencing analyses of SPF-housed mice, taxa not providing a mean relative abundance of 0.01% within a given group were excluded from downstream analysis. For RNA sequencing data, host RNA, ribosomal RNA and viral RNA were excluded from analyses. For metabolite analysis, metabolites not detected in at least 50% of the samples in at least 1 group were excluded from downstream analyses. |
| Replication | We performed EAE experiments in SPF-housed CR mice in 2 different runs. EAE experiments using SM14- and SM13-colonized mice were performed in 3 different runs (all attempts at replication were successful). As we observed the same internal group variances within each run, we determined that follow-up EAE experiments in other mice did not require multiple runs to cover the whole possible internal variance. |
| Randomization | Mice were assigned randomly to each group. After randomized group allocation, we checked whether the mean weight of each mouse group does not differ significantly from each other. |

| Blinding | Blinding was performed wherever possible. FACS analysis of isolated lymphocytes was performed blinded and the gating strategy was verified by at least two persons. EAE scoring, due to the nature of the experiment (e.g. visibly different diets, handling of mice from low to high SM to prevent contamination), complete blinding was impossible, however the EAE scoring was performed by two independent researchers (Alex Steimle and Mareike Neumann) to prevent bias. For CE-TOF/MS data generation, the researchers (Shinji Fukuda and Tomoyoshi Soga) were blinded for the identities of the experimental groups. For measurement of short-chain fatty acids from cecal samples, researchers from the responsible metabolomics facility were blinded for the identities of the experimental groups. For other group allocations during data collection and/or analysis, blinding was not possible or not needed for the following reasons: sequencing data (16S rRNA gene based and metatranscriptomics), blinding not needed as the same analysis pipeline was used and hence no bias expected; analysis of data from EAE scoring, blinding not needed as defined analysis criteria were employed and hence no bias expected; metabolomics data analysis, blinding not needed as defined analysis criteria were employed and hence no bias expected; bacterial IgA coating, blinding not needed as defined experimental protocols were employed and hence no bias expected; bacterial enzyme activities in stool samples, blinding not needed as defined experimental protocols were employed and hence no bias expected; ELISAs for LCN2, LPS, OCLN and ZO-1, blinding not needed as defined experimental protocols were employed and hence no bias expected. |
|---|---|

# Reporting for specific materials, systems and methods

We require information from authors about some types of materials, experimental systems and methods used in many studies. Here, indicate whether each material, system or method listed is relevant to your study. If you are not sure if a list item applies to your research, read the appropriate section before selecting a response.

## Materials & experimental systems

| n/a | Involved in the study |
|---|---|
| ☐ | ☒ Antibodies |
| ☒ | ☐ Eukaryotic cell lines |
| ☒ | ☐ Palaeontology and archaeology |
| ☐ | ☒ Animals and other organisms |
| ☒ | ☐ Clinical data |
| ☒ | ☐ Dual use research of concern |
| ☒ | ☐ Plants |

## Methods

| n/a | Involved in the study |
|---|---|
| ☒ | ☐ ChIP-seq |
| ☐ | ☒ Flow cytometry |
| ☒ | ☐ MRI-based neuroimaging |

## Antibodies

| Antibodies used | CD16/CD32, BD, #553142<br>rat anti-mouse IL-17A (TC11-18H10.1, 1/50, Biolegend, #506922)<br>rat anti-mouse RORγT (AFKJS-9, 1/44, eBioscience, #17-6988-82)<br>rat anti-mouse CD3 (17A2, 1/88, Biolegend, #100241)<br>rat anti-mouse CD45 (30-F11, 1/88, BD, #564225)<br>rat anti-mouse CD4 (RM4-5, 1/700, Biolegend, #100548)<br>rat anti-mouse IFN-γ (XMG1.2, 1/175, eBioscience, #61-7311-82)<br>rat anti-mouse Foxp3 (FJK-16s, 1/200, eBioscience, #48-5773-82)<br>rat anti-mouse CD8 (53-6.7, 1/700, Biolegend, #100710)<br>rat anti-mouse Siglec-F (E50-2440, BD, #740956)<br>rat anti-mouse IgA (mA-6E1, 1/700, eBioscience, #11-4204-83) |
|---|---|
| Validation | Optimal concentrations for cell staining to exclude unspecific binding were determined in our lab using spleen cells. Antibody dilutions are indicated in the box above.<br>Faecal samples from Rag1−/− mice were used as non-IgA-coated negative controls in the determination of bacterial IgA coating indices. |

## Animals and other research organisms

Policy information about studies involving animals; ARRIVE guidelines recommended for reporting animal research, and Sex and Gender in Research

| Laboratory animals | For gnotobiotic experiments, female germ-free (GF) C57BL/6N mice were used at the age of 5 to 8 weeks. The mice were originally purchased from Taconic Biosciences (USA) and were subsequently bred and housed in the GF facility of the University of Luxembourg, supervised by the Animal Experimentation Ethics Committee of the University of Luxembourg (AEEC).<br>For experiments performed under specific-pathogen-free (SPF) conditions, we used female mice of different origin. C57BL/6J wildtype mice were purchased from Charles River at the age of 5 to 8 weeks. Furthermore, we used mice lacking the Muc2 gene (strain designation: 129P2/OlaHsd×C57BL/6-Muc2<tm1Avel>), which were originally obtained from the lab of Kathy McCoy (University of Bern, Switzerland) under GF conditions. GF 129P2/OlaHsd×C57BL/6-Muc2<tm1Avel> mice were mated with SPF-housed C57BL/6J mice obtained from Charles River resulting in offspring heterozygous for presence of the Muc2 gene (Muc2+/−). Muc2+/− mice were constantly kept under the same SPF conditions as the SPF-housed parental C57BL/6J mice. Next, male and female Muc2+/− mice were mated and offspring were genotyped for absence and presence of the Muc2 gene. Homozygous Muc2−/− and Muc2+/+ mice obtained from this breeding were then used for experiments. Finally, faecal samples from SPF Rag1−/− mice (C57BL6/J background) at the Luxembourg Institute of Health were used as negative controls in IgA coating experiments.<br>Mice were housed in individually ventilated cages (Techniplast Sealsafe Plus GM500 or Allentown Sentry SPP™ Mouse cages in SPF or |
|---|---|

GF facilities, respectively) at 20-24°C with 40-70% humidity, under 12-hour light cycles. Sterile water and diets were provided ad libitum.

| | |
|---|---|
| Wild animals | No wild animals were used in the study. |
| Reporting on sex | The data reported in the manuscript was collected from female mice. We conducted preliminary experimental autoimmune encephalomyelitis (EAE) experiments on male gnotobiotic mice and found that male mice exhibited stronger EAE symptoms and could not be continued for the entire duration of the experiment owing to ethical constraints.<br>Generally, female individuals are more prone to the development of multiple sclerosis than male individuals, even though the progression of the disease is generally milder in females (DOI: https://doi.org/10.1177/1756285613488434). This is mainly due to the protective effect of female hormones and has been shown in mouse models of EAE (DOI: https://doi.org/10.1177/107385840100700310). Furthermore, male mice are more likely to be involved in territorial fights, which might influence the outcome of the experiments due to injuries. Due to these scientific facts and the results from the already performed experiments, we used only female mice. |
| Field-collected samples | No field collected samples were used in the study. |
| Ethics oversight | All mouse experiments followed a two-step animal protocol approval procedure. Protocols were first evaluated and pre-approved by either the Animal Experimentation Ethics Committee (AEEC) of the University of Luxembourg or the Animal Welfare System (AWS) of the Luxembourg Institute of Health, followed by final approval by the Luxembourgish Ministry of Agriculture, Viticulture, and Rural Development (Protocol numbers: LUPA2020/02, LUPA2020/27, LUPA2020/32, LUPA2019/43, LUPA2020/22, LUPA2019/51). All experiments were performed according to the Federation of European Laboratory Animal Science Association (FELASA). The study was conducted according to the "Règlement grand-ducal du 11 Janvier 2013 relatif à la protection des Animaux utilisés à des fins Scientifiques" based on the "Directive 2010/63/EU" of the European Parliament and the European Council from September 2010 on the protection of animals used for scientific purposes. All animals were exposed to 12 hours of light daily. |

Note that full information on the approval of the study protocol must also be provided in the manuscript.

# Flow Cytometry

## Plots

Confirm that:

☒ The axis labels state the marker and fluorochrome used (e.g. CD4-FITC).

☒ The axis scales are clearly visible. Include numbers along axes only for bottom left plot of group (a 'group' is an analysis of identical markers).

☒ All plots are contour plots with outliers or pseudocolor plots.

☒ A numerical value for number of cells or percentage (with statistics) is provided.

## Methodology

| | |
|---|---|
| Sample preparation | Lymphocyte extraction from colonic lamina propria, ileal lamina propria and spinal cords:<br>After organ removal, lymphocytes from the colonic lamina propria (CLP), small intestine lamina propria (SILP) and spinal cords (SC) were extracted. While CLP and SILP lymphocytes were extracted using the lamina propria dissociation kit (Miltenyi Biotec, #130-097-410), SC lymphocytes were extracted using a brain dissociation kit (Miltenyi Biotec, #130-107-677), according to the manufacturer's instructions. In brief, colon and ileum were dissected and stored in HBSS (w/o). Faeces and fat tissue were removed, organs were opened longitudinally, washed in HBSS (w/o), and cut laterally into 0.5 cm-long pieces. Tissue pieces were transferred into 20 mL of a predigestion solution (HBSS (w/o), 5 mM EDTA, 5% foetal bovine serum (FBS), 1 mM dithiothreitol) and kept for 20 min at 37 °C under continuous rotation. Samples were then vortexed for 10 sec and applied on a 100 μm cell strainer. Last two steps were repeated once. Tissue pieces were then transferred into HBSS (w/o) and kept for 20 min at 37 °C under continuous rotation. After vortexing for 10 sec, tissue pieces were applied on a 100 μm cell strainer. Tissue pieces were then transferred to a GentleMACS C Tube (Miltenyi Biotec, #130-093-237) containing 2.35 mL of a digestion solution and homogenized on a GentleMACS Octo Dissociator (Miltenyi Biotec, #130-096-427, program 37C_m_LPDK_1). Homogenates were resuspended in 5 mL PB Buffer (phosphate-buffered saline (PBS), pH 7.2, with 0.5% bovine serum albumin), passed through a 70 μm cell strainer and centrifuged at 300 × g for 10 min at 4 °C. Cell pellets were resuspended in ice-cold PB buffer and stored on ice until further use. Spinal cords were stored in ice-cold D-PBS until they were transferred to a GentleMACS C Tube containing a digestion solution. Samples were processed on a GentleMACS Octo Dissociator (program 37C_ABDK_01) and rinsed through a 70 μm cell strainer. The cell suspension flow through was then centrifuged at 300 × g for 10 min at 4 °C. Debris removal was performed by resuspending the cell pellet in 1550 μL D-PBS, adding 450 μL of Debris Removal Solution, and overlaying with 2 mL of D-PBS. Samples were centrifuged at 4 °C at 3,000 × g for 10 min. The two top phases were aspirated and the cell suspension was diluted with cold D-PBS. Samples were then inverted three times and centrifuged at 4 °C, 1,000 × g for 10 min and the cell were resuspended and stored in ice-cold D-PBS until further use.<br><br>Cell stimulation and flow cytometry:<br>10^6 cells (MLN cell suspensions as well as lymphocyte extracts from CLP, SILP and SC) were resuspended in 1 mL complete cell culture medium (RPMI containing 10% FBS, 2 mM glutamine, 50 U × mL-1 penicillin, 50 μg × mL-1 streptomycin, and 0.1% mercaptoethanol) supplemented with 2 μL Cell Activation Cocktail with Brefeldin A (Biolegend, #423304) and incubated for 4 h at 37 °C. Cells were centrifuged at 500 × g for 5 min, resuspended in 100 μL Zombie NIR (1:1000 in PBS, Zombie NIR™ Fixable Viability Kit, Biolegend, #423106), transferred into a 96-well plate and incubated for 20 min at 4 °C in the dark. Cells were washed two times with 150 μL PBS (centrifuged 5 min at 400 × g at 4 °C) and resuspended in 50 μL Fc-block (1:50, Purified Rat Anti-Mouse CD16/CD32, BD, #553142) diluted in FACS buffer (1x PBS/2% FBS/2mM, EDTA pH 8.0). Cells were |

incubated for 20 min at 4 °C in the dark and washed two times with 150 μL PBS with centrifugation for 5 min at 400 × g at 4 °C. All cells were fixed for 30 min with BD Cytofix/Cytoperm solution (BD, #554722) and stored in PBS overnight. For the extracellular and intracellular fluorescent cell staining, cells were permeabilized with BD Perm/Wash buffer (BD, #554723) for 15 min. T lymphocytes were evaluated using the following antibodies: rat anti-mouse IL-17A (TC11-18H10.1, 1/50; Biolegend, #506922), rat anti-mouse RORγt (AFKJS-9, 1/44, eBioscience, #17-6988-82), rat anti-mouse CD3 (17A2, 1/88, Biolegend, #100241), rat anti-mouse CD45 (30-F11, 1/88, BD, #564225), rat anti-mouse CD4 (RM4-5, 1/700, Biolegend, #100548), rat anti-mouse IFN-γ (XMG1.2, 1/175, eBioscience, #61-7311-82), rat anti-mouse FOXP3 (FJK-16s, 1/200; ThermoFisher, #48-5773-82), rat anti-mouse CD8 (53-6.7, 1/700, Biolegend, #100710). Optimal staining concentrations of all antibodies were evaluated before staining. Cells were incubated with FACS buffer diluted antibodies for 30 min at 4 °C in the dark. Samples were washed twice with 150 μL of BD Perm/Wash buffer, resuspended in 200 μL PBS and acquired using NovoCyte Quanteon (NovoCyte Quanteon 4025, Agilent). All acquired data were analysed using FlowJo™ Software (version 10.7.2, BD, 2019). Fluorescence minus one controls (FMOs) were used for each antibody-fluorophore combination to properly evaluate signal-positive and -negative cells. Single antibody-stained UltraComp eBeads™ Compensation Beads (Fisher Scientific, Ref: 01-2222-42) were used to create the compensation matrix in FlowJo. Single antibody-stained compensation beads were created for each run separately, using the exact same antibody lot that were used for the samples. As we determined insufficient binding of the BV786-coupled rat anti-mouse CD45 antibody (30-F11, 1/88, BD, #564225) to the compensation beads, we used the BV786-coupled anti-mouse Siglec-F antibody (E50-2440, BD, #740956) to calculate the compensation matrix. Compensation samples were gated on the population of compensation beads within the FSC-H and SSC-H channels and the positive and negative population for the corresponding antibody were identified in a two-dimensional depiction of channels with strong fluorescence spillover. Compensation matrices were calculated for each run separately and applied to the samples of this particular run. After applying the compensation matrix, samples underwent the gating strategy, which is explained in detail in Extended Data Figure 7a and b. FACS analysis of isolated lymphocytes was performed blinded and the gating strategy was verified by at least two persons. Sample quality was evaluated by assessing the event distribution in the "SSC-H vs. FSC-H" and the "Live/Dead vs SSC-H" depiction and samples of insufficient quality and event counts were removed from the analysis. Downstream analysis of relative proportions of target cell populations was performed using RStudio (Version 4.2.1). Individual samples were grouped by target cell population, organ and EAE group phenotype (for non-EAE-induced mice) or by target cell population, organ and individual EAE severity cluster (for EAE-induced mice). Outliers were determined by group and values not within a range of mean ± 2σ were removed from the datasets.

| | |
|---|---|
| Instrument | Quanteon NovoCyte (NovoCyte Quanteon 4025, Agilent) |
| Software | FlowJo™ Software (version 10.7.2, BD, 2019) |
| Cell population abundance | Cell population abundances are indicated for in Extended Data Figure 7 for all four organs of a representative EAE-induced (panel a) and all four organs of a representative non-EAE-induced (panel b) mouse. For the remaining mice, count data and proportions of each cell population as a percent of the defined parental population are provided in the Source Data for Extended Data Figure 7. |
| Gating strategy | The gating strategy to determine IL-17A and IFNγ expression in T-helper cells of EAE-induced and non-EAE-induced mice is depicted in Extended Data Figure 7 and described in the corresponding legend. To summarise:<br>(1) size selection of cells in SSC-H vs. FSC-H plot;<br>(2) selection of singlets in FSC-H vs. FSC-H plot;<br>(3) for MLN, CLP and SILP, but not SC, pre-gating step for CD45+ cells (not shown);<br>(4) selection of live, single cells in a Zombie NIR ("Live/Dead") vs. SSC-H plot;<br>(5) selection of single, live CD45+ cells in a CD3 vs. CD45 plot (for SC, gates were set according to the location of the CD45+ population in other organs of the same mouse due to auto-fluourescence of the SC cells);<br>(6) selection of T cells in a CD3 vs. SSC-H plot;<br>(7) selection of CD4+ T cells in a CD8 vs. CD4 plot;<br>(8) selection of IL-17A and IFNγ-expressing CD4+ T cells in an IFNγ vs. IL17A plot |

☒ Tick this box to confirm that a figure exemplifying the gating strategy is provided in the Supplementary Information.

