## [Peer Review File · Nature Microbiology]

Peer Review Information

Journal: Nature Microbiology

Manuscript Title: Gut microbial factors predict disease severity in a mouse model of multiple sclerosis

Corresponding author name(s): Professor Mahesh Desai

Reviewer Comments & Decisions:

Decision Letter, initial version:

Message: 21st August 2023

Dear Mahesh,

Thank you for your patience while your manuscript "Gut microbiome-based prediction of autoimmune neuroinflammation" was under peer-review at Nature Microbiology. It has now been seen by 3 referees, whose expertise and comments you will find at the end of this email. Although they find your work of some potential interest, they have raised a number of concerns that will need to be addressed before we can consider publication of the work in Nature Microbiology.

In particular, additional discussion and justification is required for the use of a fibre free diet, a Muc2 KO and a defined community (referees #1 and #2) and contradictory results must be discussed (referee #1). Whilst referee felt that the sample sizes were appropriate, referees #1 and #2 both raise concerns with the small sample sizes and we therefore ask that you experimentally address this point. Referee #3 asks that you validate your findings in a mouse model with a complex microbiota and referee #2 asks for validation of the main conclusion i.e. IgA coating of *B. ovatus* and the role of GABA. Should further experimental data allow you to address these criticisms, we would be happy to look at a revised manuscript.

Please include a data availability statement as a separate section after Methods but before references, under the heading "Data Availability". This section should inform readers about

2the availability of the data used to support the conclusions of your study. This information includes accession codes to public repositories (data banks for protein, DNA or RNA sequences, microarray, proteomics data etc...), references to source data published alongside the paper, unique identifiers such as URLs to data repository entries, or data set DOIs, and any other statement about data availability. At a minimum, you should include the following statement: "The data that support the findings of this study are available from the corresponding author upon request", mentioning any restrictions on availability. If DOIs are provided, we also strongly encourage including these in the Reference list (authors, title, publisher (repository name), identifier, year). For more guidance on how to write this section please see:
<http://www.nature.com/authors/policies/data/data-availability-statements-data-citations.pdf>

* If you have not done so already we suggest that you begin to revise your manuscript so that it conforms to our Resource format instructions at <http://www.nature.com/nmicrobiol/info/final-submission>. Refer also to any guidelines provided in this letter.

When submitting the revised version of your manuscript, please pay close attention to our [href="https://www.nature.com/nature-portfolio/editorial-policies/image-integrity">Digital Image Integrity Guidelines](https://www.nature.com/nature-portfolio/editorial-policies/image-integrity). and to the following points below:

2Note: This url links to your confidential homepage and associated information about manuscripts you may have submitted or be reviewing for us. If you wish to forward this e-mail to co-authors, please delete this link to your homepage first.

Nature Microbiology is committed to improving transparency in authorship. As part of our efforts in this direction, we are now requesting that all authors identified as 'corresponding author' on published papers create and link their Open Researcher and Contributor Identifier (ORCID) with their account on the Manuscript Tracking System (MTS), prior to acceptance. This applies to primary research papers only. ORCID helps the scientific community achieve unambiguous attribution of all scholarly contributions. You can create and link your ORCID from the home page of the MTS by clicking on 'Modify my Springer Nature account'. For more information please visit please visit www.springernature.com/orcid.

If you wish to submit a suitably revised manuscript we would hope to receive it within 6 months. If you cannot send it within this time, please let us know. We will be happy to consider your revision, even if a similar study has been accepted for publication at Nature Microbiology or published elsewhere (up to a maximum of 6 months).

Reviewer Expertise:

Referee #1: autoimmunity, microbiome
 Referee #2: MS, microbiome, immunity
 Referee #3: host immune response, microbiome

Reviewer Comments:

Reviewer #1 (Remarks to the Author):

This work by Steimle et al. reports potentially interesting findings about specific microbiota characteristics and the effect of varying microbial composition on the development and progression of EAE. However, I have serious concerns with the interpretation of data and conclusions drawn throughout the paper. Thus, several points should be addressed before this work is ready for publication.

Major concerns:

- Throughout the paper, the number of mice analyzed per group was very low, with Figures 1, 3 and 4 all containing panels with only 4-5 mice per group (most often 4), and Figure 5 and 6 showing further analyses of these same groups. Given that one of the major conclusions of this paper is that individual mouse to mouse variability presents challenges and limitations, larger group numbers should be used. While I understand that germ-free mouse experiments are costly, drawing conclusions from experiments which have been performed only once with low group numbers raises concerns for replicability and reproducibility.
- While mice on fiber free diets versus fiber rich diets are compared throughout the paper,

3the Muc2 knockout mice used in Figure 1 were only analyzed on a fiber rich diet. It might be helpful to see if there is a difference in fiber-free fed Muc2 knockout mice, given that some differences were observed in later figures as a result of altering diet.

- Are there any alterations detected in intestinal permeability resulting from the knockdown of Muc2? Given the known role of Muc2 in the formation of the mucus barrier, this could be one additional functional readout of these effects.
- In Figure 4c statistical significance is attributed to comparison between data shown in Figure 2 and the data shown in Figure 4. Were these EAE experiments performed at the same time? If so, it would be much more helpful to show the SM14 and SM13 disease curves in the same graph for comparison. If the experiments were not performed at the same time, this comparison should not be made, as experiment to experiment variability can be significant owing to differences in the lots of MOG, CFA, or PTX used to induce disease, or other experimental differences. For the same reason, I also have concerns about the large scale comparisons between all groups performed in Figure 5.
- The plots displaying the T cell analyses performed in Figure 5 and Extended data Fig. 5 are extremely difficult to follow (particularly the ab, ac, etc codes to delineate significance in ED Fig. 5a-b). Moreover, why were some subsets analyzed only in some organs but not in others (for example, RORgt in the mLN but no FoxP3 or IL17, and no RORgt in the colon or ileum)? Finally, surprising differences were found which don't fit with what is known in the literature, like an inverse correlation between EAE disease and frequency of CD3 and CD8 cells in the spinal cord. Showing absolute numbers of cells rather than only proportions might help to explain some of these contradictory results.
- The findings regarding GABA and IgA coating were very interesting and represented the potential functional explanations for the phenotypes observed in the mice carrying different microbial compositions. However I felt these data were not particularly well connected to the rest of the paper and should be expanded upon further to potentially elucidate a better link between microbial, metabolite, and IgA concentrations which might functionally impact disease.
- While the authors acknowledge briefly in the discussion the contradictory finding that higher levels of Akkermansia was associated with suppression of disease in Figure 1, while it was associated with worsening of disease in all later figures, additional discussion of hypotheses for why this finding might have occurred would be helpful.

Minor concerns:

- It is surprising that Akkermansia muciniphila is not mentioned in the abstract despite its central role in the paper (it is in the title of 4 out of 8 of the Results paragraph titles, in 3 out of 6 titles of the figures, and is in one way or the other present in all 6 figures). Considering the literature present on A. muciniphila and MS, I think it might be worth including in the abstract so that future readers know to consider this paper if researching such a topic.
- When introducing the controversial role of A. muciphila (line 113) and its potential as a risk factor, the authors should also mention that previous studies have also suggested that it might have a beneficial role (Cox et al., Ann Neurol. 2021; Montgomery et al., PNAS 2020).
- Line 163-164 state that GABA was significantly elevated in mice which developed severe disease, however SM01 did not result in severe disease. The line should be re-written to reflect this.
- In Figure 4, it might be helpful to investigate the role of the three mucin-degrading bacteria by generating a drop-out bacterial group which eliminates the BT, BC, and BI bacteria (with or without also dropping AM) in addition to the SM03 and SM04 groups.

4- In Extended Data Fig. 6c, why does it appear that there are only 3 data point in the “cluster 2” bar? According to Figure 5e, there should be 24 mice in cluster 2.

Reviewer #2 (Remarks to the Author):

Summary: Steimle et al set out to define microbial risk factors that predict eae disease severity. The authors tested whether changing relative abundances of taxa within a given microbiota affect EAE outcome and used WT fed either a standard chow (fiber-rich; FR) or a fiber-free diet (FF) diet followed by induction of EAE. And concluded that changing fiber levels does not impact on EAE severity. While the rationale and reason to use Muc2 deficient mice is weak-the experiments using Muc2 deficient mice show that Muc2 deficiency is linked with less severe eae. In a series of elegant gnotobiotic experiments (using defined synthetic microbiomes with and without Akkermansia muciniphilia coupled with Am monocolonization and GF mice) the authors identify that under these experimental settings Am rather displays a risk factor that is linked with more severe EAE disease. Metabolomic analysis revealed that high γ -amino butyric acid levels are linked with more severe EAE disease. Finally the authors suggest that IgA coating index of *B. ovatus* is a potential predictor of disease development by performing correlative analyses.

This study is interesting and for the most part well executed however falls short on validating their identified MS-predictors which seems critical. The identified predictions γ -amino butyric acid and IgA coated *B. ovatus* should be experimentally tested/validated whether they are causally involved with modulating EAE disease severity.

Major comments

- Does the fiber content change the composition of the microbiota? To avoid confounding factors such as other nutrients (fat, carbohydrates, vitamins, minerals) to impact on EAE susceptibility that are different between diets the authors should use a diet that is identical in its composition and only differs in its fiber content. Which type of fiber? Fermentable or non fermentable? Type of fiber matters in this context (e.g. PMID: 29993025) and should be discussed. Type of fiber differentially impacts on the microbiota including IgA producing commensals including bacteroides (e.g. 32332136).
- The authors also do not show whether the changes they observe within the microbiota in Muc2ko mice are of functional relevance (e.g. tested by FMT)? The authors mention this in the text but the contributions of host genotype and/or host-changed microbiota in EAE of MUC2 mice remain unaddressed. Altogether, having the Muc2 data set in the manuscript is confusing and raises more questions than providing answers. There is enough rationale from findings by others (which the authors also discuss) to test the role of Akkermansia in MS and whether Akkermansia presence or function is a MS predictor (see summary (36113423) of iMSMS (36113426) consortium).
- It would be important and helpful for the reader if the 5 criteria to define the potential *A.muciniphila*-associated and disease-mediating metabolite are briefly discussed in the main text.

5- The authors identify *A. muciniphilia* as a risk factor that is linked with more severe EAE disease- it would be important to test whether this is linked with an increased MOG-antigen specific Th1/17 or Tc1/17 response and IgA plasma cell response in spinal cord/CNS.
- Is γ -amino butyric acid produced by *A. muc* and sufficient to worsen EAE when administered alone?
- The authors findings suggest that IgA coating levels of *Bacteroides ovatus* represents a potential measure to predict EAE severity. However experimental validation of this prediction needs to be performed such as colonization with IgA+ *B. ovatus* compared to IgA- *B. ovatus*. It would be important whether the IgA coating index of *B. ovatus* is connected with an increased MOG-antigen specific Th1/17 or Tc1/17 response and IgA plasma cell response in spinal cord/CNS.

Minor comments

n/a

Comments on data integrity reproducibility

- Fig 1 concerning data using *Muc2* deficient and sufficient mice. An n of 4-5 seems insufficient to come to firm conclusions. The n number needs to be increased.
- Fig 2g- the authors state that there is a FF diet induced significant increase of *Akkermansia muciniphilia* however fail to provide a test for statistical significance-this should be done.

Reviewer #3 (Remarks to the Author):

The paper by Desai in colleagues seeks to identify predictors of disease development using a mouse model of MS. The group uses controlled restricted communities to determine how composition, abundance of certain members, metabolites and immune parameters correlate with disease severity. The conclusion is that IgA coating of one member, *B. ovatus*, seemed to consistently correlate with disease severity. Overall, the use of controlled communities provides a nice lesson in microbial ecology and is a well controlled system for identifying potential correlates of disease. The manuscript makes an important point that community composition matters and output of disease might change based on community and/or function. The use of the metatranscriptomics is of interest here as more of the community needs to begin to look at how the function of the microbiota changes rather than just membership. The data are well presented and good group sizes are appropriate. However, while the use of these controlled systems is nice they should be complemented by validating their findings in the context of the more natural and complex community. Without this, this study reflects what happens in a contrived 12-14 member community and likely does not reflect reality. This fact makes the conclusions of this manuscript much less impactful. Moreover, one of the main goals of this study was to identify "predictors" of disease. Indeed, in the clinic diagnosis and treatment regime of this disease it would be nice to have a readout of disease severity. Again, the identification that IgA coating of *B. ovatus* is a predictor of disease is only supported in this contrived community. What happens in a more complex community as what would be seen in humans. Does IgA coating of *B. ovatus* still act as a "predictor"? Based on this, the language and conclusions

6drawn from this dataset are oversold and the use of predictor is far too strong of a statement to be made from this experimental design.

Author Rebuttal to Initial commentsResponse to Reviewers (7 March 2024)

Manuscript ID: NMICROBIOL-23061597

Manuscript Title: Gut microbiome-based prediction of autoimmune neuroinflammation

Dear reviewers,

Thank you very much for your constructive comments and suggestions, which helped to substantially improve the quality of this manuscript. As you will see, we took care to pacify all concerns and questions raised by the reviewers. All changes to the manuscript are mentioned in this rebuttal letter and are also evident in the revised version of the manuscript in “track changes” mode.

Please note the following formatting scheme used in this rebuttal letter:

Reviewers’ original comments are formatted in	bold font, font size 11
Authors’ answers and comments are formatted in	normal font, font size 11
Quotes from the manuscript are formatted in	normal font, font size 10, grey background

This document also contains clickable hyperlinks in order to help the reviewers navigate through the document when we refer to other sections of this rebuttal letter.

Before starting to address the reviewers’ comments and concerns in detail, we would like to briefly highlight the major changes we made in the manuscript. Importantly, in the revised version of the manuscript, all major conclusions remain *unchanged*. The newly generated data during the revision majorly helped us to shore up the conclusions.

I.**I. Brief overview of major changes****1) INTRODUCTION**

We extensively rephrased the last paragraph of the Introduction.

2) RESULTS

This study contains complex and large datasets in a disease model, which we acknowledge may have been difficult to follow. Based on reviewers' helpful comments, we were able to refine our scope and clearly define the key results to focus on. The corresponding changes further helped us to respect the word count for *Nature Microbiology* (see point #6 below).

We considerably rephrased the text referring to **Fig. 1** (EAE in SPF-housed wild-type (WT) and *Muc2*^{-/-} mice). We also re-arranged and/or combined figure panels of **Fig. 1** and **Extended Data Fig. 2**. This is described in detail in section III.2 of this rebuttal letter. Since we focus on predicting EAE development using microbiome data *before* disease onset, we removed data on the microbiota composition *during* EAE. This change facilitates the comprehensibility of the manuscript and avoids additional confusion without losing crucial information.

We substantially rephrased the text referring to **Fig. 3**, which contains data on metabolomics and metatranscriptomic analyses in SM13- and SM14-colonized mice and the role of GABA. The corresponding figure panels remained unchanged. See section II.6 of this rebuttal letter for details.

We shortened the text referring to **Fig. 4**, but the corresponding figure panels remained unchanged.

We removed former **Fig. 5g,h** (immune data) and compiled these data in a new figure (**Extended Data Fig. 7**). Former **Figs. 6a–g** were moved to **Fig. 5** and are now referred to as **Fig. 5g–m**. Consequently, we adapted the text referring to **Fig. 5**.

Additional experiments performed as part of this revision are shown in a new **Fig. 6a–g** and **Extended Data Figs. 9 and 10**. See section IV of this rebuttal letter for details.

3) DISCUSSION

We majorly rephrased certain paragraphs of the Discussion.

4) MATERIALS AND METHODS

We added information related to the new experiments.

5) EXTENDED DATA FIGURES
In addition to the new **Extended Data Figs 9 and 10**, as mentioned above, former **Extended Data Fig. 7** was reworked and relabeled as **Extended Data Fig. 1** in the revised version of the manuscript. This new figure helps to introduce and summarize the rephrased storyline of the manuscript.

6) WORD COUNT FOR *NATURE MICROBIOLOGY*

Nature Microbiology asks for a word count of < 4500 words (Introduction, Results and Discussion). In general, in order to respect the word count, we streamlined the manuscript including all three sections (Introduction, Results and Discussion).II. Reviewer #1

This work by Steimle et al. reports potentially interesting findings about specific microbiota characteristics and the effect of varying microbial composition on the development and progression of EAE. However, I have serious concerns with the interpretation of data and conclusions drawn throughout the paper. Thus, several points should be addressed before this work is ready for publication.

We sincerely thank the reviewer for their valuable comments on the earlier version of our manuscript. These comments helped us to considerably improve the manuscript. All major and minor concerns raised by this reviewer are addressed below.

Major concerns (points #1–7 below)

- 1) **Throughout the paper, the number of mice analyzed per group was very low, with Figures 1, 3 and 4 all containing panels with only 4-5 mice per group (most often 4), and Figure 5 and 6 showing further analyses of these same groups. Given that one of the major conclusions of this paper is that individual mouse to mouse variability presents challenges and limitations, larger group numbers should be used. While I understand that germ-free mouse experiments are costly, drawing conclusions from experiments which have been performed only once with low group numbers raises concerns for replicability and reproducibility.**

We thank the reviewer for this comment. We understand the reviewer's point that the number of mice per group might seem low, especially for multiple group-wise comparisons. However, the minimum group size of $n = 4$ for gnotobiotic mice is backed by a statistical power calculation. The animal ethics committee at University of Luxembourg is strict on the number of mice and the animal numbers are only approved based on justified power calculations.

We calculated the group numbers for gnotobiotic experiments based on EAE disease course data obtained from SPF-housed mice, which showed a clear split into two distinct EAE phenotypes (**Fig. 1**). Input data to calculate the minimum group numbers were:

- Mean \pm SD of AUC for severe phenotype = 45.3 ± 9.3
- Mean \pm SD of AUC for mild phenotype = 8.4 ± 7.4

- Statistical test: a two-sided Wilcoxon-Mann-Whitney test was used, as the input data represent discontinuous values.
- Anticipated power ($1 - \beta$ error probability): 80%. Using a power of 80% is considered to be sufficient for statistical purposes, as such power helps to avoid use of excess mice in accordance with the 3R principles.
- As we performed a maximum of 15 groupwise comparisons across the gnotobiotic groups and readouts, we lowered the α error probability of 0.05 to 0.0033 by performing a Bonferroni correction. With a resulting effect size of 4.47, we calculated a group size of $n = 4$.

We would also like to highlight the following points, which are based on our results:

- In contrast to individual-level analyses, we find that groupwise analyses are unsuitable (and sometimes even misleading) to uncover reliable microbial risk factors or disease course predictors. Even in groups of $n = 4$, we found considerable inter-individual variations, irrespective of which “group” individuals belong to and the group size. Ultimately, we acquired and processed data of 65 EAE-induced gnotobiotic mice. Although this number may not seem especially high compared to other studies utilizing SPF mice for EAE experiments, to our knowledge, there is no study that has employed such a high number for gnotobiotic experiments using six different synthetic communities. It is also important to recognize that the work load related to performing such experiments is not trivial. These experiments were performed inside our specific-opportunistic-pathogen-free (SOPF; that is, double barrier) germ-free animal facility, with EAE scoring done for every single day for every single mouse (starting day 8 after EAE induction). Increasing mouse numbers under such a setting and protocol therefore also presents a major logistical challenge.
- We would also like to point out that in the new experiment performed, as part of the revision (new **Fig. 6** and **Extended Data Figs. 9, 10**), the sample size is 13 because we do not perform analysis on the group level (for details, see also response to Reviewer #3).
- In support of the aforementioned points, we also ran FR-fed SM13-colonized and FR-Fed SM14 colonized mice in three independent experiments with 3–5 mice per group. Importantly, we did not determine any EAE course differences between these

independent experiments (for the same SM), and we observed comparable patterns of inter-individual variation. Thus, these data from independent experiments (for the same SM) address the reviewer's point regarding the replicability and reproducibility of our experiments. Our major conclusions related to IgA coating of *B. ovatus* are derived from SM14, SM13 and SM12 and reproducing these results in SM14 and SM13 groups through three independent experiments is sufficient to back up the main conclusions.

- Finally, we would like to point out that Reviewer #3 specifically comments that “good group sizes are appropriate.”

2) While mice on fiber free diets versus fiber rich diets are compared throughout the paper, the Muc2 knockout mice used in Figure 1 were only analyzed on a fiber rich diet. It might be helpful to see if there is a difference in fiber-free fed Muc2 knockout mice, given that some differences were observed in later figures as a result of altering diet.

We thank the reviewer for this suggestion. However, we do not fully understand what the reviewer means by “given that some differences were observed in later figures as a result of altering diet”, as there were changes in the microbiota composition (gnotobiotic experiments with SM13 and SM14) following different diets, however the EAE disease scores for the same SM or GF mice were concordant between the fiber-rich and fiber-free diets (**Fig. 2c, Extended Data Fig. 4b**). As demonstrated by variances in EAE-associated readouts explained by the factor “Diet” (referred to in the manuscript as “explained variances” or “eta squared”), the impact of the diet and its associated shift in relative abundances of bacterial taxa on EAE development was generally low for all tested groups of wildtype mice. Consequently, we deemed it unnecessary to further explore this aspect in mice of other genotypes or harboring other reduced or complex microbiota compositions. Thus, dietary modulation was not performed to evaluate the impact of diet on EAE development, but rather to modulate relative abundances within established indigenous microbial communities to uncover potential microbial disease predictors.

3) Are there any alterations detected in intestinal permeability resulting from the knockdown of Muc2? Given the known role of Muc2 in the formation of the mucus barrier, this could be one additional functional readout of these effects.

This is indeed a very interesting question and an understandable assumption. We elaborated in detail on the reasoning of using this knock-out mouse strain in section III.2 of this rebuttal letter. A mechanistic explanation for the observed EAE phenotype in *Muc2*^{-/-} mice, however, was not within the scope of our study and is also not relevant for the overall conclusion in the paper. Additionally, other groups have addressed effects of Muc2 deficiency on physiological and pathophysiological

13

aspects and have observed elevated intestinal permeability relative to Muc2-competent littermate controls. Given that this is an established mouse model, we did not consider it productive to repeat these readouts. Examples of such studies are:

- Sci Rep. 2020. doi: [10.1038/s41598-020-78141-4](https://doi.org/10.1038/s41598-020-78141-4).
- Nat Commun. 2020. doi: [10.1038/s41467-019-14182-2](https://doi.org/10.1038/s41467-019-14182-2)
- Cell Mol Gastroenterol Hepatol. 2021. doi: [10.1016/j.jcmgh.2020.07.003](https://doi.org/10.1016/j.jcmgh.2020.07.003)

- 4) **In Figure 4c statistical significance is attributed to comparison between data shown in Figure 2 and the data shown in Figure 4. Were these EAE experiments performed at the same time? If so, it would be much more helpful to show the SM14 and SM13 disease curves in the same graph for comparison. If the experiments were not performed at the same time, this comparison should not be made, as experiment to experiment variability can be significant owing to differences in the lots of MOG, CFA, or PTX used to induce disease, or other experimental differences. For the same reason, I also have concerns about the large scale comparisons between all groups performed in Figure 5.**

We agree with the reviewer that different batches of reagents used for EAE induction may substantially impact the outcome of neuroinflammation in mice. This is particularly the case for PTX, as nicely demonstrated in a publication by Huntemann et al. (J Neurosci Methods. 2022. doi: [10.1016/j.jneumeth.2021.109443](https://doi.org/10.1016/j.jneumeth.2021.109443)). However, we can make the following statements concerning reagent batches and EAE induction runs:

- In groups with $n \leq 5$, all individual mice of a given group were EAE-induced at the same time with the same batch of MOG, CFA and PTX.
- In groups with $n > 5$, mice were EAE-induced on at least two distinct time points (with at least two distinct batches of MOG, CFA and PTX). Importantly, in these groups, we observed high disease induction reproducibility as demonstrated by comparable ratios R, with R defined as:

$$R = \frac{\# \text{ mice with mild disease}}{\# \text{ mice with severe disease}}$$

- No group was ever run as a single group, but always together with at least one other group.

14- For each run, irrespective of the number of groups having been EAE-induced together, we observed the full spectrum of EAE disease phenotypes, from relatively mild to relatively severe. In no run did we only observe one particular phenotype.
- Even in groups with $n \leq 5$, where all mice were EAE-induced at the same time, we observed the full spectrum of possible phenotypes.

These observations indicate that potential differences in disease-inducing capacities of the used preparations are outweighed by other factors (microbiota and host).

However, given the aforementioned challenges in achieving reliable reproducibility in EAE, we did not prepare MOG, CFA and PTX suspensions ourselves. Instead, we used a ready-to-use kit from Hooke Laboratories, as stated in the Methods section, which provides “high quality emulsions with consistent, pre-characterized potency” (<https://hookelabs.com/>). The Hooke Laboratory EAE kit has shorter expiry dates, so the solutions are used as fresh as possible to induce disease. We specifically employed this kit throughout the study to further limit batch effects and potential reagent preparation-associated discrepancies. A good quality control of our experiments is data shown in **Fig. 2**: we ran 13SM and 14SM experiments three independent times and, although we observe variability in disease course within the SM–diet combinations, we did not find significant variations in disease scores between batches (see also the second to last bullet point in major point #1 above).

At the same time, we completely understand the concern of this reviewer, because the published literature largely contains experiments performed with EAE reagents prepared and mixed in-house (sometimes even by different researchers within the same group), which understandably could have an impact from one batch of animals to the other. Thus, while we understand the reviewer’s general concern, we have no reason, within our experiments and data, to assume that distinct batches of EAE-inducing reagents (in our case, kits purchased from Hooke Laboratories) severely affected our downstream experiments and analyses.

- 5) **The plots displaying the T cell analyses performed in Figure 5 and Extended data Fig. 5 are extremely difficult to follow (particularly the ab, ac, etc codes to delineate significance in ED Fig. 5a-b). Moreover, why were some subsets analyzed only in some organs but not in others (for example, RORgt in the mLN but no FoxP3 or IL17, and no RORgt in the colon or ileum)? Finally, surprising differences were found which don’t fit with what is known in the literature, like an inverse correlation between EAE disease and frequency of CD3 and CD8 cells in the spinal cord. Showing absolute numbers of cells rather than only proportions might help to explain some of these contradictory results.**

15We thank the reviewer for this comment as it encouraged us to revisit and reanalyze our flow cytometry data, which revealed a previously undetected phenomenon that was only apparent in spinal cord samples. We encountered significant autofluorescence of non-CD45⁺ cells, especially in the detection channels V725/60-H and V780/60-H that were used to detect CD3 and CD45 expression, respectively. This, in turn, led to inclusion of false positive events when gating for these two particular populations in spinal cord cells. Our previous gating strategy, which represents a commonly used gating approach, was unsuitable to reveal this phenomenon. To address this issue, we eventually gated for CD45⁺ cells from a dot plot displaying CD3 vs. CD45, which made this autofluorescence detectable and allowed us to properly distinguish false-positive from correct-positive events. To visualize this new gating approach, we provide the revised gating strategy in detail in **Extended Data Fig. 7a,b**. This figure exhibits stepwise gating for two representative mice (one EAE-induced and one non-EAE-induced) for each of the four organs separately. Importantly, samples for each mouse were detected in the same run and with the same antibody master mix, thus allowing for direct comparison of staining intensities between organs. The respective figure legend provides further information. We re-analyzed all samples and focused only on CD45⁺ cells, CD3⁺ cells, CD8⁺ cells, CD4⁺ cells as well as on IL-17 and IFN γ -expression in CD4⁺ cells to streamline the data presentation. Additionally, we applied more stringent criteria on sample quality for sample inclusion into downstream analysis by excluding samples with low less than 1000 events after gating on CD45⁺ cells. Outliers were also determined by group and values not within a range of mean $\pm 2\sigma$ were removed from the datasets.

We also revisited our approach to report populations-of-interest in the spinal cords (SC) as proportions of CD45⁺ cells. Finally, we agree with the reviewer that this approach is very prone to mask quantitative infiltration of CD45⁺ cells, CD45⁺CD3⁺ cells and further subpopulations into SCs and reporting on absolute numbers is required to do so for meaningful interpretation of the obtained data. Given that we did not include this aspect in our sample preparation pipeline, we are unable to retrospectively report absolute cell numbers per SC. To circumvent this aspect, we suggest a different approach to highlight infiltration of T cells into SC in mice with severe EAE symptoms. Although the proposed approach is unconventional, in the absence of absolute cell count data, we find it to be a justifiable alternative. Importantly, we only applied this approach to EAE-induced mice, as CD45⁺ cell counts in SCs of non-EAE-mice were too low to be subjected to downstream analysis.

Evaluation of CD45⁺ cell infiltration into SCs *solely based on detected events* requires a reliable parental population. All three possibilities (FSC/SSC, singlets and viable cells) are usually considered unsuitable to serve as such parental populations. We evaluated the quality of the “Viable” gate as a parental population. Naturally, events in this gate were already pre-gated for location in the FSC/SSC plot as well as for singlets. While we consider the singlets gating for

16relatively stringent and un-biased, initial gating in FSC/SSC plots are more prone to bias. Thus, we used samples from mesenteric lymph nodes (MLN) of the same run as a template to set this gate in the FSC/SSC depiction for SC as MLN provide high numbers of CD45⁺ cells. Consequently, cells within this gate should contain all CD45⁺ cells and, particularly in SCs, other non-CD45⁺ cells. While we pre-gated for CD45⁺ cells in the “Live/Dead”-vs-SSC depiction in organs other than SC by excluding events with higher values on the SSC axis (**Extended Data Fig. 7a, b**), we included *all* “live” events for SC analysis to obtain a less biased reference set of events that could be used as a parental population. Our final goal was to demonstrate differences in CD45⁺CD3⁺ cell infiltration into SC between mice of different disease clusters (Cluster 1 with severe symptoms and Cluster 2 with minor symptoms). First, we evaluated whether the relative proportions of viable cells among all detected events (“%V”, before any gating) were different between clusters. Significant differences would result in a downstream analysis bias. Indeed, we detected significant differences in %V between the two clusters. In order to draw samples of comparable %V between the two clusters, and therefore reduce downstream analysis bias, we only included samples that provided %V within a range of mean \pm SD of *all* analyzed samples. After sample drawing, we re-assigned samples to their respective clusters and their %V revealed no significant difference, thus eliminating remaining gating bias and providing a *homogenous* parental population that can be used to address *differences* in downstream gates. This approach revealed significantly increased infiltration of various lymphocyte populations as shown in **Extended Data Fig. 7e**.

However, in case the reviewer does not agree with the proposed approach, we will report populations-of-interest in the spinal cord as percentages of CD45⁺. However, the proportions *within* the infiltrating populations are comparable between the clusters, therefore, with CD45⁺ as the parental population, we do not detect any differences between the clusters. See figure below as an alternative to **Extended Data Fig. 7e**, in case the reviewer does not agree with our approach explained above.

Given that the immune data are not essential for the main conclusions of this manuscript and given that we had to add a new main **Fig. 6** and two new **Extended Data Figs. 9** and **10**, as part of the revision of this manuscript, we decided to consolidate our immune data into one **Extended Data Fig. 7**.

- 6) **The findings regarding GABA and IgA coating were very interesting and represented the potential functional explanations for the phenotypes observed in the mice carrying different microbial compositions. However I felt these data were not particularly well connected to the rest of the paper and should be expanded upon further to potentially elucidate a better link between microbial, metabolite, and IgA concentrations which might functionally impact disease.**

We thank the reviewer for recognizing our GABA and IgA coating results as “very interesting”. We agree with the reviewer that these points appeared to be somewhat disconnected in the previously submitted version of the manuscript. Therefore, we made extensive changes to the text to address this issue. First, we substantially rephrased the GABA-related paragraph in the Results section. This paragraph now reads as follows (lines 132-141 in the revised version of the manuscript):

Among the 14 metabolites (**Fig. 3c**), whose concentrations at the end of the disease course significantly correlated with AUC, only γ -amino butyric acid (GABA) emerged as a metabolite-of-interest in cecal samples, showing a positive association with EAE (**Fig. 3c**). Importantly, GABA concentration was significantly elevated in non-EAE-induced mice harbouring *A. muciniphila*-containing SM combinations (**Fig. 3d**). These results suggest that increased cecal GABA levels, even before disease induction, were primarily a result of *A. muciniphila* presence and not a consequence of EAE induction. Thus, assessment of

cecal GABA concentrations might allow for prediction of EAE severity. However, not all SM01-colonized mice developed severe disease upon EAE induction (**Extended Data Fig. 4b–d**), despite consistent GABA concentrations in non-EAE-induced (**Fig. 3d**, left panel) and EAE-induced (**Fig. 3d**, right panel) SM01-colonized mice, raising the need for individualized disease predictors.

We also rephrased the GABA-related paragraph in the Discussion, which reads now as follows (lines 317-329 in the revised version of the manuscript):

We next sought to investigate whether concentrations of certain bacterial metabolites might predict disease development, as these metabolites could be a result of inter-microbial networks. Importantly, our analysis approach demonstrated that focusing only on correlations between metabolite concentrations and disease phenotypes resulted in shortlisting false-positive potential predictors. Applying a more stringent, context-focused analysis pipeline revealed that only elevated cecal concentrations of GABA were associated with increased EAE in two *A. muciniphila*-encompassing microbiota compositions. However, this was tested only in four reduced microbiota compositions and pre-EAE GABA concentrations only allowed to assess risk for severe EAE but did not predict individual disease development. Contrary to our findings, faecal GABA concentrations in MS patients were decreased compared to healthy controls⁴⁵. Although the ability of *A. muciniphila*⁴⁶ and other gut commensals^{47,48} to produce GABA have previously been reported, the effect on other microbes and the resulting neuroinflammation-promoting or -preventing properties remain elusive. Thus, the potential predictive qualities of GABA or other metabolites must be seen in the context of a given microbiota, which reduces their practicability as universal or even individual disease predictors.

The reviewer comments on a missing link between microbes, metabolites and IgA coating indices, which might functionally impact disease. We agree with the reviewer that a functional link between the microbes, their metabolites and IgA is missing. However, elucidating such functional connections was out of this manuscript's scope, as the goal of our study was to demonstrate and highlight common misconceptions on how to address the quest for microbiota-associated disease predictors (or "risk factors") in general as well as to offer more promising alternatives. Examining a functional link between different microbes, metabolite (GABA) and IgA concentrations is interesting and as such is a topic for a future study. Nonetheless, in response to the reviewer's comment, we have done all possible modifications in the text as mentioned above to pacify this comment as much as possible. To establish a functional link between the said parameters would require additional gnotobiotic animal experiments, including new animal protocol applications/approvals. Such new experiments would require obtaining ethical approvals for new animal protocols and would not be feasible within the six-month timeline available for the revision of this manuscript.

On the other hand, we are convinced that this study represents a valuable contribution to the field of microbiota and autoimmune diseases for the following reasons:

- We focused on exploring different microbiota-associated readouts to uncover practical measures that allow for prediction of autoimmune neuroinflammation development, which could be used as a *template* for other diseases as well.
- We concluded that disease-relevant interactions between the microbiota and the host are rooted in a *bi-directional* interaction between the host and a host-specific microbial network. Consequently, we recommend to rethink how microbiota-related data are analyzed in the context of a particular disease to account for these complex, but crucial interactions.
- One of our main conclusions is that focusing on single taxa alone fails to provide meaningful predictions in an individual. However, we identified IgA coating indices of certain commensals as a measure that allowed for individual disease course prediction.

Taken together, we consider our study to be crucial for directing the field towards a more targeted strategy to uncover disease-relevant microbial characteristics. These characteristics could eventually be used for diagnostic purposes or to develop novel therapeutic strategies. Associated functional implications and mechanisms underlying our observations will be evaluated in a follow-up study.

7) While the authors acknowledge briefly in the discussion the contradictory finding that higher levels of *Akkermansia* was associated with suppression of disease in Figure 1, while it was associated with worsening of disease in all later figures, additional discussion of hypotheses for why this finding might have occurred would be helpful.

We thank the reviewer for this point. In the revised manuscript, we addressed these “contradictory” findings at several places, which are listed and quoted below.

Roughly 20% of the rephrased version of the Discussion is attributed to this topic. By offering additional discussion of underlying reasons for these contradictory findings (including possible strain specific differences of *Akkermansia* between the SPF and gnotobiotic mice), we believe we have adequately addressed this point, both qualitatively and quantitatively. This rephrased section in the Discussion reads as follows (starting at line 293 in the revised version of the manuscript):

Using mice of different genetic backgrounds and of distinct complex microbiota compositions, we found the genus *Akkermansia* to be the most negatively associated with EAE disease development. Next, we examined whether this finding could be reproduced in gnotobiotic, genetically homogenous mice harbouring different

20

combinations of a reduced reference microbiota, with or without *A. muciniphila*. Intriguingly, we found *A. muciniphila* to be positively associated with EAE severity in certain mice harbouring specific reduced communities. These contradictory results on *A. muciniphila* are corroborated by findings from other groups. *A. muciniphila* was reportedly increased in MS patients across various human cohorts^{2,5,7,10,35}. Other studies, however, report on positive effects of *A. muciniphila* on maintaining general gut homeostasis^{36,37} or on progression of autoimmune neuroinflammation in mice³⁸ or humans^{39,40}. Given the genotypic and phenotypic diversity of *A. muciniphila*⁴¹, these reported discrepancies might be rooted in strain-specific effects on the host. However, our study using the same *A. muciniphila* strain in all gnotobiotic experiments implies that mutual influences between a suspected risk species and the microbial environment crucially shape the overall microbiota's disease-impacting potential. This potential was unrelated to the presence or absence of a particular taxon, suggesting that focusing on certain combinations of taxa, rather than single taxa alone, is essential to derive more reliable conclusions from microbiota profiles.

We also address this point in the Results section starting at line 220, which reads as follows:

Presence or absence of a specific strain or strain combination was insufficient to predict the individual outcome of any of the tested EAE-associated readouts (**Fig. 5j, Extended Data Fig. 8b**), indicating that the potential disease-driving or -preventing properties of a given strain or strain combination is determined by the background microbiota.

Minor concerns (points #8–12 below)

- 8) **It is surprising that *Akkermansia muciniphila* is not mentioned in the abstract despite its central role in the paper (it is in the title of 4 out of 8 of the Results paragraph titles, in 3 out of 6 titles of the figures, and is in one way or the other present in all 6 figures). Considering the literature present on *A. muciniphila* and MS, I think it might be worth including in the abstract so that future readers know to consider this paper if researching such a topic.**

We agree with the Reviewer's suggestion and have changed the abstract accordingly to highlight the existing link in the literature between MS and *Akkermansia muciniphila*. At the same time, while streamlining the manuscript as per the reviewers' suggestions, we ended up reducing the mention of *A. muciniphila* in Results titles and figure titles.

- 9) **When introducing the controversial role of *A. muciphila* (line 113) and its potential as a risk factor, the authors should also mention that previous studies have also suggested that it might have a beneficial role (Cox et al., Ann Neurol. 2021; Montgomery et al., PNAS 2020).**

We completely agree with the reviewer's suggestion and now mention the indicated papers (Ann Neurol. 2021. doi: 10.1002/ana.26084; PNAS. 2020. doi: 10.1073/pnas.2002817117) in line 301 in the revised version of the manuscript.

- 10) **Line 163-164 state that GABA was significantly elevated in mice which developed severe disease, however SM01 did not result in severe disease. The line should be re-written to reflect this.**

We thank the reviewer for this comment and we apologize for this misleading statement. We rephrased the whole paragraph, which now reads as previously mentioned in section II.6 of this rebuttal letter.

- 11) **In Figure 4, it might be helpful to investigate the role of the three mucin-degrading bacteria by generating a drop-out bacterial group which eliminates the BT, BC, and BI bacteria (with or without also dropping AM) in addition to the SM03 and SM04 groups.**

This is indeed a very understandable suggestion and we were also thinking about performing such an experiment. Additionally, we were also considering to run an “SM10” group, lacking all mucin glycan-degrading bacteria (removal of AM, BT, BC and BI from the SM14 community). However, we eventually decided not to run these groups because we already assessed indirect measures of mucosal barrier integrity in SM14-, SM13-, SM12-, SM04-, SM03- and SM01-associated mice (results of these mucosal barrier integrity-associated readouts are shown in **Extended Data Fig. 6**). We demonstrated that there was no evidence on the changed mucosal barrier integrity decisively influencing EAE disease progression. Consequently, there was (and still is) no reason to further explore the potential contribution of mucin glycan-degrading commensals on EAE development. For similar reasons, it would be difficult to obtain approval for a new animal protocol approval from the animal ethics committee of University of Luxembourg, as the committee also evaluates the scientific justification for performing new animal experiments, which is weak in this case.

- 12) **In Extended Data Fig. 6c, why does it appear that there are only 3 data point in the “cluster 2” bar? According to Figure 5e, there should be 24 mice in cluster 2.**

We agree with the reviewer about this discrepancy. Unfortunately, we only had enough remaining fecal material from three “Cluster 2” mice to determine secretory IgA (sIgA) concentrations. Fecal material of the other 21 “Cluster 2” mice had already been used for other readouts. We have made a mention of this point in the legend of new **Extended Data Fig. 8c**. Please note that we do not draw any crucial conclusion from this particular data set (sIgA). However, we found it interesting to show that sIgA concentrations are unrelated to emerging EAE disease phenotypes. In fact, this is

22

already demonstrated by the considerable dispersion of sIgA concentrations in Cluster-1 mice alone.III. Reviewer #2

Summary: Steimle et al set out to define microbial risk factors that predict EAE disease severity. The authors tested whether changing relative abundances of taxa within a given microbiota affect EAE outcome and used WT fed either a standard chow (fiber-rich; FR) or a fiber-free diet (FF) diet followed by induction of EAE. And concluded that changing fiber levels does not impact on EAE severity. While the rationale and reason to use Muc2 deficient mice is weak-the experiments using Muc2 deficient mice show that Muc2 deficiency is linked with less severe EAE. In a series of elegant gnotobiotic experiments (using defined synthetic microbiomes with and without *Akkermansia muciniphila* coupled with Am monocolonization and GF mice) the authors identify that under these experimental settings Am rather displays a risk factor that is linked with more severe EAE disease. Metabolomic analysis revealed that high γ -amino butyric acid levels are linked with more severe EAE disease. Finally the authors suggest that IgA coating index of *B. ovatus* is a potential predictor of disease development by performing correlative analyses.

This study is interesting and for the most part well executed however falls short on validating their identified MS-predictors which seems critical.

The identified predictions γ -amino butyric acid and IgA coated *B. ovatus* should be experimentally tested/validated whether they are causally involved with modulating EAE disease severity.

We thank the reviewer for several supportive comments on our study and also for finding our gnotobiotic experiments “elegant”. We address the point about validating our MS predictor in detail below. We also address the rationale for using Muc2-deficient mice in detail below.

Major comments (points #1–6 below)

- 1) Does the fiber content change the composition of the microbiota? To avoid confounding factors such as other nutrients (fat, carbohydrates, vitamins, minerals) to impact on EAE susceptibility that are different between diets the authors should use a diet that is identical in its composition and only differs in its fiber content. Which type of fiber? Fermentable or non fermentable? Type of fiber matters in this context (e.g. PMID: 29993025) and should be

24discussed. Type of fiber differentially impacts on the microbiota including IgA producing commensals including bacteroides (e.g. 32332136).

We thank the reviewer for this interesting comment. We would like to highlight that by no means we aimed to investigate the role of dietary fibers in the context of EAE, which was done before by other groups, including study mentioned by the reviewer from the Krishnamoorthy group (Sci Rep. 2018. doi: 10.1038/s41598-018-28839-3). In this study, we employ a fiber-free (FF) diet with the sole purpose of modulating relative abundances of SM14-constituent commensals. The FF diet-mediated microbiota changes within this consortium were predictable based on previous experiments and corresponding publications from our group:

- Nat Microbiol. 2023. doi: 10.1038/s41564-023-01464-1
- Cell. 2016. doi: 10.1016/j.cell.2016.10.043
- EMBO Mol Med. 2023. doi: 10.15252/emmm.202217241
- mSystems. 2021. doi: 10.1128/mSystems.00717-21
- Cell Host & Microbe, in press, preprint doi: 10.1101/2022.04.03.486886

Furthermore, we performed this diet-mediated microbiota manipulation for the 13SM- and 14SM-colonized mice and did not detect any differences in group means between FR- and FF-fed mice for any EAE-associated readouts. At the same time, however, we observed considerable intra-group variations in EAE-associated readouts (see **Fig. 5b** in the revised version of the manuscript). Consequently, we deemed it unnecessary to modulate relative abundances through dietary modulation for other SM combinations with less constituent strains.

The reviewer mentioned several confounding factors that are important to consider when the impact of dietary fiber on EAE disease progression is investigated. Although this is certainly true, addressing this point was not the purpose of our study. As outlined in detail in the manuscript, we strongly emphasize to analyze data not based on group characteristics, such as diet, but performed the analyses on an individual level, treating diet as a background factor that contributes to the variation between individuals. Such an approach allowed us to uncover disease predictors that can be generalized across diets and microbiotas (assuming the microbial predictor is present in a particular microbiota).

The reviewer mentioned a very interesting paper that reports on altered host IgA secretion due to changed abundances of members of the *Bacteroides fragilis* group species (such as *Bacteroides faecis*, *Bacteroides caccae*, and *Bacteroides acidifaciens*) in response to a diet rich in soluble fibers (Appl Environ Microbiol. 2020. doi: 10.1128/AEM.00405-20). However, we do not have any reason to believe that a comparable effect plays an important role in our experiments, as there were

25

no significant differences in EAE courses when comparing FR-fed to FF-fed mice harboring the same set of intestinal microbes.

Finally, to clarify the point of the Reviewer that “*the authors should use a diet that is identical in its composition and only differs in its fiber content*”: we previously performed the suggested feeding experiment with diets that are identical in their composition but only differ in their fiber context, although this was not done in the context of EAE. In the first study (Cell Host & Microbe, *in press*, preprint doi: 10.1101/2022.04.03.486886), we replaced the simple sugar in our FF diet with the host digestible starch or microbiota-digestible fibers such as those from apple, oat and wheat.

- 2) **The authors also do not show whether the changes they observe within the microbiota in Muc2ko mice are of functional relevance (e.g. tested by FMT)? The authors mention this in the text but the contributions of host genotype and/or host-changed microbiota in EAE of MUC2 mice remain unaddressed. Altogether, having the Muc2 data set in the manuscript is confusing and raises more questions than providing answers. There is enough rationale from findings by others (which the authors also discuss) to test the role of Akkermansia in MS and whether Akkermansia presence or function is a MS predictor (see summary (36113423) of iMSMS (36113426) consortium)**

We thank the reviewer for this important comment; the points raised by the reviewer are well justified. We agree that our study nicely builds on earlier observations made for the association of *A. muciniphila* with MS, which, as the reviewer suggests, are nicely listed in the summary (PMID: 36113423) of iMSMS (PMID: 36113426) consortium. We apologize that our response to the points raised by the reviewer are very lengthy, however we hope that, by discussing these points in detail, we avoid any further confusion or misconception. In the following text, we will specifically address three points: (1) why we used *Muc2*^{-/-} mouse model; (2) the aspect of FMT and the influence of the host genotype background; and (3) the question of *A. muciniphila* in *Muc2*^{-/-} mice.

Use of *Muc2*^{-/-} mice: We agree that the justification for the use of *Muc2*^{-/-} mice (i.e. to simulate the variation in microbial communities in a typical human cohort using a genetically controlled model) was not well conveyed in the earlier version of the manuscript. Consequently, we substantially edited the text referring to **Fig. 1** and **Extended Data Fig. 2**. To account for the changes in the text, we also modified **Fig. 1** and **Extended Data Fig. 2**. Importantly, the underlying *datasets are the same* and no new data were added. However, we removed data on the microbiota composition during EAE development, as these data did not add valuable information to the story and may have contributed to the confusion in the earlier version of the manuscript.

To justify the use of mice of varying genetic backgrounds in general, we rephrased the scientific background as follows (lines 57-59):

To evaluate common approaches to identify disease-associated commensals, we implemented a prospective cohort study design in mice of varying genetic backgrounds and diets, harbouring distinct complex microbial communities (**Extended Data Fig. 1**).

We also rephrased the rationale behind inducing EAE in mice harboring three distinct microbiota composition “makeups”, as defined by the presence/absence of microbial taxa. As the origin and genetic background of a given mouse impacts the ability of certain microbes to colonize a given host, we decided to use mice of three different origins and genetic backgrounds to achieve this goal:

- C57BL/6J wildtype mice, which were purchased from Charles River, 3 weeks prior to induction of EAE.
- *Muc2*^{+/+} wildtype mice, which are homozygous for the presence of the *Muc2* gene. These mice were the offspring of heterozygous *Muc2*^{+/-} breeding pairs. *Muc2*^{+/+} did not display any apparent physiological differences compared to C57BL/6J wildtype mice.
- *Muc2*^{-/-} mice, which lack the gene encoding MUC2. As this glycoprotein is a major component of the mouse intestinal mucus barrier, we expected a severely different microbiota composition due to alterations in the niche for mucin-degrading commensals.

In this context, *Muc2*^{-/-} mice acted as *controls*. The deletion of the *Muc2* gene results in a drastic shift of the microbiota composition as well as in physiological alterations (see section II.3). However, using these mice allowed us to evaluate two possible scenarios:

- If the host genetic background did not matter to shortlist microbial risk factors, inclusion of *Muc2*^{-/-} mice into our shortlisting approach should not affect the output of microbial risk factors.
- If the genetic background did matter, we would have to come up with an alternative way to identify disease-prone microbiota compositions. Such an alternative way would account for discrepancies in host genetics. This scenario relates to our main hypothesis of this paper, which is that microbial risk factors of disease (e.g. taxa presence or abundance) cannot be generalized unless they consider bi-directional interactions with the host or the surrounding microbial community. Eventually, we were able to test this

27

hypothesis using an alternative approach to predict the disease-mediating properties of the microbiota in a given host (IgA coating indices). Importantly, IgA coating indices were suitable to predict EAE development also in *Muc2*^{-/-} mice harboring a complex microbiota (see new **Fig. 6** and **Extended Data Figs. 9, 10**). See section IV of this rebuttal letter for more details.

To further modify pre-existing microbiota compositions, a subgroup of mice were fed a fiber-free (FF) diet. This approach was intended to modify the relative abundances of a shared core microbiota between standard chow (FR)-fed and FF-fed wildtype mice (both C57BL/6J, from Charles River and *Muc2*^{+/+}). As we did not intend to evaluate the effect of dietary fiber on the progression of EAE, we significantly re-phrased the text in the Results section referring to the dietary aspect. We combined all five emerging groups of mice, as defined by the combination of genetic background and diet, into one analysis. In the previous version of the manuscript, we split the panels according to the genetic background, which overemphasized the dietary aspect and contributed to the misunderstanding of our aims.

FMT to evaluate role of microbiota in *Muc2*^{-/-} mice: At first, it seems reasonable that cross-transplantation of microbiota between WT and *Muc2*^{-/-} mice could shed light on the functional implications of either WT or *Muc2*^{-/-}-adapted microbiota on EAE disease progression. Here, we would like to elaborate why we intentionally *did not* perform this experiment in the framework of the previously submitted version of the manuscript, and why we eventually decided to perform it for the revised version of the manuscript.

- Host genetics crucially influence the microbiota composition in humans and mice (Nat Genet. 2016. doi: 10.1038/ng.3663; Science, 2016, doi:10.1126/SCIENCE.AAD9379). Consequently, we expected a re-shaping of a transplanted donor microbiota according to the genetic context of the recipient host. This is particularly the case when genetics between donor and recipient differ in factors that directly impact the intestinal microbiota, as is the case in *Muc2*^{-/-} mice. Thus, we expected the engrafted microbiota in recipient mice to differ significantly from the original donor microbiota, which would then influence the subsequent EAE disease course. This effect was convincingly demonstrated in a study where the authors performed microbiota cross-transplantation between WT and *Il17*^{-/-} mice (Sci Immunol. 2021. doi: [10.1126/sciimmunol.aaz6563](https://doi.org/10.1126/sciimmunol.aaz6563)). Consequently, we questioned the significance of a fecal microbiota transfer experiment to evaluate disease promoting properties of *Muc2*^{-/-} and WT microbiota as the engrafted microbiota would likely not reflect the composition of the donor microbiota. Thus, no reliable conclusion on the disease-mediating properties of the donor microbiota could be made.

28- We elaborate in detail on the reasoning for performing this cross-transplantation experiment in section IV of this rebuttal letter. Results of this new experiment confirmed our initial reluctance to perform this experiment, as the engrafted recipient microbiota provided significant alterations based on the recipient host. We found that the host genotype was more important for the final composition of the engrafted microbiota than the composition of the initial donor microbiota. We also found that host genotype was more important for EAE development than the composition of the initially transplanted microbiota. These decisive, host-mediated effects, however, were reflected by the IgA coating indices of select commensals within the engrafted microbiota, as outlined in section IV.

Host contribution to EAE development in *Muc2*^{-/-} mice: We have substantially rephrased the rationale for this particular experiment in the revised version of the manuscript and have further elaborated on this concern in the above section as well as in section IV.

Rationale to focus on *Akkermansia* without referring to *Muc2*^{-/-} mice: We agree with the reviewer that the role of *Akkermansia* in the context of murine and human autoimmune neuroinflammation has already been addressed by multiple papers. The reported controversial results alone would probably justify investigating the role of *Akkermansia* in more detail. However, we are convinced that our experiments described in **Fig. 1** further elucidate the controversial role of *Akkermansia* for the following reasons:

- In SPF (but not gnotobiotic) experiments, our group observed a negative association of *Akkermansia* with EAE progression in experiments performed by the *same* persons and using the *same* tools, resources, analysis methods and reagents.
- In the gnotobiotic setting, we found that the detrimental properties of *Akkermansia* (disease-associated vs disease-preventing) were mediated by the surrounding background microbiota.
- Consequently, we provide statistical evidence that neither the relative abundance of *Akkermansia* (**Fig. 5h,i**), nor presence or absence of it (**Fig. 5j**), are reliable universal disease predictors and the potential disease-mediating role of *Akkermansia* has to be seen in the context of a given host–microbiota combination.

Having observed both effects (*Akkermansia*=disease-associated and *Akkermansia*=disease-preventing) ourselves in one study significantly influenced design of subsequent experiments and analysis approaches. Results of these experiments and analyses led us to conclude that is not

29

possible to attribute disease-promoting or -preventing properties to a given microbiota composition in a certain host based on a single bacterial taxon alone. In summary, this conclusion is more logically conveyed when *Muc2*^{-/-} mice experiments are kept in the manuscript. Additionally, we would like to highlight that this study does not focus on disease-mediating properties of *Akkermansia* or its type strain *A. muciniphila*. *Akkermansia* was used only as a model commensal to illustrate downfalls of commensal risk factor evaluation. It was selected based on initial results from experiments shown in **Fig. 1** and based on the fact that it is already connected to MS by several studies (see Introduction). We do not indicate that our results in mice necessarily apply to *Akkermansia* in humans, but our study instead serves as an important first step to consider interactions of individual taxa with and the background microbiotas in human MS disease.

- 3) It would be important and helpful for the reader if the 5 criteria to define the potential *A. muciniphila*-associated and disease-mediating metabolite are briefly discussed in the main text.**

We completely understand the reasoning behind this suggestion. However, we would like to point out that *Nature Microbiology* has a limit of 4500 words and we are already at the limit. In an effort to find a compromise, we describe our criteria in detail in the Methods section and explicitly refer to this section in the main text.

Please also note that we elected to remove the second criterion (contribution to axis PC2) because our aim was to identify potential *A. muciniphila*-associated metabolites, as a logical consequence of the experimental setup. Following the removal of that criterion, the metabolite-of-interest should still meet the following four criteria: (1) significant difference in concentration between SM or diet groups, (2) significant correlation with overall disease course, (3) significant correlation with presence or absence of *Akkermansia*, and (4) significant difference in concentration upon EAE induction.

In the manuscript, the text where we first mention this analysis pipeline now reads as follows (lines 128-130 in the revised version of the manuscript):

To identify such potential EAE-impacting metabolites, we developed a metabolite-of-interest screening pipeline including 18 independent analyses to evaluate four different criteria (**Extended Data Fig. 5a–c**; see Methods section “Metabolite-of-interest screening pipeline”).

- 4) The authors identify *A. muciniphilia* as a risk factor that is linked with more severe EAE disease- it would be important to test whether this is linked with an increased MOG-antigen specific Th1/17 or Tc1/17 response and IgA plasma cell response in spinal cord/CNS**

30We thank the reviewer for this interesting suggestion. As shown in the new **Extended Data Fig. 7e**, we find more CD3⁺CD4⁺IL-17⁺ and CD3⁺CD4⁺IL-17⁺IFN γ ⁺ cells in the spinal cords in mice with severe disease (Cluster 1 mice). These cell types are usually considered to be the main mediators of demyelination in the spinal cords of mice. It is true, though, that we did not check whether these are exclusively MOG-reactive T cells, i.e. by stimulation with the MOG peptide and determining Ki67 expression on CD4⁺ cells.

We agree that it would be interesting to further investigate plasma cell responses, especially in the context of our findings regarding IgA coating indices (ICIs). Unfortunately, we determined ICIs to be microbiota-related disease predictors only after dissection of all used animals and we did not include any plasma cell specific markers in the staining panel for our ex vivo analyses. Consequently, we cannot address this point for the mice reported in the manuscript, as new mouse experiments would be required. However, we will definitely include this analysis in our follow-up studies.

5) Is γ -amino butyric acid produced by *A. muc* and sufficient to worsen EAE when administered alone?

In a recently published paper (<https://doi.org/10.1128/aem.01121-23>), Konstanti *et al.* reported that Amuc is capable of producing GABA under certain environmental conditions. This paper was added to our manuscript's bibliography and cited in the Discussion (see also section II.6). Using our metatranscriptomic data, we investigated whether the presence or absence of *A. muciniphila* led to a significant difference in abundance of transcripts corresponding to glutamate decarboxylase (GAD), which converts glutamate to GABA. Although approximately 16% of transcripts for the GAD enzyme were contributed by *A. muciniphila* in the SM14 mice, the overall levels of this transcript were unchanged in the absence of *A. muciniphila* (SM13) (data not shown). Thus, we carefully phrased the observed association between the presence of *A. muciniphila* and increased cecal GABA concentrations and we refrained from claiming a direct link. When addressing the effect of GABA on EAE, previous studies heavily focused on GABA-mediated signaling in neurological tissues and environments. One study, however, investigated the role of orally administered GABA during EAE (Sci Rep. 2021. doi: [10.1038/s41598-021-84751-3](https://doi.org/10.1038/s41598-021-84751-3)). They did not detect any disease-ameliorating or -promoting effect of GABA, while homotaurin, another GABA receptor antagonist, provided beneficial effects. The authors suggest that this might be due to GABA not being able to cross the blood–brain barrier, while homotaurine does. We do not know whether the increased GABA concentrations in the cecum of EAE-prone mice have any mechanistic implications. The rephrased GABA-related section in the Discussion is quoted in section II.6.

- 6) **The authors findings suggest that IgA coating levels of *Bacteroides ovatus* represents a potential measure to predict EAE severity. However experimental validation of this prediction needs to be performed such as colonization with IgA⁺ *B. ovatus* compared to IgA⁻ *B. ovatus*. It would be important whether the IgA coating index of *B. ovatus* is connected with an increased MOG-antigen specific Th1/17 or Tc1/17 response and IgA plasma cell response in spinal cord/CNS.**

We thank the reviewer for this intuitive suggestion for a new experiment. The reviewer suggests to colonize mice with IgA⁺ *B. ovatus* and IgA⁻ *B. ovatus*, followed by EAE induction. In general, such an experiment could be performed either through mono-association or by embedding BO⁺ or BO⁻ into a microbial background community of choice, i.e. a SM14-derived community. While the idea of this proof-of-concept experiment is sound, it comes with considerable shortcomings that might confound the results and conclusions. We would like to briefly summarize these aspects here:

- Obtaining IgA⁺ and IgA⁻ *B. ovatus* fractions for colonization is not trivial. It would require mono-association of GF mice with *B. ovatus* to allow for physiological *in vivo* IgA-coating, followed by separation of coated and non-coated fractions from fecal samples under anaerobic conditions to keep bacterial cells alive.
- Re-introducing IgA⁺ and IgA⁻ *B. ovatus* fractions into mice would very likely change the coating rates of these fractions. IgA⁻ *B. ovatus* will probably not remain uncoated and some IgA⁺ *B. ovatus* might lose their IgA coating, at least to a certain degree. As reported in the manuscript, we have already demonstrated that coating of *B. ovatus* with IgA strongly depends on: (1) the individual host and (2) the background microbiota (**Fig. 5k** in the revised version of the manuscript). Consequently, the anticipated conclusion cannot be made from such an experiment, as coating with IgA is not a static phenomenon, but rather a dynamic process shaped by a particular host and influenced by other microbes within the microbial community.
- An alternative to overcome this possible “re-coating” effect would be using genetically modified mice with impaired IgA production. Indeed, various IgA-deficient mouse lines exist (i.e.: J Immunol1999. doi: [10.4049/jimmunol.162.5.2521](https://doi.org/10.4049/jimmunol.162.5.2521) or Gut. 2022. doi: [10.1136/gutjnl-2020-322873](https://doi.org/10.1136/gutjnl-2020-322873)). However, these mice exhibit a severely disturbed immune system and are prone to intestinal inflammation. Thus, we consider these mice suboptimal for testing the experiment suggested by this Reviewer.
- We do not state that IgA⁺ *B. ovatus* directly or indirectly promotes autoimmune neuroinflammation. Our main statement is that the coating behavior of *B. ovatus* only

32reflects the bi-directional interactions between a microbial network in a given host, introducing the concept of a “reporter species”.

Minor comments

n/a

Comments on data integrity and reproducibility (points #7 & 8 below)

- 7) **Fig 1 concerning data using Muc2 deficient and sufficient mice. An n of 4-5 seems insufficient to come to firm conclusions. The n number needs to be increased.**

We have used a higher number of animals for Charles River (CR) mice in **Fig. 1**, and it turned out that such high numbers were not essential for experiments with Muc2-deficient and -sufficient mice. We observed consistent EAE disease phenotypes among wild-type mice (CR and *Muc2*^{+/+}) and consistently reduced disease in *Muc2*^{-/-} animals (**Fig. 1f**), which is beyond what would be expected by chance. We therefore do not find increasing the sample size for these groups justified.

- 8) **Fig 2g- the authors state that there is a FF diet induced significant increase of Akkermansia muciniphilia however fail to provide a test for statistical significance-this should be done.**

We address this point in more detail in the description of **Fig. 4** and provide a statistical test in **Fig. 4a**.

IV. Reviewer #3

- 1) **The paper by Desai in colleagues seeks to identify predictors of disease development using a mouse model of MS. The group uses controlled restricted communities to determine how composition, abundance of certain members, metabolites and immune parameters correlate with disease severity. The conclusion is that IgA coating of one member, *B. ovatus*, seemed to consistently correlate with disease severity. Overall, the use of controlled communities provides a nice lesson in microbial ecology and is a well controlled system for identifying potential correlates of disease. The manuscript makes an important point that community composition matters and output of disease might change based on community and/or function. The use of the metatranscriptomics is of interest here as more of the community needs to begin to look at how the function of the microbiota changes rather than just membership. The data are well presented and good group sizes are appropriate. However, while the use of these controlled systems is nice they should be complemented by validating their findings in the context of the more natural and complex community. Without this, this study reflects what happens in a contrived 12-14 member community and likely does not reflect reality. This fact makes the conclusions of this manuscript much less impactful. Moreover, one of the main goals of this study was to identify “predictors” of disease. Indeed, in the clinic diagnosis and treatment regime of this disease it would be nice to have a readout of disease severity. Again, the identification that IgA coating of *B. ovatus* is a predictor of disease is only supported in this contrived community. What happens in a more complex community as what would be seen in humans. Does IgA coating of *B. ovatus* still act as a “predictor”? Based on this, the language and conclusions drawn from this dataset are oversold and the use of predictor is far too strong of a statement to be made from this experimental design.**

We would like to thank the Reviewer for pointing out a number of key strengths of our study. We also took a good note of the comment by this Reviewer about potentially validating our findings in a more complex microbial community. Thus, we decided to perform an additional experiment to evaluate our predictor hypothesis in mice harboring complex communities (see new **Fig. 6** and **Extended Data Figs. 9, 10**). This experiment had to fulfill multiple criteria to make meaningful conclusions about the use of a common reporter species' IgA coating index (ICI) as an individual predictor for EAE development in mice. These criteria were as follows:

- Predictability across distinct complex microbiota compositions.
- Predictability across distinct genetic backgrounds.

34

- Experiment had to follow animal protocols that were already approved by the Luxembourgish authorities. Obtaining approval for a new animal protocol and then performing the experiments would have taken well over a year to resubmit the manuscript. Since we were given a timeline of six months to submit the revised submission, this is an important aspect to consider.

To satisfactorily address the Reviewer's comment, while still considering the above three criteria, we performed the experiment as follows:

- We used mice of two different genetic backgrounds: C57BL/6J wildtype mice and *Muc2*^{-/-} mice. *Muc2*^{-/-} mice were used to challenge our ICI prediction model and to verify that it is superior to the approach outlined here. So far, we have only tested our prediction model in gnotobiotic mice of the same genetic background. We proposed that our prediction model allowed for individual disease course prediction irrespective of the host characteristics. Thus, the model should also predict disease development in hosts with drastically altered genetics, which would further strengthen our model. Our initial conclusion, as shown in **Fig. 1** ("higher abundances of *Akkermansia* are associated with less severe EAE across genetically distinct mice harboring a complex microbiota"), was based on experiments including *Muc2*^{-/-} mice. Consequently, we aimed to demonstrate that our prediction model worked in mice with which we obtained the initial—ultimately misleading—conclusion and also accounts for mice with drastic genetic modifications.
- To achieve distinct complex microbiota compositions (without using multiple distinct mouse origins or genotypes), we decided to use a fecal microbiota transfer approach. To do so, we transferred microbiota from SPF-housed *Muc2*^{+/+} or *Muc2*^{-/-} mice to germ-free (GF) C57BL/6J or *Muc2*^{-/-} mice. Performing all possible cross-transplantations, we ended up with 4 distinct microbiota donor/recipient combinations (new **Fig. 6a**), harboring distinct microbial taxa within their microbiota (new **Extended Data Fig. 9a**). We performed microbiota transfer by housing GF mice in spent litter of SPF-housed mice for 21 days. Details of this procedure are described in the Methods section. This microbiota transfer approach was chosen due to animal protocol-related restrictions, as we did not have approval to perform intragastric gavage on these particular mice. We are aware that this approach generally leads to less re-colonization of transplanted microbes in recipient mice compared to fecal microbiota transplantation through intragastric gavage of fecal slurries. However, this restriction did not affect our final conclusions, as our goal was not to evaluate disease-mediating properties of one defined complex microbiota.

We induced EAE in a total of 13 mice of which at least 3 mice belonged to the same donor/recipient combination (new **Fig. 6a**). Since we did not intend to perform any group-wise analysis, the number of mice per donor/recipient combination was not based on a statistical power calculation for such group-wise comparisons. Instead, our main readout was correlation analyses of individual species-specific ICIs with individual EAE-associated measures. Thus, we used the following four criteria to define the number of mice to be used for this experiment, all of which had to be fulfilled:

- 1) Comparable number of mice within each donor/recipient combination to avoid bias towards a given combination when performing individual correlation analysis.
- 2) Providing a sufficient number of individual data points to achieve significance for either Pearson's or Spearman's correlation analysis. Commonly, seven individual data points are considered to be the required minimum number to perform proper correlation analysis.
- 3) We had to consider that probably only a limited number of species would be present in all recipient mice. However, as long as a particular species was present in at least one mouse of one donor/recipient combination, the ICI of this species was considered fit for evaluation of its disease prediction qualities.
- 4) The number of available mice on a pre-authorized animal protocol (also keeping in mind a six-month timeline provided for the revision of this manuscript).

Considering all the above-mentioned criteria, we set the total number of recipient mice at 13.

To obtain sequencing results with proper resolution on a species level, we performed *full-length* 16S rRNA gene amplicon sequencing (MinION) followed by taxonomic annotation using the Emu 3.0+ database to obtain reliable relative abundances of bacterial taxa on a species level. Results of the experiment and the related conclusions are shown in **Fig. 6a–h** and **Extended Data Figs. 9, 10**. **Fig. 6a–h** replaces the previous *Fig. 6*, which is now merged with the previous *Fig. 5*. In other words: previous *Fig. 6* + previous *Fig. 5* = new **Fig. 5**; the new data generated during revision = **Fig. 6** (and **Extended Data Figs. 9, 10**).

Of the 71 bacterial species, which were part of a shared core microbiome between all four donor/recipient combinations, we identified 3 bacterial species, whose ICIs before EAE induction correctly predicted disease development in each individual. These species were: *Eubacterium coprostanoligenes*, *Phocaeicola dorei* and *Enterocloster bolteae*. Unfortunately, we could not

assess prediction quality of *B. ovatus* ICI in mice harboring the tested complex microbiota compositions, as *B. ovatus* prevalence across all mice was too low (**Extended Data Fig. 10f**). The only SM14-constituent that was sufficiently prevalent to calculate correlations was *A. muciniphila*, and in line with findings using mice harboring a reduced microbiota (**Fig. 5I**), *A. muciniphila* ICI was disconnected from subsequent EAE development (**Extended Data Fig. 10g**).

At the same time, we would like to highlight that although we discovered an association of *B. ovatus* ICI to EAE disease severity discovered using the synthetic communities, it is not unexpected that the same bacterium (*B. ovatus*) would not be a predictor in the complex communities. Going from a complex mouse community to the human gut microbiota, such a “reported species” might further change. The important point, however, is that the overall concept of using ICI of a strain before inducing disease to predict disease is successfully validated in a complex microbial community. From another point of view, having tested such a concept using reduced communities, in the face of an array of readouts, makes our conclusions fairly robust. We have mentioned these points in the Discussion (lines 336-344):

After extensive evaluation of multiple microbiota-associated readouts, we found that the IgA coating index (ICI) of certain commensals reflects these disease-driving host–microbiota interactions. Thus, the ICIs of these species before disease onset allowed us to correctly predict the individual EAE progression across all microbiota–host combinations. However, due to minimal core microbiota overlap between reduced and complex communities, we could not single out the ICI of one particular species as a universal predictor: while we identified *Bacteroides ovatus* ICI in reduced communities, we determined ICIs of *Eubacterium coprostanoligenes* and *Phocaeicola dorei* to be reliable predictors in complex communities in a mouse model. We propose that these species act as “reporter species”, reflecting the individual bi-directional host–microbiota influences on EAE progression.

In conclusion, we sincerely thank the reviewer for suggesting this experiment, which we agree was absolutely necessary to shore up the implications of our previous conclusions to a complex microbiota setting. These new data and conclusions have helped us to sufficiently address the reviewer’s concerns. Altogether, our results provide a potential direction for the applicability of the microbiome data combined with host responses (sIgA) in research on multiple sclerosis.

Decision Letter, first revision:

Message: Our ref: NMICROBIOL-23061597A

9th May 2024

Dear Mahesh,

Thank you for your patience as we've prepared the guidelines for final submission of your Nature Microbiology manuscript, "Gut microbiome-based prediction of autoimmune neuroinflammation" (NMICROBIOL-23061597A). Please carefully follow the step-by-step instructions provided in the attached file, and add a response in each row of the table to indicate the changes that you have made. Ensuring that each point is addressed will help to ensure that your revised manuscript can be swiftly handed over to our production team.

In recognition of the time and expertise our reviewers provide to Nature Microbiology's editorial process, we would like to formally acknowledge their contribution to the external peer review of your manuscript entitled "Gut microbiome-based prediction of autoimmune neuroinflammation". For those reviewers who give their assent, we will be publishing their names alongside the published article.

Nature Microbiology offers a Transparent Peer Review option for new original research manuscripts submitted after December 1st, 2019. As part of this initiative, we encourage our authors to support increased transparency into the peer review process by agreeing to have the reviewer comments, author rebuttal letters, and editorial decision letters published as a Supplementary item. When you submit your final files please clearly state in your cover letter whether or not you would like to participate in this initiative. Please note that failure to state your preference will result in delays in accepting your manuscript for publication.

Cover suggestions

COVER ARTWORK: We welcome submissions of artwork for consideration for our cover. For more information, please see our guide for cover artwork.

Nature Microbiology has now transitioned to a unified Rights Collection system which will allow our Author Services team to quickly and easily collect the rights and permissions required to publish your work. Approximately 10 days after your paper is formally accepted, you will receive an email in providing you with a link to complete the grant of rights. If your paper is eligible for Open Access, our Author Services team will also be in touch regarding any additional information that may be required to arrange payment for

38your article.

Please note that *Nature Microbiology* is a Transformative Journal (TJ). Authors may publish their research with us through the traditional subscription access route or make their paper immediately open access through payment of an article-processing charge (APC). Authors will not be required to make a final decision about access to their article until it has been accepted. Find out more about Transformative Journals

Reviewer #1:
Remarks to the Author:
The authors addressed most of our concerns.

Reviewer #2:
Remarks to the Author:
The authors sufficiently addressed my concerns. Thus, overall this MS will be a resource for several future MS. Important for the scientific community is an addition of a "limitations of study section" discussing several of the raised concerns that could not be addressed/validated.

Author Rebuttal, first revision:

Response to Reviewers (10 May 2024)

39Manuscript ID: NMICROBIOL-23061597A

Manuscript Title: Gut microbial factors predict disease severity in a mouse model of multiple sclerosis

Reviewer #1:

Remarks to the Author:

The authors addressed most of our concerns.

We appreciate the confirmation from this reviewer.

Reviewer #2:

Remarks to the Author:

The authors sufficiently addressed my concerns. Thus, overall this MS will be a resource for several future MS. Important for the scientific community is an addition of a "limitations of study section" discussing several of the raised concerns that could not be addressed/validated.

We appreciate these final remarks from the reviewer. Please note that, due to word limits, we do not elaborate limitations in a separate section as suggested. However, we have added a brief statement regarding the limitations of the study related to the previously requested validation experiments from reviewers (GABA/mono-colonization with reporter species), to the penultimate paragraph: *"A limitation of our study is that we have not mechanistically verified the associations of these reporter species or metabolites such as GABA with disease severity."* Please also note that we already discuss limitations in terms of translation and clinical relevance of the findings.

Final Decision Letter:**Message:** 14th June 2024

Dear Mahesh,

I am pleased to accept your Article "Gut microbial factors predict disease severity in a mouse model of multiple sclerosis" for publication in Nature Microbiology. Thank you for having chosen to submit your work to us and many congratulations.

Acceptance of your manuscript is conditional on all authors' agreement with our publication policies (see <https://www.nature.com/nmicrobiol/editorial-policies>). In particular your

41manuscript must not be published elsewhere.

Please note that *Nature Microbiology* is a Transformative Journal (TJ). Authors may publish their research with us through the traditional subscription access route or make their paper immediately open access through payment of an article-processing charge (APC). Authors will not be required to make a final decision about access to their article until it has been accepted. Find out more about Transformative Journals

With kind regards,